# Cell-type-directed design of synthetic enhancers

Ibrahim I. Taskiran[1,2,3], Katina I. Spanier[1,2,3], Hannah Dickmänken[1,2,3], Niklas Kempynck[1,2,3], Alexandra Pančíková[1,2,3,4], Eren Can Ekşi[1,2,3], Gert Hulselmans[1,2,3], Joy N. Ismail[1,3,5], Koen Theunis[1,2,3], Roel Vandepoel[1,2,3], Valerie Christiaens[1,2,3], David Mauduit[1,2,3] & Stein Aerts[1,2,3 ✉]

Transcriptional enhancers act as docking stations for combinations of transcription factors and thereby regulate spatiotemporal activation of their target genes[1]. It has been a long-standing goal in the field to decode the regulatory logic of an enhancer and to understand the details of how spatiotemporal gene expression is encoded in an enhancer sequence. Here we show that deep learning models[2–6], can be used to efficiently design synthetic, cell-type-specific enhancers, starting from random sequences, and that this optimization process allows detailed tracing of enhancer features at single-nucleotide resolution. We evaluate the function of fully synthetic enhancers to specifically target Kenyon cells or glial cells in the fruit fly brain using transgenic animals. We further exploit enhancer design to create 'dual-code' enhancers that target two cell types and minimal enhancers smaller than 50 base pairs that are fully functional. By examining the state space searches towards local optima, we characterize enhancer codes through the strength, combination and arrangement of transcription factor activator and transcription factor repressor motifs. Finally, we apply the same strategies to successfully design human enhancers, which adhere to enhancer rules similar to those of *Drosophila* enhancers. Enhancer design guided by deep learning leads to better understanding of how enhancers work and shows that their code can be exploited to manipulate cell states.

Cell-type-specific expression of a target gene is achieved when a unique combination of transcription factors (TFs) activates a specific enhancer; whereas this enhancer remains either passively (default-off[7,8]) or actively repressed in other cell types (for example, through repressor binding[9] or corepressor/polycomb recruitment). Typically, when an enhancer is translocated to another chromosome or to an episomal plasmid, it maintains cell-type-specific control of its nearby reporter gene[1,10]. Therefore, its regulatory capacity is contained in the enhancer DNA sequence and has co-evolved to respond uniquely to a specific *trans*-environment in a cell type. A thorough understanding of how enhancer activation is encoded in its DNA sequence is important, as it is a key component for the modelling and prediction of gene expression[2,11]; for the interpretation of non-coding genome variation[12,13]; for the improvement of gene therapy; and for the reconstruction and manipulation of dynamic gene regulatory networks underlying developmental, homeostatic and disease-related cell states.

Many complementary approaches and techniques have been used to decode enhancer logic[1]. These include studies of individual enhancers by mutational analysis[14–16], in vitro TF binding (for example, electrophoresis mobility shift assay), cross-species conservation[17] and reporter assays. The upscaling of such studies led to the identification of common features of coregulated enhancers[18–20]. These experimental findings also triggered the improvement of computational methods for the prediction of *cis*-regulatory modules, whereby feature selection and parameter optimization led to new insights into how binding sites cluster and how their strength (or binding energy) impacts enhancer function[15,16,21–24]. Wider adoption of genome-wide profiling of chromatin accessibility[25], single-cell chromatin accessibility[26–28], histone modifications[29,30], TF binding[31] and enhancer activity[19,32] led to significantly larger training sets of coregulated enhancers that could then be used for a posteriori discoveries of TF motifs and enhancer rules, aided by the growing resources of high-quality TF motifs[33,34]. Further mechanistic insight has been provided by thermodynamic modelling of enhancers[35,36], in vivo imaging of enhancer activity[37], the analysis of genetic variation through expression quantitative trait loci and chromatin-accessibility quantitative trait loci analysis[12,38] and high-throughput in vitro binding assays[39,40]. Recently, the enhancer biology field embraced the use of convolutional neural networks (CNN) and network-explainability techniques that again provided a substantiall leap forward in terms of prediction accuracy and syntax formulation[2–6,41–44].

An orthogonal strategy to decode enhancer logic is to engineer synthetic enhancers from scratch. This approach has the advantage that the designer knows exactly which features are implanted, so that the minimal requirements for enhancer function can be revealed. Recent work

[1]Laboratory of Computational Biology, VIB Center for AI & Computational Biology (VIB.AI), Leuven, Belgium. [2]VIB-KULeuven Center for Brain & Disease Research, Leuven, Belgium. [3]Department of Human Genetics, KU Leuven, Leuven, Belgium. [4]VIB-KULeuven Center for Cancer Biology, Leuven, Belgium. [5]Present address: UK Dementia Research Institute at Imperial College London, London, UK. ✉e-mail: stein.aerts@kuleuven.be

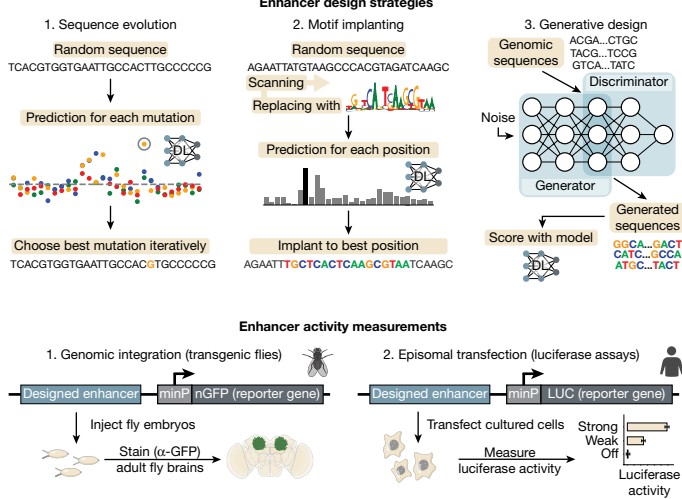

**Enhancer design strategies**

1. Sequence evolution
2. Motif implanting
3. Generative design

**Enhancer activity measurements**

1. Genomic integration (transgenic flies)
2. Episomal transfection (luciferase assays)

**Fig. 1 | Deep learning-based enhancer design.** Overview of enhancer design strategies and activity measurements of designed enhancers in *Drosophila* brains and human cell lines. DL, deep learning.

showed the promise of CNN-driven enhancer design by successfully designing yeast promoters[45] and by using a CNN to select high-scoring enhancers for S2 cells, from a large pool of random sequences[4]. Here, we tackle the next challenge in enhancer design: to design enhancers that are cell-type specific. To this end, we used previously trained deep learning models for which we have already validated the accuracy of nucleotide-level interpretation and motif-level predictions[12,5] (Supplementary Note 1). Using these enhancer models as a guide (or 'oracle'), we tested three different sequence design approaches[46,47] (Fig. 1).

## In silico evolutions

As a first strategy for enhancer design, we created synthetic enhancers to specifically target Kenyon cells (KC) in the mushroom body of the fruit fly brain, using a nucleotide-by-nucleotide sequence evolution approach[45] (Methods). This approach starts from a 500 base pair (bp) random sequence that is evolved from scratch (EFS) in silico towards a chosen cell type through several iterations. Prediction scores are calculated using DeepFlyBrain[5], a deep learning model trained on differentially accessible regions across several cell types of the *Drosophila* brain and that can recognize motif-level nucleotide arrangements for many cell types (Supplementary Note 1). At each iteration we performed saturation mutagenesis[13,44,48] whereby all nucleotides were mutated one by one and each sequence variation was scored by DeepFlyBrain to select the mutation with the greatest positive delta score for the KC class (among 81 classes representing different cell types that the model learned to predict). We performed this procedure starting from 6,000 GC-adjusted random sequences and observed that after 15 iterations, DeepFlyBrain KC prediction scores increased from around the minimal score (0) to nearly the maximum score (1), while remaining low for other cell types (Fig. 2a and Extended Data Fig. 1a,b). We found this greedy search to provide a good balance between computational cost and ability to efficiently yield high-scoring sequences, compared to alternative state space searches (Extended Data Fig. 2a–d and Methods).

Next, we investigated the initial (random) sequence and the specific paths that are followed through the search space towards local optima. For only a small fraction (3%) of random sequences, the prediction score remained below 0.5 even after 15 mutations (Extended Data Fig. 1c). These sequences were mostly characterized by more instances of repressor binding sites together with an increased number of mutations required to generate sufficient activator binding sites. A second observation is that, even though 500 bp space is given to the model, the

selected mutations accumulated in about 200 bp space, preferentially at the centre of the random sequence (Extended Data Fig. 1d,e).

We investigated the consequences of each mutation on shaping the enhancer code using DeepExplainer-based contribution scores (Fig. 2b; Methods). This revealed that initial random sequences harbour several short repressor binding sites by chance and these are preferentially destroyed during the first iterations (Extended Data Fig. 1f,g). These repressor sites contribute negatively to the KC class prediction and represent candidate binding sites for KC-specific repressor TFs such as Mamo and CAATTA[5]. The nucleotides with the highest impact represent mutations that destroy a repressor binding site and simultaneously generate a binding site for the key activators Eyeless (Ey), Mef2 or Onecut. Eventually, DeepExplainer highlighted several candidate activator binding sites, whereby Ey, Mef2 and Onecut sites dominate (Fig. 2b and Extended Data Fig. 1f,g).

To test whether the in silico evolved enhancers can drive reporter gene expression in vivo, we randomly selected 13 sequences after 10 or 15 iterations (Fig. 2c and Supplementary Figs. 1 and 2) and integrated them into the fly genome with a minimal promoter and a GFP reporter gene (Methods). Investigating the GFP expression pattern by confocal imaging showed that 10 of these 13 tested synthetic enhancers were active specifically in the targeted cell type, the KC (Fig. 2d and Extended Data Fig. 1h). Some enhancers did not show activity after ten mutations but became active after an extra five mutations (Fig. 2d, Extended Data Fig. 1i,j and Supplementary Fig. 3). The three enhancers without GFP signal in KC were found to also be Dachshund negative, indicating the potential loss of KC (Extended Data Fig. 1k). Using assay for transposase-accessible chromatin by sequencing (ATAC-seq) on the brains of the transgenic lines, we verified that the synthetic enhancers become accessible when integrated into the genome (Extended Data Fig. 1l), as predicted by the model.

We also generated transgenic lines to test enhancers at different steps during the evolutionary design process (Supplementary Figs. 4 and 5). We found that random sequences or sequences with only few mutations remain inactive, whereas enhancer activity is initiated when repressor sites are removed and Ey and Mef2 sites are generated; and activity further increases with more and stronger instances of activator motifs (Extended Data Fig. 1m,n).

To demonstrate that enhancers can be generated for other cell types, we started from the same random sequences as above and evolved them into perineurial glia (PNG) enhancers (Extended Data Fig. 2e). After 15 mutations, putative PNG repressor sites have been destroyed and activator sites have been generated (Fig. 2e and Supplementary Fig. 6). We validated six designed sequences by creating transgenic GFP reporter flies and confirmed that four were positive, as they drive GFP specifically in PNG cells (Fig. 2f and Extended Data Fig. 2f). Because the same random sequence was evolved into either KC or PNG enhancers, this experiment underscores that the chosen mutations and the candidate binding sites they destroy or generate, causally underlie the activity of these synthetic enhancers.

Given that KC enhancers can arise from random sequences after 10 or 15 mutations, we proposed that certain genomic regions may require even fewer mutations to acquire KC enhancer activity. We scanned the entire fly genome and identified regions with high prediction scores but without chromatin accessibility in KC (Extended Data Fig. 2g,h; Methods). By applying sequence evolution to these sequences, three of four sequences became positive KC enhancers with only six mutations (Fig. 2g,h, Extended Data Fig. 2i,j and Supplementary Fig. 7). When the negative enhancer was further evolved, with an extra five mutations, it also became positive (Fig. 2g and Extended Data Fig. 2i,j). This suggests that KC enhancers, and probably other cell-type enhancers as well, can arise de novo in the genome with few mutations.

To summarize the changes that happened during the design process, we performed motif discovery across all 6,000 sequences, at each step of the optimization path (Extended Data Fig. 1f,g). This confirmed that

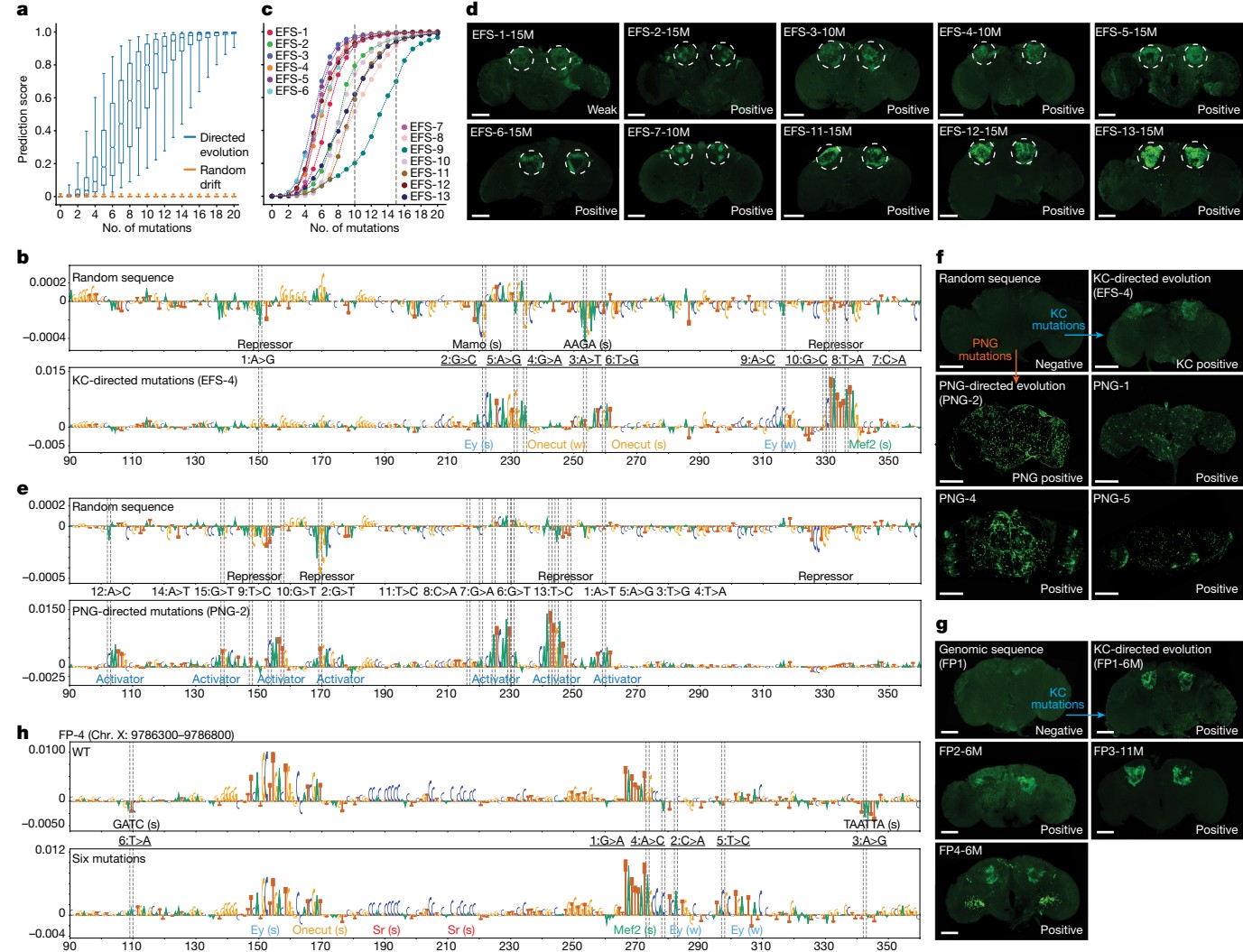

**Fig. 2 | In silico sequence evolution towards functional enhancers.**
**a**, Prediction score distribution of the sequences for the γ-KC class (*n* = 6,000 sequences) after each mutation. The box plots show the median (centre line), interquartile range (box limits) and 5th and 95th percentile range (whiskers) for KC-directed (blue) or random drift (orange) mutations. **b**, Nucleotide contribution scores for the γ-KC class of a selected random sequence in its initial form (top) and after in silico evolution (bottom). **c**, Prediction scores of 13 selected sequences at each mutational step. Dashed line indicates the selected iteration (10th or 15th mutation). **d**, In vivo enhancer activity of the cloned KC sequences with positive enhancer activity. **e**, Nucleotide contribution scores for the PNG class of the same selected random sequence as in **b** (top) and after PNG-directed mutations (bottom). **f**, In vivo enhancer

activity of the cloned PNG sequences. Top-middle, initial random sequence; top-left, random sequence after 10 mutations toward KC evolution; top-right, random sequence after 15 mutations toward PNG evolution; bottom, three other random sequences after mutations toward PNG evolution. **g**, In vivo enhancer activity of the cloned genomic sequences with 6 mutations (11 for FP3). **h**, Nucleotide contribution scores of a selected genomic sequence in its initial form (top) and after six iterations (bottom). In **b**,**e**,**h**, dashed line shows the position of the mutations, the mutational order and type of nucleotide substitutions are written in between top and bottom plots and motif annotation is indicated with strong (s) or weak (w) motif instances. In **d**,**f**,**g**, the expected location of KC is shown with dashed circles. Scale bars, 100 μm.

repressor sites are often present in random sequences and that they are preferentially destroyed during the first steps of the search algorithm. To experimentally test that these short repressor sites functionally cause repression, we selected three positive synthetic enhancers and three of the near-enhancers rescued from the genome and evolved these to become non-functional by manually choosing the mutations that decrease the prediction score by creating repressor binding sites (Extended Data Fig. 2i and Supplementary Figs. 8 and 9). We avoided mutating any of the predicted activator sites (Fig. 3a); thus, placed repressor motifs in between activator sites. New transgenic lines with these sequences integrated into the genome confirm that all tested enhancers have entirely lost their activity (Fig. 3b). This shows that enough repressor sites can dominate over a functional combination of activator sites.

The sequence evolution strategy thus represents an intuitive and efficient approach to generate cell-type-specific enhancers and to characterize their functional constituents.

## Multiple cell-type codes

A single enhancer can be active in several different cell types[49], and our earlier work suggested that this can be achieved by enhancers that contain several codes for different cell types, intertwined in a single approximately 500 bp sequence[5]. On the basis of this finding, we wondered whether a genomic enhancer that is active in a single cell type, could be synthetically augmented to become also active in a second cell type. To test this, we started with two optic lobe enhancers (*amon* and *CG15117*) that are accessible and active in T4/T5 and T1 neurons,

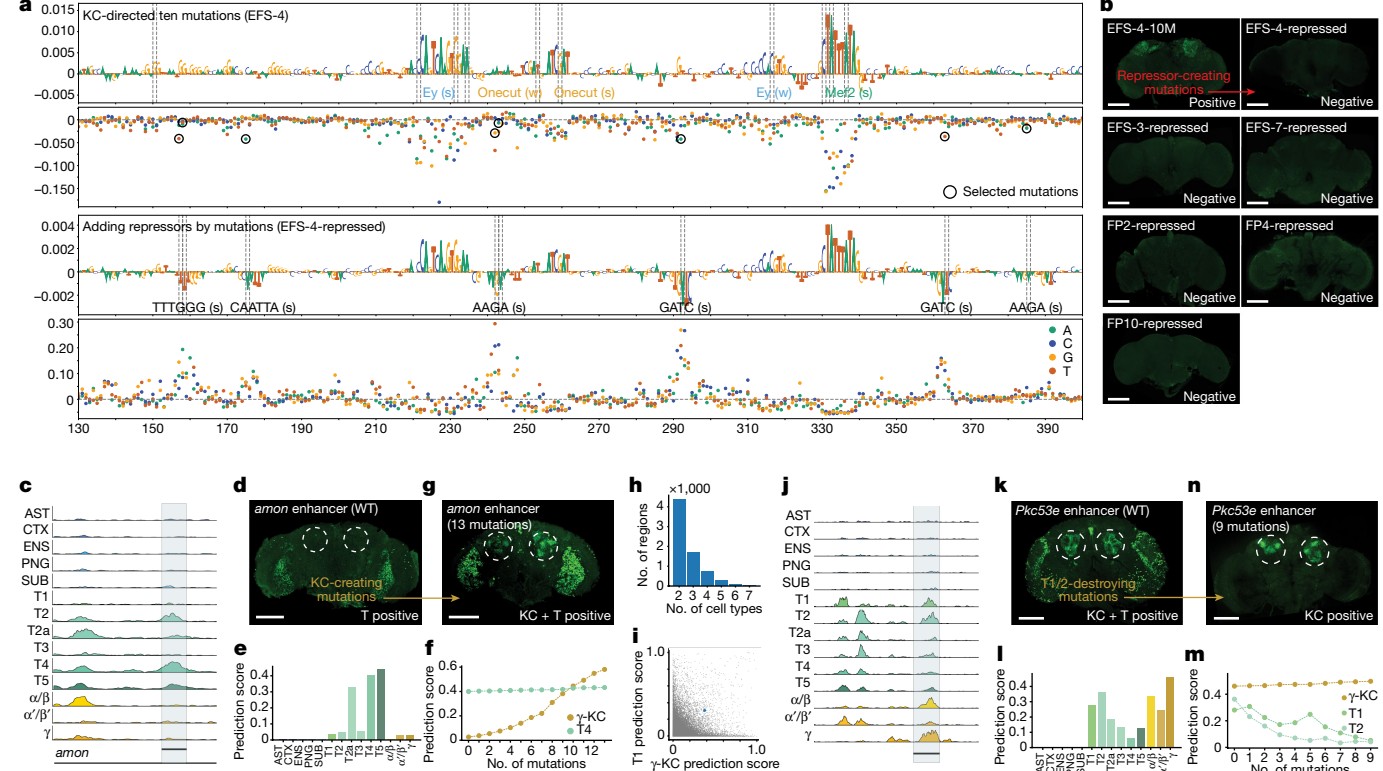

**Fig. 3 | Spatial expansion and restriction of enhancer activity. a**, Nucleotide contribution score and delta prediction score for in silico saturation mutagenesis of the EFS-4 enhancer after ten mutations (first and second row) and after adding repressors (third and fourth row). Dashed line shows the position of the mutations. Black circles, selected mutations to generate repressor sites. Motif annotation is indicated with strong (s) or weak (w) motif instances. **b**, In vivo enhancer activity of enhancers before (top-left) and after adding repressor sites. **c**, Chromatin accessibility profile near the *amon* gene. **d**, In vivo enhancer activity of the *amon* enhancer. **e**, The *amon* enhancer prediction scores for each cell type. **f**, Prediction scores for the γ-KC and T4 classes after each mutational step. **g**, In vivo enhancer activity of the *amon* enhancer after 13 mutations. The *amon* enhancer conserved exactly the same pattern of activity for T4 following

incorporation of the KC code. **h**, Number of regions that score high (>0.3) for several cell types. **i**, Comparison between γ-KC and T1 prediction score for the accessible regions in fly brain (*n* = 95,931). The selected region with high γ-KC and T1 prediction is highlighted with a blue dot. **j**, Chromatin accessibility profile of this region (*Pkc53e*) in several cell types. **k**, In vivo enhancer activity of the *Pkc53e* enhancer. **l**, *Pkc53e* enhancer prediction scores for each cell type. **m**, Prediction scores for the γ-KC, T1 and T2 classes after each mutational step. **n**, In vivo enhancer activity of the multi cell-type enhancer after nine mutations. In **b**,**d**,**g**,**k**,**n** dashed circles show the expected location of KC. Scale bars, 100 μm. AST, astrocytes; CTX, cortex glia; ENS, ensheathing glia; PNG, perineurial glia; SUB, subperineurial glia; T1–T5, T1–T5 neurons; α/β,α/β-KC; α′/β′, α′/β′-KC; γ, γ-KC.

respectively[5], and whose activity per cell type is also predicted correctly by DeepFlyBrain (Fig. 3c–e and Extended Data Fig. 3a–c). We then performed in silico evolution on these enhancers towards KC, while simultaneously maintaining a high prediction score for the original cell type. After 13 and 14 mutations, the enhancers were also predicted as KC enhancers but retained T4 and T1 binding sites. Testing the augmented sequences in vivo with a GFP reporter confirmed the spatial expansion of the enhancer activity to KC (Fig. 3f,g, Extended Data Fig. 3c–f and Supplementary Fig. 10; Methods).

Reciprocally, enhancers active in several cell types may be pruned towards a single cell-type code. We searched for genomic enhancers that score high for several cell types (Fig. 3h–l). We selected a *Pkc53e* enhancer that is accessible and active in both optic lobe T neurons and KC and predicted correctly by the model. This time, we drove the in silico evolution to maintain the KC prediction score, while decreasing the T neurons prediction score (Methods). After nine mutations, the sequence was predicted to have only KC activity (Fig. 3m). Nucleotide contribution scores show that the most important binding sites for KC were unaffected after nine mutations, whereas the activator binding sites were destroyed and new repressor binding sites were created for T neurons (Extended Data Fig. 3g). Testing the final sequence in vivo confirmed the spatial restriction of the enhancer activity (Fig. 3n). Together, our results indicate that, guided by the DeepFlyBrain model,

intertwined enhancer codes can be independently dissected and altered.

## Motif implantation

As a second strategy, we used a classical motif implantation approach to design KC enhancers. The rationale behind this strategy is based on our results above: nucleotide-by-nucleotide sequence evolution showed that all the selected mutations were associated with the creation or destruction of a TF binding site, rather than affecting contextual sequence between motif instances (Fig. 2b,e,h and Extended Data Fig. 3d,e,g). This suggested that a combination of appropriately positioned activator motifs, without the presence of repressor motifs, would be sufficient to create a cell-type-specific enhancer. Furthermore, we reasoned that by applying this design strategy to thousands of random sequences we could gain more insight into the KC enhancer logic. To this end, we iteratively implanted strong TF binding site instances in 2,000 random sequences, selecting locations with the highest prediction score towards the KC class. We first implanted a single binding site for one of the four key activators of KC enhancers, namely, Ey, Mef2, Onecut and Sr[5], and then specific combinations of sites in a particular implantation order (Extended Data Fig. 4a; Methods). This revealed that Ey and Mef2 had the strongest effect on the prediction score, while

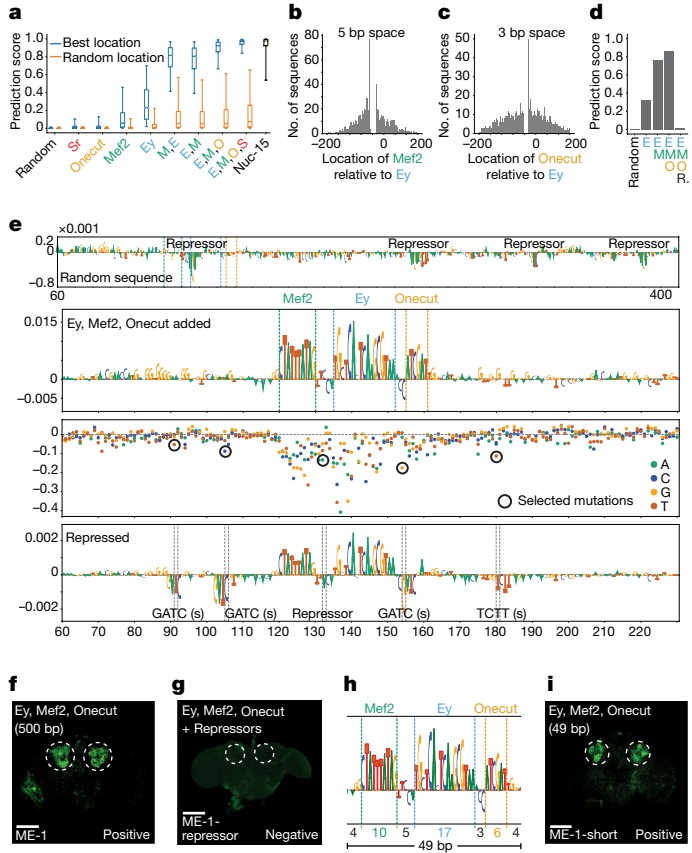

**Fig. 4 | Motif implantation towards minimal enhancer design. a**, Prediction score distribution of the sequences for the γ-KC class (n = 2,000 sequences) after each motif implantation at best location (blue), random location (orange) and after 15 mutations (Nuc-15). The box plots show the median (centre line), interquartile range (box limits) and 5th and 95th percentile range (whiskers). **b**, Distribution of Mef2 locations relative to Ey (n = 2,000). **c**, Distribution of Onecut locations relative to Ey (n = 2,000). **d**, Prediction scores for motif implanted sequence (ME-1) after each motif implanting and generation of repressor sites. **e**, Nucleotide contribution scores of ME-1 in its initial form (first track) and after Ey, Mef2 and Onecut implantations (second track). Dashed lines show the position of the motifs. Delta prediction score for in silico saturation mutagenesis (third track). Black circles, selected mutations to generate repressor sites. Nucleotide contribution scores after generation of repressor sites (fourth track). Dashed lines show the position of the mutations. **f,g**, In vivo enhancer activity of the cloned 500 bp sequence with Ey, Mef2 and Onecut implantations (**f**) and after generation of repressor sites (**g**). **h**, Zoom into the selected 49 bp part of the 500 bp sequence from **e**. The size of the motifs, the spaces between motifs and the flankings are shown at the bottom. **i**, In vivo enhancer activity of the cloned 49 bp sequence with Ey, Mef2 and Onecut implantations. In **f,g,i**, the expected location of γ-KC is shown with dashed circles. Scale bars, 100 μm. Ey (E), Mef2 (M), Onecut (O) and Sr (S).

Onecut and Sr increased the prediction score only marginally (Fig. 4a). Implanting Ey and Mef2 consecutively increased the score more than the sum of their individual contribution and their implantation order did not affect the final score. Adding Onecut and then Sr on top of Ey and Mef2 sites increased the scores even further until it reached the level that we obtained above after 15 mutations through in silico sequence evolution (Fig. 4a). We could also observe some minor preferences in the motif flanking sequence (for example, Mef2 is flanked by T or G in 5′ and A or C in 3′; Extended Data Fig. 4a)

We also found that high-scoring configurations consisted of activator sites that are positioned close together within a distance usually smaller than 100 bp (Fig. 4b,c and Extended Data Fig. 4b). When the

Ey and Mef2 pair were implanted on the same strand, we observed strong preference for a 5 bp distance (or 4 bp when implanted on opposite strands) between the two binding sites whereby Mef2 was located upstream of Ey (Fig. 4b and Extended Data Fig. 4c). For the Ey and Onecut pair, there was a strong preference for a 3 bp space and Onecut preferred the downstream side of Ey (Fig. 4c and Extended Data Fig. 4d).

We investigated the nucleotide contribution scores before and after motif implantations for an example sequence with high prediction score in which motifs were inserted close together (Fig. 4d,e and Supplementary Fig. 11). The initial random sequence contained several repressor binding sites and the Ey binding site implantation destroyed the strongest repressor binding site. Mef2 and Onecut implantations followed the predicted spacing relative to Ey, with a distance of 5 and 3 bp, respectively. This can explain why implantation of motifs at random locations yields lower scoring sequences (Fig. 4a). Even though some repressor binding sites were still present at further distances, their relative negative contribution was decreased after the activator binding site implantations (Fig. 4e). Testing this designed 500 bp sequence in vivo confirmed specific activity in KC (Fig. 4f). Introduction of mutations to generate repressor sites close to the implanted motifs (none of the activator sites was modified) resulted in complete loss of enhancer activity in vivo, suggesting dominance of repressor motifs (Fig. 4d,e,g). Furthermore, a 49 bp subsequence, containing just the three binding sites, resulted in the same activity and specificity in vivo (Fig. 4h,i and Supplementary Fig. 12). We further confirmed the robustness of the motif implanting design by validating in vivo a second 500 bp sequence showing increased spacing between motifs (Extended Data Fig. 4e,f,g). This result suggests that a functional KC enhancer can be created through motif-by-motif implantation with just these three binding sites and its size can be decreased to the minimal length required to contain these binding sites.

As a third strategy for enhancer design, we used generative adversarial networks (GAN) that have been shown to be powerful generators in different fields[43,48], including the generation of functional genomic sequences[46]. This method was less interpretable than in silico evolution or motif implanting but still allowed the generation of functional and specific enhancers (Supplementary Note 2).

## Human enhancer design

We used our previously trained and validated melanoma deep learning model, DeepMEL2 (ref. 12) (Supplementary Note 1) with the same three strategies as before, to design human melanocyte or melanocyte-like melanoma (MEL) enhancers. As for the *Drosophila* experiments, we started from GC-adjusted random sequences (Extended Data Fig. 5a) and, by following the nucleotide-by-nucleotide sequence evolution approach, we evolved them into sequences with high prediction scores for the MEL class. This process drove the generation of activator binding sites (SOX10, MITF and TFAP2) and the destruction of ZEB motifs to resemble MEL genomic enhancers; the prediction scores started to plateau after 15 mutations (Fig. 5a and Extended Data Fig. 5b,c). We randomly selected ten regions that were evolved from scratch (EFS-1–10) with 15 mutations and tested their activity with a luciferase assay in vitro, in a MEL cell line (MM001) (Fig. 5b,c and Methods). Seven of ten tested enhancers showed activity in the range of previously characterized positive control (native) enhancers and none of them showed activity in a cell line that represents another melanoma cell state (mesenchymal-like, MM047) in which the MEL-specific TFs (SOX10, MITF and TFAP2) are not expressed (Fig. 5d and Extended Data Fig. 5d). When we integrated these synthetic enhancers into the genome of the MM001 cell line using lentiviral vectors (Methods), they generated an ATAC-seq peak, whereas neither the random sequences nor the evolved sequence when integrated in a non-MEL cell line are accessible (Fig. 5e and Extended Data Fig. 5e,f).

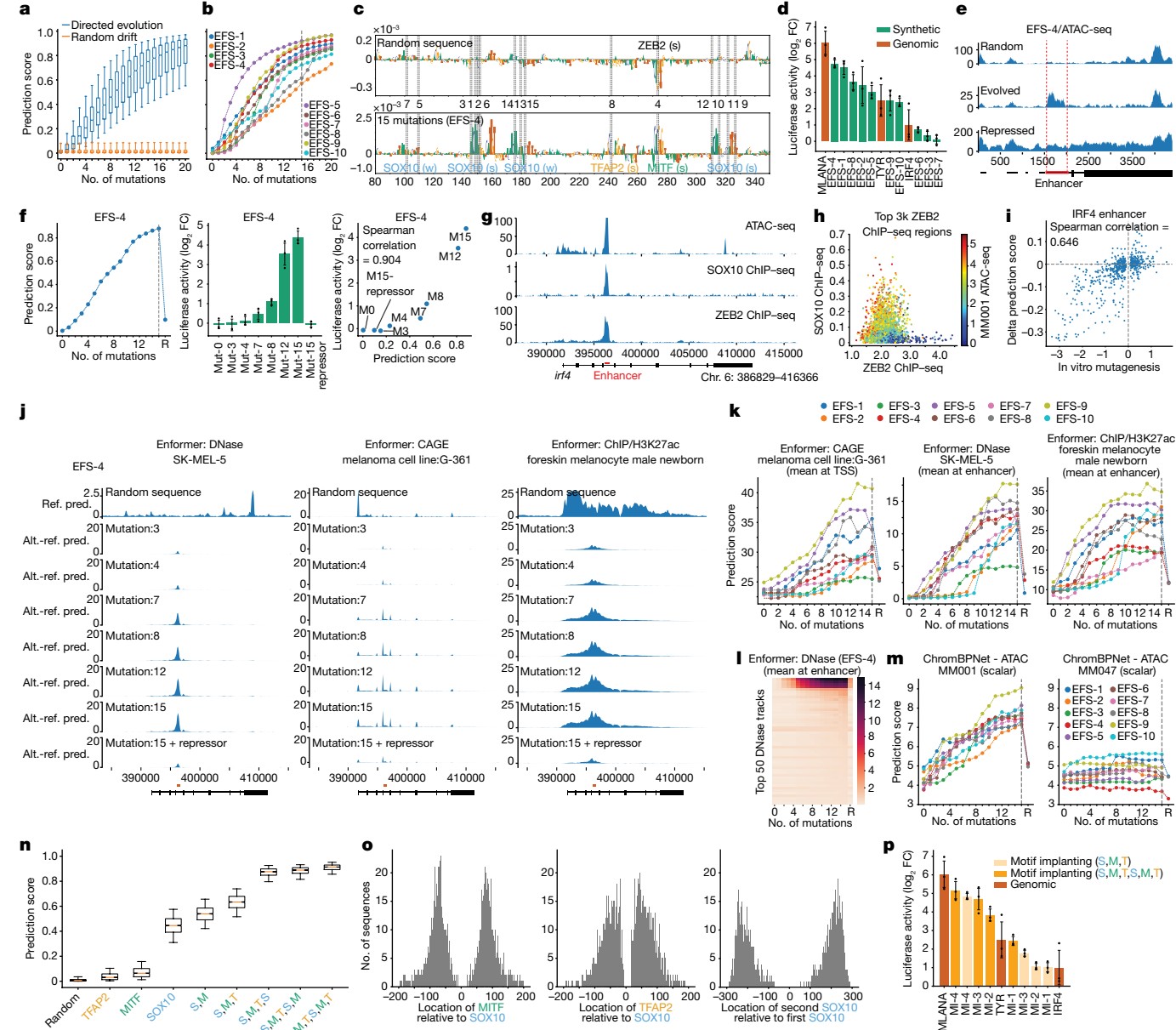

**Fig. 5 | Human enhancer design. a,b,** Prediction score distribution (MEL class, *n* = 4,000 sequences (**a**) and ten selected sequences (**b**)) after each mutation. **c,** Nucleotide contribution scores of a synthetic sequence pre (top) and post (bottom) 15 mutations, with binding site names, mutation positions (dashed lines) and orders (between top and bottom plots). **d,** Mean luciferase signal (log₂ fold-change (FC)) over *Renilla* of synthetic sequences from in silico sequence evolution and genomic enhancers. **e,** MM001 ATAC-seq profile of three integrated EFS reporters: initial, evolved and evolved with repressor sites. Red lines, enhancer boundaries. **f,** DeepMEL2 prediction scores (left), luciferase activity (middle) and their correlation (right) for EFS-4 sequences. **g,** MM001 ATAC-seq, SOX10 and ZEB2 ChIP–seq tracks for *IRF4* gene; enhancer location in red. **h,** ZEB2 ChIP–seq signal (*x* axis), SOX10 ChIP–seq signal (*y* axis) and ATAC-seq signal (colour) for top ZEB2 regions in MM001. **i,** In vitro and in silico saturation mutagenesis values of the *IRF4* enhancer. **j,** Enformer predictions for EFS-4 sequences replacing *IRF4* enhancer: initial score and score changes postmutations. Ref. pred., reference predictions; Alt. - ref. pred., delta alternative versus reference predictions. **k,** Enformer predictions per mutation step and after repressor addition for MEL EFS sequences. **l,** Prediction scores for top 50 DNase tracks for EFS-4 sequences. Four first tracks are foreskin melanocyte male newborn and SK-MEL-5 tracks. **m,** ChromBPNet ATAC MM001 (MEL) and MM047 (MES) prediction scores for EFS sequences, across mutations and postrepressor addition. **n,** Prediction score distribution for MEL class (*n* = 2,000 sequences) after motif implantation. **o,** Relative TF locations distribution (*n* = 2,000). **p,** Luciferase signal (log₂ FC over *Renilla*) comparison of motif-implanted sequences and genomic enhancers. In **a,n**, box plots show the median (centre line), interquartile range (box limits) and 5th and 95th percentile range (whiskers). Error bars in **d,f,p** denote mean standard error (*n* = 3 biological replicates). S, SOX10; M, MITF; T, TFAP2.

Next, we tested the activity of a series of synthetic sequences, along the design path, from a random sequence to an active enhancer (Extended Data Fig. 6 and Supplementary Figs. 13 and 14). This shows that the predicted activity by DeepMEL2 correlates with the luciferase reporter activity in vitro (Fig. 5f and Extended Data Fig. 5g), suggesting that the steps of increased activity are not biased to our

DeepMEL2 model but reflect biological activity. Functional in silico evolved enhancers lost their activity and accessibility, when ZEB sites were generated in proximity of activator sites (Fig. 5e,f and Extended Data Figs. 5g and 8) and this repressive mechanism depended on the number and the strength of repressor sites (Extended Data Fig. 8a,b–e and Supplementary Fig. 15). We confirmed that the same principles of

repression apply to genomic enhancers, using the MEL enhancer in an *IRF4* intron as example and through ChIP–seq we identified ZEB2 as the actual repressor TF (Fig. 5g,h and Supplementary Note 3). Mutating the endogenous ZEB2 site in the *IRF4* enhancer causes a significant increase in activity, whereas mutations that generate more ZEB2 sites (without touching activator sites) decrease its activity (Fig. 5i and Supplementary Note 3).

These findings could be further corroborated by scoring all sequences during the optimization process with two other deep learning models, namely, a newly trained ChromBPNet model[50] on bulk MM001 ATAC-seq data (Methods) and the previously published Enformer model, for which the SK-MEL-5 ATAC-seq class represents the MEL state[2]. The Enformer model has a receptive field of 200 kilobases (kb) and can be used to predict both enhancer activity and target gene expression in the context of an entire gene locus. To simulate whether our synthetic enhancers do function like genomic enhancers in a complex locus, we replaced the *IRF4* enhancer studied above with synthetic enhancers, thus performing an in silico CRISPR experiment. Replacement of the *IRF4* enhancer by a random sequence results in no predicted accessibility, whereas replacement by different synthetic enhancers along their design path gradually obtains increased prediction scores for accessibility, H3K27Ac signal and CAGE gene expression (Fig. 5j,k and Extended Data Fig. 7b). Because Enformer contains more than 600 chromatin accessibility (DNase hypersensitivity) output classes, across a wide variety of cell types, we used it to assess the specificity of our designed enhancers and found high prediction scores for only four classes, each representing either melanocytes or melanocyte-like melanoma cell states (Fig. 5l and Extended Data Fig. 7a). The ChromBPNet model shows continuous increases of predicted enhancer activity along the optimization path (Fig. 5m). Again, all three models correctly predict that synthetic enhancers, after they reach their highest activity level, can be switched off entirely by introducing point mutations that generate ZEB binding sites (Fig. 5j,k,m and Extended Data Fig. 7a,b). Furthermore, changing the location of the enhancer relative to the transcriptional start site did not alter its functionality, suggesting that the enhancers are not dependent on the local sequence context around the *IRF4* enhancer location to be functional (Extended Data Fig. 7c). As a final example of in silico evolution, we identified a human 'near-enhancer' and rescued its activity with only four mutations (Extended Data Fig. 9a–d).

We also applied the motif implantation strategy to design human enhancers. We implanted SOX10, MITF and TFAP2 binding sites to 2,000 random sequences of 500 bp. Whereas implanting only MITF or TFAP2 resulted in a small increase in the prediction score, implanting SOX10 alone had the strongest effect (Fig. 5n). Adding MITF and then TFAP2 on top of SOX10 sites increased the prediction scores to 0.6 on average. The prediction scores continued increasing even further after adding another set of SOX10, MITF and TFAP2 binding sites (Fig. 5n). We did not observe a preferential location for the implantation of MITF or TFAP2 relative to SOX10; however, both binding sites were located within 100 bp of SOX10 (Fig. 5o). The second SOX10 binding site was placed further away at a 200–250 bp distance relative to the first SOX10 (Fig. 5o). We selected four sequences with either single or double SOX10, MITF and TFAP2 implanted sites and tested their activity with luciferase assays. All enhancers showed activity in the range of native enhancers and adding the binding sites twice consistently increased the activity of the enhancers (Fig. 5p and Extended Data Fig. 10a,b,c). Replacing the implanted binding sites with their weaker versions taken from a native enhancer (IRF4) decreased the activity of the enhancers dramatically (Extended Data Fig. 10a,b,c). To confirm that the activity of the enhancers was driven by the implanted binding sites, we cut the sequences from the most upstream binding site to the most downstream binding site. These subsequences (116–164 bp) were also active with a slight change in their activity levels (Extended Data Fig. 10a,b,c). Finally, instead of choosing the best location for MITF

and TFAP2 implantation, we implanted them at the closest location to the SOX10 binding site that would result in a positive change in the prediction score. These minimal enhancers (51–64 bp) were as active as their longer (500 bp) version (Extended Data Fig. 10a,b,c).

Finally, we applied the GAN-based sequence generation approach to the generation of human enhancers and obtained similar performances as with the *Drosophila* GAN-generated enhancers (Supplementary Note 2).

In conclusion, these results show that enhancer design strategies are adaptable to different biological systems and even other species, including human.

## Discussion

Understanding the code of transcriptional regulation and using this knowledge to design synthetic enhancers has been a persistent challenge. We successfully designed synthetic enhancer sequences in human and fly guided by deep learning models. By combining a stepwise enhancer design approach alongside model interpretation techniques, we followed the trajectories of in silico enhancer emergence in *Drosophila* and human, towards local optima. Nucleotide-by-nucleotide evolution revealed that the selected mutations predominantly destroy candidate repressor TF binding sites and create candidate activator sites. Mostly, ten iterative mutations were sufficient to convert a random sequence into a cell-type-specific functional enhancer. Similarly, for native yeast promoter sequences, it was recently shown that only four mutations could dramatically increase or decrease their activities[45]. This evolutionary design process may represent an optimized version of natural evolution of genomic enhancers. We found that the fly and human genomes contain 'near-enhancers' that require few mutations to become functional.

The location, orientation, strength and number of TF motifs in a single enhancer and their distance to other motifs are important features determining an enhancer code that is unique to each cell type. This array of well-arranged TF binding sites constitutes a docking platform for a specific combination of TFs. Their cooperative binding makes the enhancer accessible/active at different levels and in different cell types. We found certain enhancers to be active in several cell types. Besides the trivial possibility whereby two cell types share a common set of TFs that bind to a common set of sites (for example, different KC subtypes), we showed that some enhancers have evolved several intertwined codes (for example, KC and T neurons). We could prove this by either removing a code from a native dual-code enhancer or adding a second code to a native single-code enhancer.

The consequence of this motif-driven enhancer model is that it allows enhancer design by motif implantation. Several studies have used motif implantation in an attempt to reconstitute enhancer activity but successes of accurate in vivo activity have been limited[51,52]. More recently, motif embedding has also been used in combination with deep learning models[4,6,53] with the advantage that many different motif implanting scenarios can be tested in silico, before performing experimental validation[4,6,43,53], as compared to high-throughput testing of random implantations[32,54,55]. By exploiting motif implantation further, particularly by scoring each possible implant position, as well as combinations of motifs, we could reveal motif synergies (for example, Ey + Mef2 or SOX10 + MITF), as well as preferred orientations and distances between motifs, motif strengths and motif copy number. A minimal fly brain enhancer designed with three abutting motif instances illustrates that functional enhancers can be created without further sequence context. Compared to random insertions of motif instances[52,56], deep learning guided implantation has the capacity to take the entire enhancer sequence into account. Consequently, what makes an enhancer is not only the optimal combination of motifs used (including each motif's strength and copy number) but also the optimal

balance between repressor and activator motifs and the optimal motif arrangement.

Two of 13 KC enhancers remain negative, whereas one is inconclusive. Nevertheless, this leads to a conservative success rate greater than 75%. We also envision several routes for further improvement in enhancer design. First, whereas our examples focused on adult cell types, we did not consider temporal changes. It thus remains to be investigated whether developmental enhancers with highly dynamic and complex output functions can be decoded and designed along the same principles. Studies of the *shavenbaby* enhancer in *Drosophila* showed that its output is affected by mutations in most of its nucleotides[57]. This may be due to a densely packed motif content, such as our minimal enhancer or to yet-unknown sequence features. It may be interesting to investigate such developmental enhancers with deep learning models[58]. Also, we observed slight variations in the GFP output pattern of (genomic and synthetic) enhancers. Incorporating such high-resolution variations in the training data may yield models with improved spatial and quantitative resolution. Lastly, the repressor motifs identified by our models recruit TFs that cause a decrease in chromatin accessibility. However, this is probably not true for all transcriptional repressors (for example, binding sites of the REST repressor overlap with accessible chromatin[59]). A future challenge will be to take repressor motifs into account that do not decrease chromatin accessibility. To train such models, more enhancer activity data or gene expression data will be needed.

The successful application of enhancer design on both fly brain and human cancer cells has shown that simple, yet powerful strategies guided by deep learning models are adaptable to different organisms or systems. Our proof-of-concept study is an encouraging step forward towards the development of organism-wide deep learning models. Such models will facilitate the generation of synthetic enhancers during development, disease and homeostasis; and will further improve our understanding and control of the genomic *cis*-regulatory code.

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

# Methods

## Data reporting

No statistical methods were used to predetermine sample size. The number of synthetic enhancers that were tested using transgenic flies was determined to be minimally six per cell type and it was bounded by the feasibility of the transgenic animal generation experiments. In total, 68 transgenic fly lines were generated. The number of synthetic enhancers that were used with luciferase assays was determined to be minimally ten per different category (in silico evolution, motif embedding, GAN, repressors and mutational steps). In total, 97 sequences were tested using luciferase assay. The initial random sequences (used for sequence evolution and motif implantation) were sampled from the sequence space that matches the GC content of the genomic sequences. Flies fitting the sex (equal amount of male flies and female flies) and age (less than 10 days) criteria were selected randomly for all experiments. In this study, we didn't perform experiments that needed to be allocated into different groups. The investigators were blinded when performing cloning, transfection, antibody staining and luciferase experiments by using enhancer IDs.

## Statistics and reproducibility

Statistics were calculated using Scipy (v.1.6.0; RRID: SCR_008058)[60]. The results here and throughout the manuscript were visualized using matplotlib (v.3.1.1; RRID: SCR_008624)[61]. The deep learning models were run in a conda environment in which python (v.3.7; RRID: SCR_008394), tensorflow-gpu (v.1.15; RRID: SCR_016345)[62], numpy (v.1.19.5; RRID: SCR_008633)[63], ipykernel (v.5.1.2; RRID: SCR_024813) and h5py (v.2.10.0; RRID: SCR_024812) packages were installed. The same results were obtained from different replication experiments. Several brains (at least ten) were stained and imaged for the fly experiments. Three biological replicates were performed for the main luciferase experiments. Two biological replicates were performed for the negative control luciferase experiments. No biological replicates were performed for ATAC-seq or ChIP–seq experiments.

## In silico saturation mutagenesis

To measure the effect of each possible single mutation on a given DNA sequence, we performed in silico saturation mutagenesis, as described earlier[13,48,64]. We first generated the sequences of all single mutations for a given 500 bp sequence (three possible mutations for each nucleotide, making 1,500 sequences in total). We scored these sequences and the initial sequence with the deep learning models. For a chosen class, we calculated the delta prediction score by subtracting the score of the initial sequence from the score of the mutated sequence for each mutation.

## Random sequence generation

We generated random 500 bp sequences to use as a previous set for the in silico sequence evolution and motif implantation by using the numpy.random.choice(["A","C","G","T"]) command. For each position, instead of using 25% probability for each nucleotide to be chosen, we used the frequency of the nucleotides from fly or human genomic regions for each position. In these genomic regions, the GC content was higher in the centre of the regions on average relative to the flankings. We used 6,126 KC regions for fly and 3,885 MEL regions for human that we identified in our previous publications[3,5].

## In silico sequence evolution

By using the saturation mutagenesis scores mentioned above, we performed in silico sequence evolution. For the in silico evolution from random sequences, we calculated saturation mutagenesis scores for a random sequence. Then, we selected the mutation that had the highest positive delta prediction score for the selected class (for γ-KC, class no. 35 in DeepFlyBrain; for PNG, class no. 34 in DeepFlyBrain; for MEL, class no. 16 in DeepMEL2). For the selected sequence with one mutation, we recalculated the saturation mutagenesis scores for each nucleotide and again selected the mutation with the highest delta score and repeated this procedure until the initial random sequence accumulated 20 mutations.

Even though we used a simple objective function to direct the sequence evolution towards a single cell type, without explicitly penalizing off-target cell types, the generated sequences were mostly active only in the targeted cell type. We believe this is because of the type of enhancer models we are using, which were trained on cell-type-specific accessible regions. When more general models are used, for example trained on entire ATAC-seq tracks, adapted objective functions can be used and are available in our code. The cell-type-specific activity of our synthetic enhancers suggests that: (1) activator binding sites were not created for other cell types; and (2) repressor sites, which are present in random sequences by chance, were not destroyed for other cell types. For example, in KC we observed that activator binding sites are usually longer than repressor sites (18 and 10 bp versus 5 and 6 bp for Ey, Mef2, Mamo and CAATTA, respectively). This implies that a random sequence is more likely to have several repressor binding sites by chance compared to activator sites (Extended Data Fig. 1f). Indeed, the average prediction scores of our initial 6,000 random sequences were close to zero for all classes. This may at least in part explain why earlier enhancer design efforts may have failed.

We used 6,000 initial random sequences for KC and PNG and 4,000 for MEL. For the generation of KC enhancers from genomic regions, we performed six iterative mutations. For the many cell-type code enhancers, we started from optic lobe enhancers and in each iteration we manually selected the mutations that increased the γ-KC prediction score while maintaining the optic lobe prediction scores high. For the pruning experiment of a multiple cell-type code enhancer into only KC code, we manually selected the mutations that maintain the γ-KC prediction score high while decreasing the optic lobe prediction scores. The DeepFlyBrain class numbers used for optic lobe neurons are 23 for T1, 20 for T2 and 2 for T4 neurons.

To rescue the designed enhancers that were weak or negative, we performed five more mutations on both from-scratch and from-genomic sequences.

To repress the sequences with the creation of repressor binding sites, we selected single or double mutations manually, by going over in silico saturation mutagenesis plots calculated on the evolved sequences.

To explore the alternative in silico sequence evolution paths besides choosing the best mutation (greedy algorithm), we chose the top 20 mutations on each sequence for every incremental step starting from a random sequence. We followed this procedure for five incremental mutational steps. Starting from the random sequence used to generate enhancer KC EFS-4, we obtained 3.2 million paths/sequences at the end.

## Nucleotide contribution scores

We used a network explaining tool, called DeepExplainer (SHAP package[65,66]; RRID: SCR_021362), to calculate the contribution of each nucleotide to the final prediction of the deep learning model for the chosen class. We used randomly selected 250 genomic regions to initialize the explainer.

DeepFlyBrain model takes a single strand as an input. For a given 500 bp, we multiplied the explainer's output by the one-hot encoded DNA sequence and visualized it as the height of the nucleotide letters. DeepMEL2 model takes forward and reverse strands separately as an input. In this case, the explainer results in contribution scores for each strand. We first took the average contribution score for each nucleotide and then multiplied it by the one-hot encoded DNA sequence to visualize.

## Motif annotation

To identify TF binding sites during the in silico evolution of designed sequences, we used TF-Modisco (v.0.5.5.4; RRID: SCR_024811)[67] and

Cluster-Buster(RRID: SCR_024810)[68]. First, we calculated the nucleotide contribution scores on every mutational step, including random sequences. Then, we ran TF-Modisco on each mutational step separately to identify which patterns are appearing/disappearing. The TF-Modisco parameters we used were num_to_samp=5000, sliding_window_size=15, flank_size=5, target_seqlet_fdr=0.15, trim_to_window_size=15, initial_flank_to_add=5, final_flank_to_add=5, final_min_cluster_size=60. After investigating the TF-Modisco patterns that were identified on each mutational step, we used mutational step 1 for KC and mutational step 4 for MEL to collect the identified patterns, as they contained all the activator and repressor patterns. (Earlier steps did not have good representation of activators because they are close to random sequences. Later steps did not have good representation of repressors because they were destroyed during the mutational steps.) We trimmed the patterns on the basis of information content (threshold = 0.1) and saved them as a .cb file to be used by the Cluster-Buster.

By using the TF-Modisco patterns, we ran Cluster-Buster (with -c 0 and -m 3 options) to identify motifs on each mutational step, including random sequences. We selected only the motif instances from Cluster-Buster results and merged (by using BEDTools v.2.30.0; RRID: SCR_006646; ref. 69) the overlapping hits of the motifs into a single hit. We calculated mean + s.d. on the hit scores coming from random sequences for each motif separately and used these thresholds to get the significant hits.

Identification of TF binding sites similar to TF-Modisco patterns was performed using Tomtom (RRID: SCR_024809)[70] using the cisTarget motif collection (RRID: SCR_024808)[71].

### Scoring the fly genome

To identify the regions that have high prediction scores for γ-KC but have less accessibility in γ-KC, we scored the whole fly genome. We used the bedtools makewindows -g dm6.chromsize -w 500 -s 50 command[69] to create the coordinates of the binned fly genome with a 500 bp window and 50 bp stride. We removed the regions that are not exactly 500 bp. This resulted in 2,750,893 regions to be scored with the DeepFlyBrain model. We used the stats function of deeptools/pyBigWig package (RRID: SCR_024807)[72] to calculate mean γ-KC accessibility values for each bin.

### Motif implanting

To implant binding sites into 500 bp sequences, we started from a random sequence. We implanted a binding site into every possible location on the random sequence one-by-one by replacing the nucleotides on the random sequences with the binding site. Then, we scored these sequences with the model. We selected the binding site position that gives the highest prediction score and implanted the motif on that position. Then, starting from this sequence with one binding site implanted, we implanted the next binding sites one-by-one by using the same procedure. The sequence of binding sites that maximize the TF-Modisco pattern score were selected to implant and they are as follows: Ey, TGCTCACTCAAGCGTAA; Mef2, CTATTTATAG; Onecut, ATCGAT; Sr, CCACCC; SOX10, AACAATGGGCCCATTGTT; MITF, GTCACGTGAC; and TFAP2, GCCTGAGGC. We used 2,000 initial random sequences for KC and 2,000 for MEL. The weaker binding sites taken from the *IRF4* enhancer are as follows: SOX10_1, GTGAATGACAGCTTTGTT; SOX10_2, TACAAGTATCTCCATTGT; MITF_1, ATCATGTGAA; MITF_2, GCCATAT GAC; TFAP2_1, TCTTCAGGC; and TFAP2_2, CCCTGTGGT.

When TF motifs are implanted at random positions in a random sequence, prediction scores are very low, probably because repressor sites remain present. Likewise, to be able to generate a functional enhancer through random sequence generation, many sequences need to be generated (that is, 100 million and 1 billion; refs. 38,73).

To measure if there is a preference for a flanking sequence when performing motif implanting, we aggregated all the sequences aligned by the location of the implanted motif. Then, we calculated the position probability matrix and visualized it by subtracting 0.25 from each position.

To measure the effect of different background sequences on the minimal KC enhancer, we generated 1 million random sequences with the size of 20 bp. Then, we replaced the 20 bp spanning the position where Ey, Mef2 and Onecut binding sites implanted that occupied the 6 bp flankings on both sides and 8 bp intermotif space. Then, we scored the sequences with the model and measured the effect of different backgrounds around the motif implantation area.

### Generative adversarial network

To train a GAN model, we used Wasserstein GAN architecture with gradient penalty[74] similar to earlier work[47]. The model consists of two parts: generator and discriminator. Generator takes noise as input (size is 128), followed by a dense layer with 64,000 (500 × 128) units with ELU activation, a reshape layer (500, 128), a convolution tower of five convolution blocks with skip connections, a one-dimensional (1D) convolution layer with four filters with kernel width 1 and finally a SOFTMAX activation layer. The output of the generator is a 500 × 4 matrix, which represents one-hot encoded DNA sequence. Discriminator takes 500 bp one-hot encoded DNA sequence as input (real or fake), followed by a 1D convolution layer with 128 filters with kernel width 1, a convolution tower of five convolution blocks with skip connections, a flatten layer and finally a dense layer with one unit.

Each block in the convolution tower consists of a RELU activation layer followed by 1D convolution with 128 filters with kernel width 5. The noise is generated by the numpy.random.normal(0, 1, (batch_size, 128)) command. We used a batch size of 128. For every train_on_batch iteration of the generator, we performed ten train_on_batch iterations for the discriminator. We used Adam optimizer with learning_rate of 0.0001, beta_1 of 0.5 and beta_2 of 0.9. We trained the models for around 260,000 batch training iteration for KC and around 160,000 batch training iteration for MEL.

We used 6,126 KC regions for the fly model and 3,885 MEL regions for the human model, which we identified in our previous publications, as real genomic sequences to train the models. After the training, we sampled 6,144 (48 × batch size) sequences for KC and 3,968 (31 × batch size) sequences for MEL by using the generator for every 10,000 batch training iteration. The sampled synthetic sequences were generated by calculating predictions on noise and then the numpy.argmax() command was used to convert the predictions into one-hot encoded representations.

### Background model

To compare against the GAN-generated sequences, we generated random sequences in different orders by using the CreateBackgroundModel function from the INCLUSive package (RRID: SCR_013488)[75] based on the same genomic regions that we used to train GANs.

### Training ChromBPNet models

For training ChromBPNet models we used a prereleased version (v. 1.3-pre-release; RRID: SCR_024806) from the ChromBPNet GitHub repository (https://github.com/kundajelab/chrombpnet/tree/v1.3-pre-release). We followed all the preprocessing and training steps as described in the tutorial: from the aligned ATAC reads in the MM001 BAM file, we made a BigWig of Tn5 insertion sites, trained a bias model that predict Tn5 binding sites in non-peak regions which is then used in the ChromBPNet model to filter out Tn5 bias. ChromBPNet uses 2,114 bp DNA sequence as input and predicts both the ATAC track and the natural log count of the aligned reads for the central 1,000 bp. To be able to score 500 bp DNA sequences (*IRF4* enhancer and synthetic enhancers), we used the flanking sequences of the cloned/integrated enhancer sequences surrounded by the integrated cassette. Both scalar and track prediction were plotted. Flanking sequences are provided in the Supplementary Code.

## Using the Enformer model

We used the Enformer model (RRID: SCR_024805) to do in silico CRISPR experiments. We took the *IRF4* locus (chr. 6: 339010:453698) centred by the *IRF4* enhancer (chr. 6: 396104:396604). We replaced the endogenous *IRF4* enhancer with the random/ evolved/ repressed designed sequences and calculated the prediction scores for the related cell types. The prediction scores were plotted as showing the whole locus. For DNase and ChIP-Histone:H3K27ac tracks, the mean values were calculated using the middle three bins or one bin spanning the enhancer location. For CAGE tracks, the mean values are calculated using one bin spanning the transcriptional start site of *IRF4*. The index of the tracks that we used to get the prediction scores are as follows−4,832: CAGE/melanoma cell line:G-361, 162: DNase/SK-MEL-5, 2,162: ChIP-Histone:H3K27ac/foreskin melanocyte male newborn.

To measure the locational effect of the designed enhancers on gene expression, chromatin accessibility and histone modification, we moved the synthetic enhancer around the *IRF4* locus; (1) to 10 kb upstream, (2) 5 kb upstream (which is next to the promoter of the *IRF4* gene) and (3) 17.5 kb downstream of the original location.

## Cloning of synthetic *Drosophila* enhancers

Synthetic sequences were ordered from Twist Bioscience, precloned in the pTwist ENTR vector. The motif-implantation and double-coded sequences were synthesized with an extra 5′ CACC sequence as double-stranded DNA (gBlocks Gene Fragments) by IDT. The 49 bp motif-implantation sequence was ordered from IDT as forward and reverse single-stranded DNA oligos, which were then annealed for 5 min at 95 °C and cooling down to RT over 1 h. The double-stranded DNA sequences were then cloned into the pENTR/D-TOPO plasmid (Invitrogen).

All sequences were introduced in a modified pH-Stinger vector[76], containing nuclear GFP, Hsp70 promoter, gypsy insulators and attB site for phiC31 integration, through Gateway LR recombination reaction (Invitrogen). A total of 2 µl of the reaction was transformed into 25 µl of Stellar chemically competent bacteria (Takara). Plasmid minipreps were performed using the NucleoSpin Plasmid Transfection-grade Mini kit (Macherey-Nagel) and sequenced with Sanger sequencing to confirm the correct insertion of the regions in the destination plasmid. After confirmation of the sequence, plasmid midipreps were performed using the NucleoBond Xtra endotoxin-free Midi kit (Macherey-Nagel). Next, the plasmids were sent to FlyORF (Switzerland) for injection in *Drosophila* embryos (21F site on chromosome 2l) and positive transformants were selected on the basis of eye colour.

*Drosophila* flies were raised on a yeast-based medium at 25 °C under a 12 h/12 h day/night light cycle.

## Immunohistochemistry analysis of *Drosophila* brains

Brains of adult flies (*Drosophila melanogaster*, less than 10 days old, equally mixed sex) were dissected in PBS and transferred to a tube for fixation in 4% formaldehyde in PBS for 20 min. All incubations were done at room temperature, unless otherwise indicated. Brains were washed in PBS with 0.3% Triton-X (PBST) three times for 10 min each, then they were placed in blocking solution (5% normal goat serum (Abcam) in PBST) for 3 h. We incubated the brains overnight at 4 °C in primary antibodies diluted in blocking solution (rabbit anti-GFP, IgG (Invitrogen), 1:1,000 and mouse anti-Dachshund, mAB dac1-1 (DSHB), 1:250). The brains were then washed in PBST three times for 10 min each and incubated with the fluorochrome-conjugated secondary antibodies diluted in blocking solution for 2 h (Alexa Fluor 488 donkey anti-rabbit IgG (Invitrogen), 1:500 and Alexa Fluor 647 goat anti-mouse IgG (Invitrogen), 1:500). Next, brains were washed in PBS three times for 10 min each. Finally, samples were mounted onto microscope slides with Prolong Glass Antifade Mountant (Invitrogen).

For image acquisition, a Zeiss LSM900 microscope equipped with Airyscan2 in combination with a ×20 objective (Plan Apo 0,80 Air) was used. The setup was controlled by ZEN blue (v.3.4.91, Carl Zeiss Microscopy GmbH). GFP was excited with a blue diode 100 mW at 488 nm and tiled images were collected with emission filter BP450-490/BS495/BP500-550.

## Cloning of synthetic human enhancers

The 500 bp synthetic sequences were ordered from Twist Bioscience, precloned in the pTwist ENTR vector. The 500 bp regions were introduced in the pGL4.23-GW luciferase reporter vector (Promega) through Gateway LR recombination reaction (Invitrogen) and 2 µl of the reaction was transformed into 25 µl of Stellar chemically competent bacteria (Takara).

Synthetic sequences shorter than 150 bp were ordered as gBlocks from IDT (Integrated DNA Technologies) with 5′ (cccgtcgacgaattctgca gatatcacaagtttgtacaaaaaagcaggct) and 3′ (acccagctttcttgtacaaagtg gtgataaacccgctgatcag) adaptors. The pGL4.23-GW luciferase reporter vector was linearized through inverse PCR with primers Lin_pSA335_short_ME_For (gtggtgataaacccgctgatcag) and Lin_pSA335_short_ME_Rev (tctgcagaattcgtcgacggg). The short sequences and the linearized vector were combined in an NEBuilder reaction (New England Biolabs) and 2 µl of the reaction was transformed into 25 µl of Stellar chemically competent bacteria.

For all cloning procedures, plasmid minipreps were performed using the NucleoSpin Plasmid Transfection-grade Mini kit (Macherey-Nagel) and sequenced with Sanger sequencing to confirm the correct insertion of the regions in the destination plasmid.

To generate stable cell lines with synthetic enhancers, the synthetic sequences were cloned into the pSA351_SCP1_intron_eGFP vector (Addgene no. 206906). The vector was linearized through inverse PCR with primers Lin_pSA351_For (ctgagctccctagggtact) and Lin_pSA351_Rev (cgactcgaggctagtctc). The synthetic sequences were PCR-amplified from their respective pGL.23-GW vector with their respective primer pairs: MM_EFS_1_For (gagactagcctcgagtcgctgatt gtttgaaccattgttacgatttgg) and MM_EFS_1_Rev (agtaccctagggagctcag caattttgttttttgcgcgtgac) for MM-EFS-1 sequences; MM_EFS_4_For (gagactagcctcgagtcgtgatatgtattcacccatgccctca) and MM_EFS_4_Rev (agtaccctagggagctcaagggtttgtatatgtatgctcctttatacga) for MM-EFS-4 sequences; MM_EFS_8_For (gagactagcctcgagtcgatacgcacgacaaagcct cat) and MM_EFS_8_Rev (agtaccctagggagctcacactgtacaaggcatcccgc) for MM-EFS-8 sequences; IRF_4_For (gagactagcctcgagtcggctgccattggtgtg gattttaag) and IRF_4_Rev (agtaccctagggagctcaactggcatcgagacggg) for IRF4 sequences. The PCR amplicons and the linearized vector were combined in an NEBuilder reaction and 2 µl of the reaction was transformed into 25 µl of Stellar chemically competent bacteria. Plasmid minipreps were performed using the NucleoSpin Plasmid Transfection-grade Mini kit (Macherey-Nagel) and sequenced with Sanger sequencing to confirm the correct insertion of the regions in the vector. After confirmation of the sequence, plasmid maxipreps were performed using the NucleoBond Xtra endotoxin-free Maxi kit (Macherey-Nagel).

## Transfection and luciferase assay

MM001 and MM047 were seeded in 24-well plates and transfected with 400 ng of pGL4.23-enhancer vector + 40 ng of pRL-TK *Renilla* vector (Promega) with Lipofectamine 2000 (Thermo Fisher Scientific). As positive controls, the previously published enhancers MLANA_5-I, IRF4_4-I and TYR_−9-D or ABCC3_11-I and GPR39_23-I were used for MM001 and MM047, respectively[77]. One day after transfection, luciferase activity was measured through the Dual-Luciferase Reporter Assay System (Promega) by following the manufacturer's protocol. Briefly, cells were lysed with 100 µl of passive lysis buffer for 15 min at 500 rpm. A total 20 µl of the lysate was transferred in duplicate in a well of an OptiPlate-96 HB (PerkinElmer) and 100 µl of luciferase assay reagent II was added in each well. Luciferase-generated luminescence

was measured on a Victor X luminometer (PerkinElmer). A total of 100 μl of the Stop & Glo Reagent was added to each well and the luminescence was measured again to record *Renilla* activity. Luciferase activity was estimated by calculating the ratio luciferase/*Renilla*; this value was normalized by the ratio calculated on blank wells containing only reagents. Three biological replicates were done per condition for MM001 and two biological replicates for MM047.

### Production of lentivirus
The lentivirus plasmids were transfected in HEK 293 T cells by use of the Lipofectamine 3000 reagent (Thermo Fisher Scientific). A total of 30 μg of pooled plasmid DNA was combined with 20 μg of a Pax2 plasmid (Addgene no. 12260; RRID: Addgene_12260) and 10 μg of the MD2.G plasmid (Addgene no. 12259; RRID: Addgene_12259). At 48 h posttransfection, medium was collected and refreshed. At 72 h posttransfection, medium was collected a second time. Both medium collections were combined and spun down for 5 min at 1,500 rpm. Supernatants was carefully collected with a blunt needle and a syringe and filtered through a 45 μm syringe disc filter (Millex-HV Millipore) into an Ultra-15 MWCO100 centrifugal filter (Amicon). The concentrator tube containing the supernatants was spun down at 4,000 rpm for approximately 45 min until the desired volume of 250 μl was reached. The virus suspension was aliquoted and stored at −80 °C.

### Transduction of melanoma cells
The MM001 cells were seeded into a six-well plate at a density of 250,000 cells per well. Transduction was performed by adding 5–40 μl of lentivirus and Polybrene at 8 μg ml$^{-1}$. Cells were incubated for 24 h before washing away the Polybrene with PBS and with growth medium. After 3 days the cells were split and expanded further.

### OmniATAC-seq
Omni-assay for transposase-accessible chromatin using sequencing (OmniATAC-seq) was performed as described previously[78]. Briefly, 50,000 MM001 cells transduced with the enhancer pools were resuspended in 50 μl of cold ATAC-seq resuspension buffer (RSB; 10 mM TrisHCl pH 7.4, 10 mM NaCl and 3 mM MgCl$_2$ in water) containing 0.1% NP40, 0.1% Tween-20 and 0.01% digitonin by pipetting up and down three times. This cell lysis reaction was incubated on ice for 3 min. After lysis, 1 ml of ATAC-seq RSB containing 0.1% Tween-20 was added and the tubes were inverted to mix. Nuclei were then centrifuged for 10 min at 500$g$ in a prechilled (4 °C) fixed-angle centrifuge. Supernatant was removed and nuclei were resuspended in 50 μl of transposition mix (25 μl of 2× TD buffer, 2.5 μl of transposase (Nextera Tn5 transposase, Illumina), 16.5 μl of PBS, 0.5 μl of 1% digitonin, 0.5 μl of 10% Tween-20 and 5 μl of water) by pipetting up and down six times. Transposition reactions were incubated at 37 °C for 30 min in a thermoblock. Reactions were cleaned-up by MinElute (Qiagen). Transposed DNA was amplified (ten cycles) with primers i5_Indexing_For (aatgatacggcgaccaccgagatctacacnnnnnnnnntcgtcg gcagcgtcagatgtg) and i7_Indexing_Rev (caagcagaagacggcatacgagat nnnnnnngtctcgtgggctcggagatgt). All libraries were sequenced on a NextSeq2000 instrument (Illumina).

Reads were demultiplexed using bcl2fastq (v.2.20; RRID: SCR_015058; https://emea.support.illumina.com/sequencing/sequencing_software/bcl2fastq-conversion-software.html). Adaptors were trimmed by trimgalore (v.0.6.7; RRID: SCR_011847; https://github.com/Felix-Krueger/TrimGalore). Reads were mapped to a custom hg38 genome, which contains integrated sequences as extra chromosomes, using bwa-mem2 (v.2.2.1; RRID: SCR_022192)[79]. By using SAMtools (v.1.16.1; RRID: SCR_002105)[80], reads were sorted and deduplicated and reads from the blacklisted regions (https://www.encodeproject.org/files/ENCFF356LFX/) were cleaned. Bigwig files with RPGC normalization were generated by using deepTools (v.3.5.0; RRID: SCR_016366) bamCoverage[72].

### ChIP–seq
ChIP–seq was performed by following the Myers Lab ChIP–seq Protocol v.011014 on 2 × 10$^7$ MM001 cells. A total of 5 μg of rabbit anti-ZEB2 (1 mg ml$^{-1}$; Bethyl A302-473A; RRID: AB_3076293) was used for ChIP. A total of 15 ng of immunoprecipitated DNA was used to perform library preparation according to the Illumina TruSeq DNA Sample preparation guide. Briefly, the immunoprecipitated DNA was end-repaired, A-tailed and ligated to diluted sequencing adaptors (1/100). After PCR amplification with i5_Indexing_For and i7_Indexing_rev (18 cycles) and bead purification (Agencourt AmpureXP, Analis), the libraries with fragment size of 300–500 bp were sequenced using the NextSeq2000 instrument (Illumina).

Reads were demultiplexed using bcl2fastq (v.2.20; RRID: SCR_015058). Adaptors were trimmed by trimgalore (v.0.6.7; RRID: SCR_011847). Reads were mapped to hg38 using bwa-mem2 (v.2.2.1; RRID: SCR_022192)[79]. By using SAMtools (v.1.16.1; RRID: SCR_002105)[80], reads were sorted and deduplicated and reads from the blacklisted regions (https://www.encodeproject.org/files/ENCFF356LFX/) were cleaned. Bigwig files with RPGC normalization were generated by using deepTools (v.3.5.0; RRID: SCR_016366) bamCoverage[72]. Peaks were called using MACS2 (v.2.1.2.1; RRID: SCR_013291) callpeak[81].

### Cell lines
MM001, MM047 and MM099 were obtained from G. Ghanem and were cultured in Ham's F-10 Nutrient Mix (Invitrogen) + 10% FBS (Invitrogen). We authenticated the cell lines by checking their genomic, transcriptomic and epigenomic profiles[12,82,83]. HEK293T used for lentivirus production was obtained from ATCC (catalogue no. CRL-3216; RRID: CVCL_0063) and were cultured in DMEM (Invitrogen) + 10% FBS (Invitrogen). Cell lines were tested for mycoplasma contamination before experiments and were found to be negative.

### Reporting summary
Further information on research design is available in the Nature Portfolio Reporting Summary linked to this article.

## Data availability
Cloned *Drosophila* and human sequences were provided as Supplementary Tables. DeepMEL, DeepMEL2 and DeepFlyBrain deep learning model files were obtained from Kipoi[84] (http://kipoi.org/models/DeepMEL; https://kipoi.org/models/DeepFlyBrain) with Zenodo record ids 3592129, 4590308 and 5153337. The fasta files used to train GAN models and the trained GAN models are available on Zenodo at https://doi.org/10.5281/zenodo.6701504. Custom genomes (hg38 and dm6) generated in this study are available on Zenodo at https://doi.org/10.5281/zenodo.10184648. Chromatin accessibility values in KC in adult *Drosophila* brains were obtained from GSE163697 (ref. 39). In vitro saturation mutagenesis on *IRF4* data were obtained from https://kircherlab.bihealth.org/satMutMPRA/[85]. Chromatin accessibility of *Drosophila* and transduced melanoma lines and ZEB2 ChIP–seq data generated for this study have been submitted to the NCBI Gene Expression Omnibus (GEO; https://www.ncbi.nlm.nih.gov/geo/) under accession number GSE240003.

## Code availability
Code used to load deep learning models, create random sequences, perform sequence evolution, perform motif implantation and train GAN models together with the IPython Notebooks that reproduces all the figures were provided as Supplementary Code. The data to run the scripts, the models and the intermediate files can be found together with the code at https://doi.org/10.5281/zenodo.10184648.

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

**Acknowledgements** We acknowledge A. Kundaje and A. Pampari for assistance and early access to their ChromBPNet code; the Genomics Core Leuven for assistance in sequencing; and the VIB Bio Imaging Core for their support and assistance in imaging. Computing was performed at the Vlaams Supercomputer Center. This work is funded by the following grants to S.A.: ERC Consolidator Grant (724226_cis-CONTROL), ERC Proof of Concept (963884), ERC Advanced Grant (101054387_Genome2Cells), Special Research Fund (BOF) KU Leuven (grant C14/18/092; C14/22/125), Foundation Against Cancer (F/2020/1396) and FWO (grants G094121N; G0B5619N; G0I2722N - EOS ID: 40007513); Michael J. Fox Foundation for Parkinson's Research (Michael J. Fox Foundation) (ASAP-000430).

**Author contributions** I.I.T. and S.A. conceived the study. I.I.T. performed all computational analyses and designed synthetic enhancers. V.C. performed enhancer cloning with assistance from K.I.S. and D.M. V.C. performed luciferase assays with assistance from D.M. K.I.S. performed antibody staining and visualization with assistance from I.I.T., H.D. and J.I. R.V. performed lentivirus production and cell line transduction. V.C. performed ATAC-seq and ChIP–seq experiments with assistance from H.D., K.T. and A.P. G.H. performed ATAC-seq and ChIP–seq data preprocessing. N.K. trained ChromBPNet models with assistance from E.C.E. I.I.T. and S.A. wrote the manuscript with assistance from D.M.

**Competing interests** The authors declare no competing interests.

**Additional information**
**Correspondence and requests for materials** should be addressed to Stein Aerts.

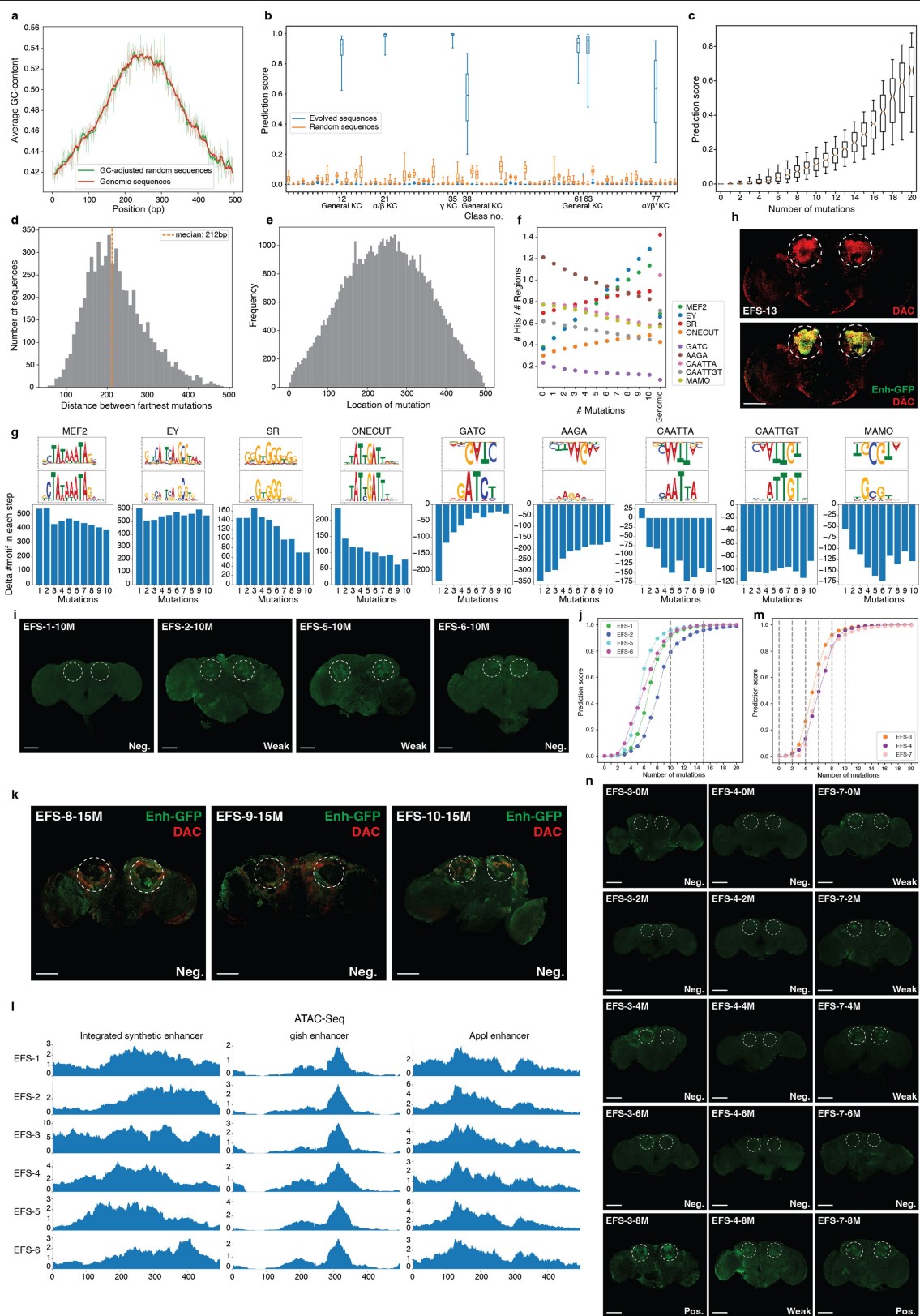

**Extended Data Fig. 1** | See next page for caption.

**Extended Data Fig. 1 | In silico sequence evolution from random sequences.**
**a**, Distribution of GC content in GC-adjusted random sequences (green) and fly genomic regions (red). **b**, Prediction score distribution of the sequences (n = 6,000 sequences) for all classes after 10 mutations. The KC-specific classes and their class number are indicated. In **b**,**c**, the box plots show the median (centre line), interquartile range (box limits) and 5th and 95th percentile range (whiskers). **c**, Prediction score distribution of the sequences that do not reach 0.5 prediction score threshold after 15 mutations for the γ-KC class (n = 180 sequences) after each mutation. **d**, Distribution of distances (n = 6,000) between farthest mutations on each sequence after 10 iterative mutations. The orange line shows the median. **e**, Location of the generated mutations across the random sequences (n = 6,000 sequences). **f**, Average number of motif hits at each mutational step compared to genomic enhancers. **g**, Delta number of motifs in each mutational step. The TF-Modisco patterns and the most similar position weight matrices from the cisTarget motif database are shown at the top of each plot. The patterns that are upside-down are the ones contributing negatively to the model's prediction and they are destroyed by the model on each step. **h**, Top: Dachshund staining (red) highlights KC location in the fly brain. Bottom: colocation of the Dachshund (red) and GFP (green) staining from enhancer EFS-13. **i**, In vivo enhancer activity of the cloned sequences with no or weak enhancer activity. **j**, Prediction scores, at each mutational step, of 4 sequences with no enhancer activity after 10 mutations. The selected iterations (10th and 15th mutations) are indicated with a dashed line. **k**, Dachshund (red) and GFP (green) staining for three negative enhancers. **l**, *Drosophila* adult brain bulk-ATAC-seq profile of 6 transgenic flies that have the designed enhancers integrated. The chromatin accessibility profile of the integrated enhancers (left) and two control regions gish enhancer (middle) and Appl enhancer (right) are shown. **m**, Prediction scores, at each mutational step, of 3 EFS sequences. The selected iterations to study intermediate mutational steps (0, 2, 4, 6, 8, 10 mutations) are indicated with a dashed line. **n**, In vivo enhancer activity of fly lines with subsequent mutational steps. After 8 mutations of a random sequence, the enhancer becomes active in all three lines (EFS-3, 4 and 7) marked by GFP expression. In panels **h**,**i**,**k**,**n**, the expected location of γ-KC is shown with dashed circles. Scale bars, 100 μm.

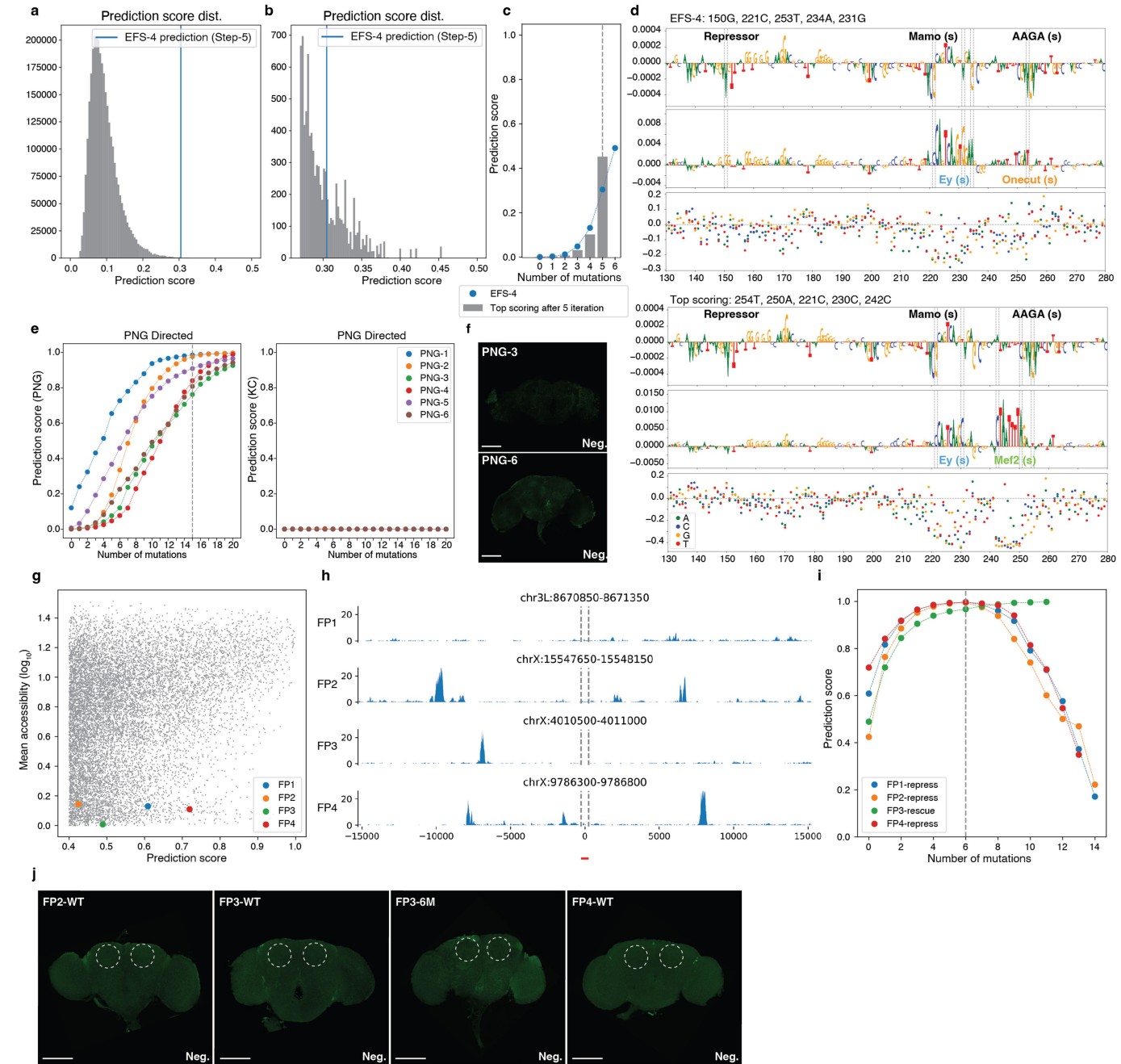

**Extended Data Fig. 2 | State space optimization, design of perineurial glia enhancers and modification of genomic sequences toward KC enhancers.**
**a**, Prediction score distribution for 3 million sequences generated by selecting the top 20 best mutations for 5 incremental mutational steps. Blue line, score of EFS-4 from the greedy algorithm. **b**, Zoomed-in version of panel **a** to the sequences that have higher prediction score than 0.25. **c**, Prediction score of evolved sequences by greedy algorithm (EFS-4) vs the best of 3 million sequences on each mutational step. **d**, Nucleotide contribution score of the original and evolved sequences as well as delta prediction score of in silico saturation mutagenesis for EFS-4 (top) and the top scoring sequence (bottom) **e**, Prediction scores of 6 selected PNG sequences at each mutational step for PNG model (left) and KC model (right). The selected iteration (15th mutation) is indicated with a dashed line. **f**, In vivo enhancer activity of the cloned PNG

sequences with no enhancer activity. **g**, Comparison between γ-KC prediction score and mean γ-KC accessibility for the binned fly genome regions. The selected regions with high prediction and low accessibility are highlighted with blue, orange, green and red dots. **h**, γ-KC ATAC-seq profile of the four selected regions. The exact location of the regions is indicated with dashed lines. **i**, Prediction scores of 4 selected KC near-enhancer sequences at each mutational step for KC model. The selected iteration (6th mutation) is indicated with a dashed line. After the 6th mutation, 4 more mutations are performed in FP3 to improve prediction score while 7 or 8 mutations are performed in the three other sequences to generate repressor sites. **j**, In vivo enhancer activity of the cloned WT genomic "near-enhancer" sequences with no enhancer activity. The expected location of KC is shown with dashed circles. Scale bars, 100 μm.

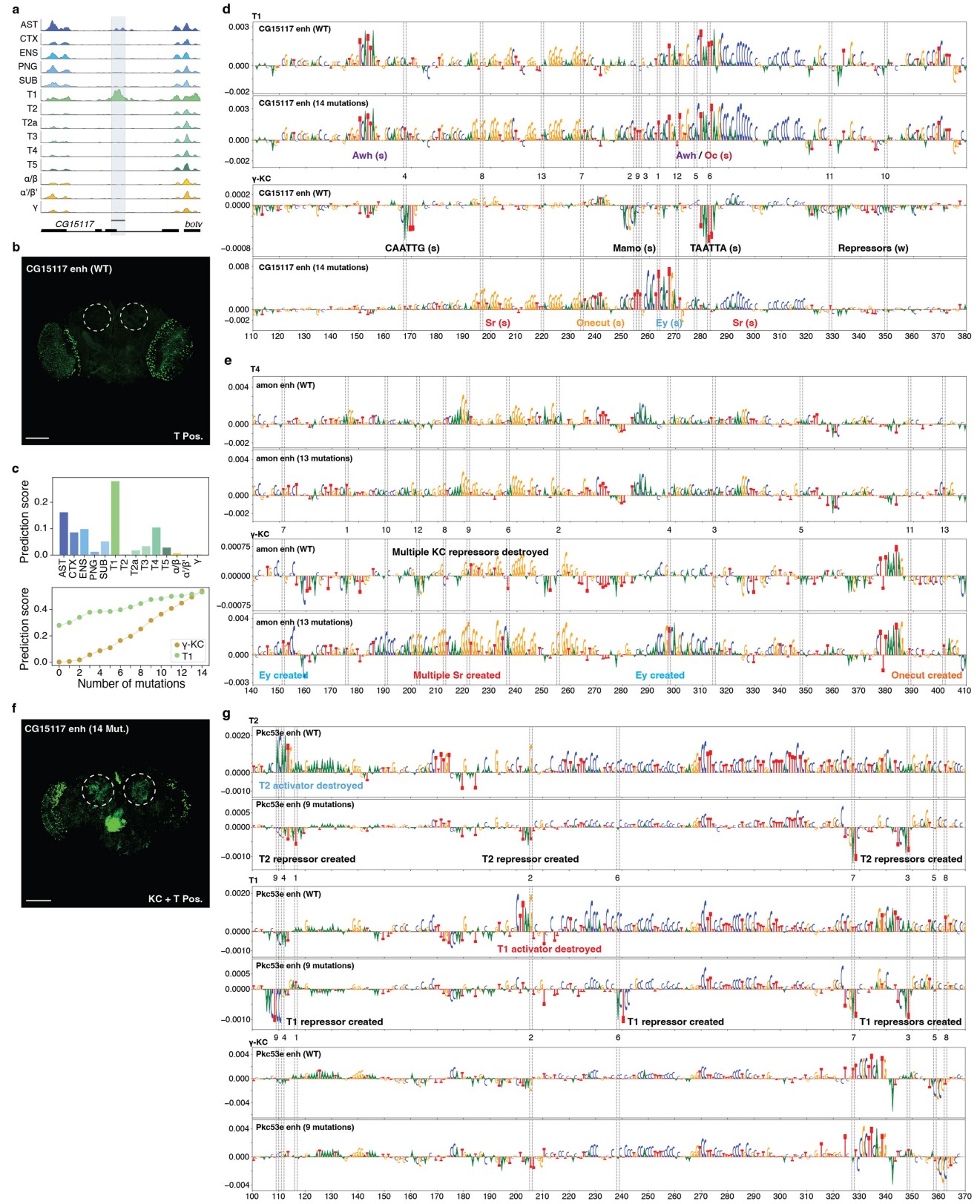

**Extended Data Fig. 3 | Enhancer design toward multiple cell type codes.**
**a**, Chromatin accessibility profile near *CG15117* gene. **b**, In vivo enhancer activity
of the wild-type (WT) *CG15117* enhancer. **c**, *CG15117* enhancer prediction scores
for each cell type (top) and prediction scores for the γ-KC and T1 classes after
each mutational step. **d**, Nucleotide contribution scores of WT *CG15117* enhancer
sequence and after 14 mutations for T1 (top) and γ-KC (bottom). **e**, Nucleotide
contribution scores of WT *amon* enhancer sequence and after 13 mutations
for T4 (top) and γ-KC (bottom). **f**, In vivo enhancer activity of the WT *CG15117*

enhancer and after 14 mutations. The CG15117 enhancer showed a slightly
altered T1 pattern following incorporation of the KC code. **g**, Nucleotide
contribution scores of WT *Pkc53e* enhancer sequence and after 9 mutations for
T2 (top), T1 (middle) and γ-KC (bottom). In **b**, **e**, the expected location of KC is
shown with dashed circles. Scale bars, 100 μm. In **d**,**f**,**g**, the position of the
mutations is shown with dashed lines, the mutational order is written in between
top and bottom plots and motif annotation is indicated with strong (s) or weak
(w) motif instances.

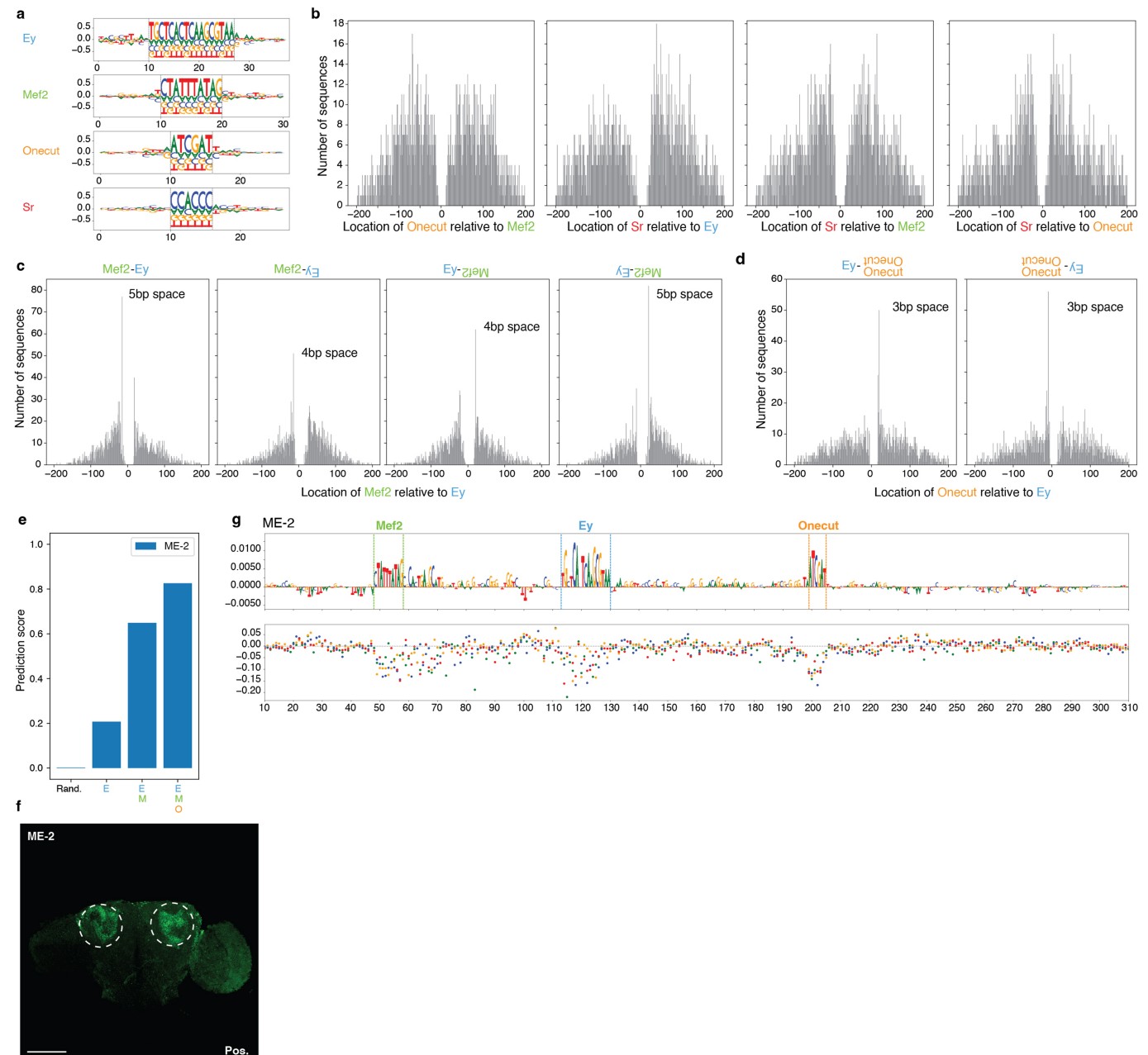

**Extended Data Fig. 4 | Enhancer design by motif implanting. a**, Preferred nucleotides flanking implanted motifs (n = 2,000). Dashed lines, motifs boundaries. **b**, Distribution of Onecut locations relative to Mef2, Sr to Ey, Sr to Mef2 and Sr to Onecut, respectively (n = 2,000). **c**, Distribution of Mef2 locations relative to Ey when both are on the same strand, Ey is on the negative strand, Mef2 is on the negative strand and both are on the negative strand, respectively (n = 2,000). **d**, Distribution of Onecut locations relative to Ey when Ey is on the positive strand and when Ey is on the negative strand, respectively (n = 2,000). **e**, DeepFlyBrain KC prediction score of the ME-2 sequence after consecutive motif implanting. **f**, In vivo enhancer activity of ME-2 enhancer. The expected location of KC is shown with dashed circles. Scale bar, 100 μm. **g**, Nucleotide contribution scores of the ME-2 motif implanting sequence (top) and in silico saturation mutagenesis assays (bottom). Each dot on the saturation mutagenesis plot represents a single mutation and its effect on the prediction score (y axis).

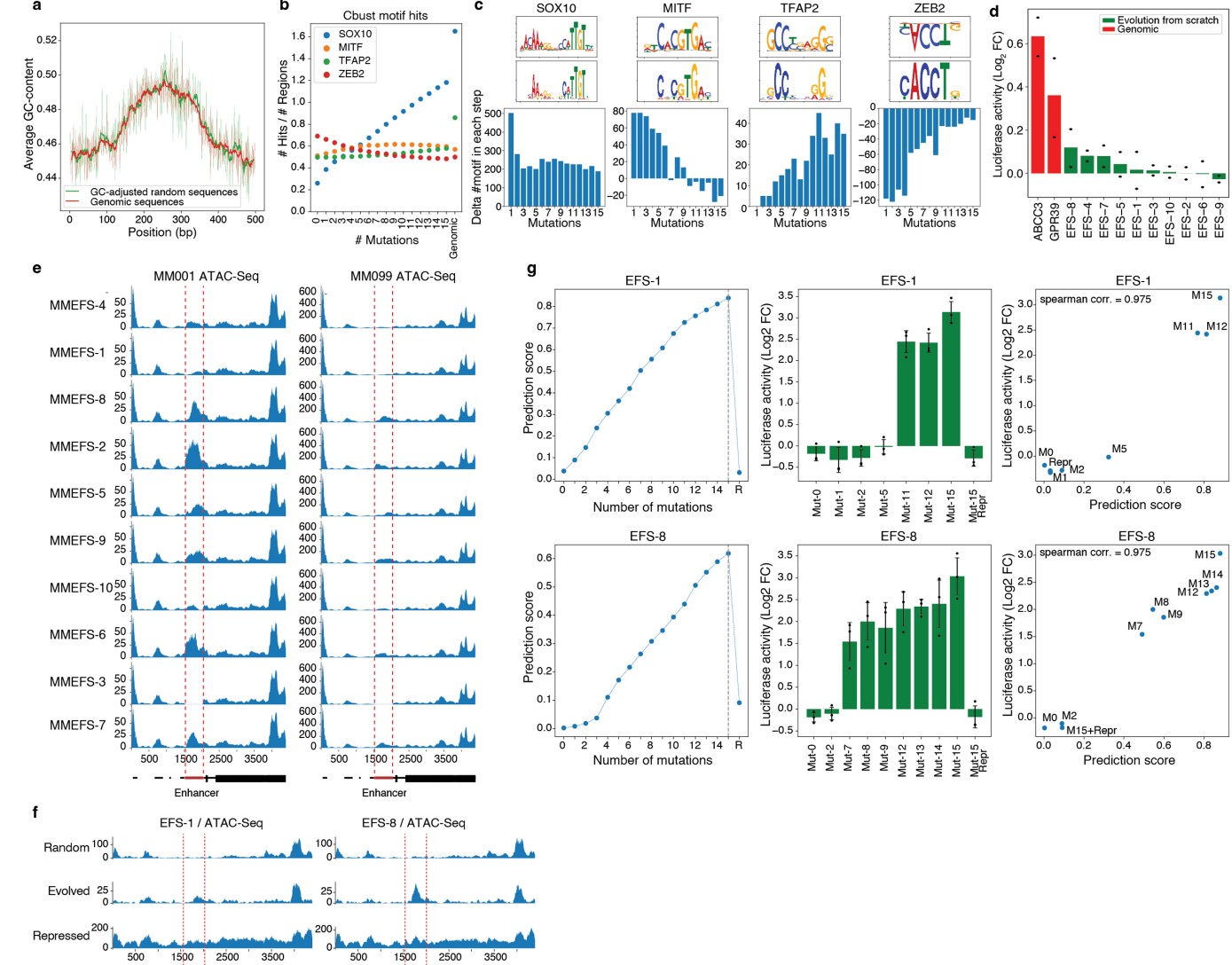

**Extended Data Fig. 5 | Human enhancer design by in silico evolution.**
**a**, Distribution of GC content in GC-adjusted random sequences (green) and human genomic regions (red). **b**, Average number of motif hits at each mutational step compared to genomic enhancers. **c**, Delta number of motifs in each mutational step. The TF-Modisco patterns and the most similar position weight matrices from the cisTarget motif database are shown at the top of each plot. ZEB2 upside-down pattern is contributing negatively to the model's prediction and is destroyed by the model on each step. **d**, Bar plot showing the mean luciferase signal (log$_2$ fold-change over *Renilla*) in a MES melanoma line (MM047) of the synthetic MEL enhancers (generated by in silico sequence

evolution), showing no activity compared to positive control genomic MES enhancers. The bar shows the mean (n = 2 biological replicates). **e**, MM001 (left) and MM099 (right) ATAC-seq profiles of all integrated lentiviral EFS reporters. **f**, MM001 ATAC-seq profile of 3 integrated EFS reporters: initial (top), evolved (middle) and post-evolution with repressive sites (bottom). **g**, DeepMEL2 prediction score (left), luciferase activity levels in MM001 (middle) and correlation between prediction score and activity (right) for EFS-1 (top) and EFS-8 (bottom) sequences after incremental mutation steps. In **e**, **f**, red dashed lines indicate boundaries of the enhancer. In **g**, the error bars show the standard error of the mean (n = 3 biological replicates).

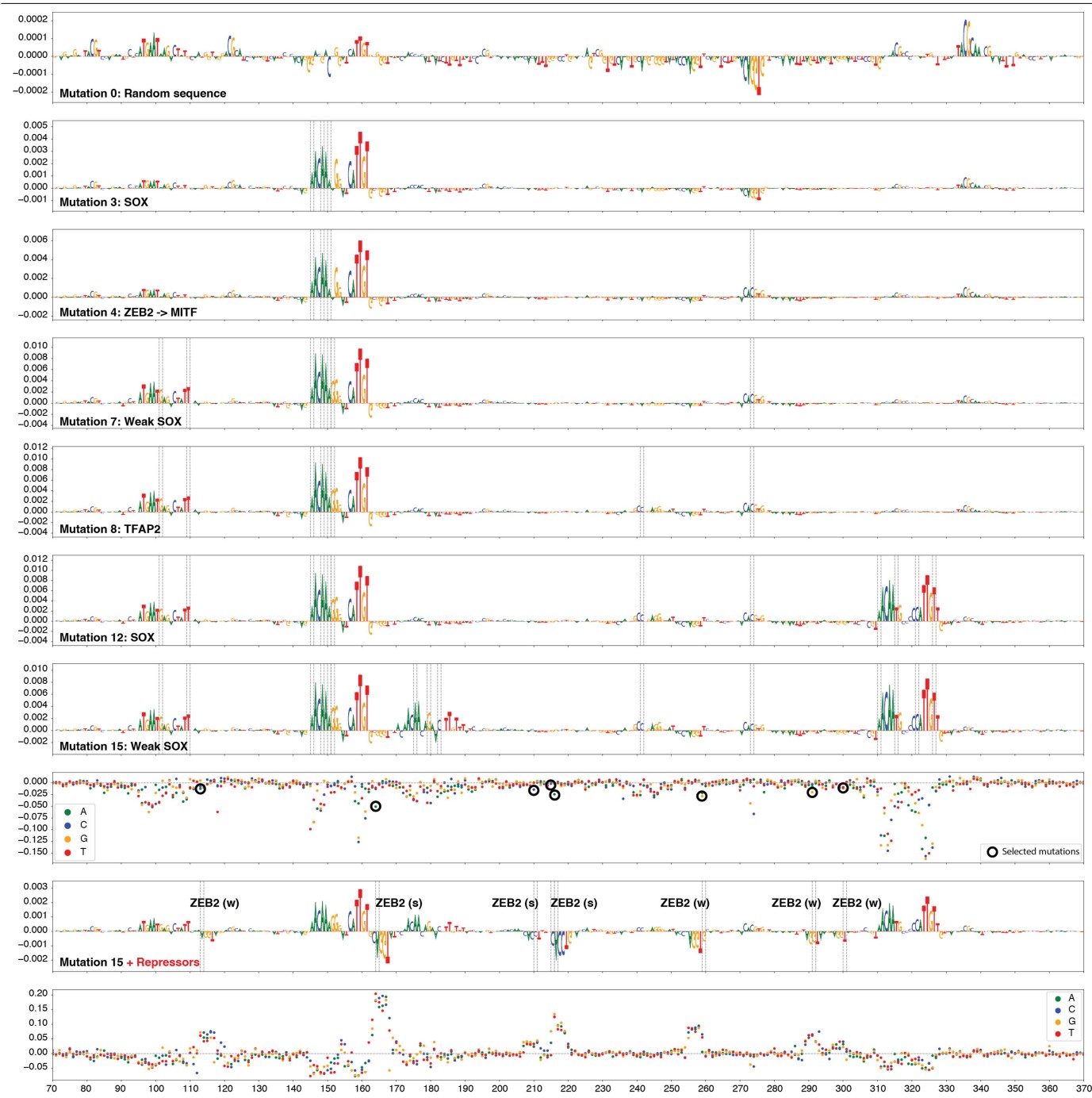

**Extended Data Fig. 6 | Intermediate steps of in silico evolution and generation of repressor sites in human generated enhancers.** Nucleotide contribution scores of EFS-4 at different mutational steps; 0 (random sequence), 3, 4, 7, 8, 12, 15, 15+Repressors. ZEB2 motif annotation is indicated with strong (s) or weak (w) motif instances.

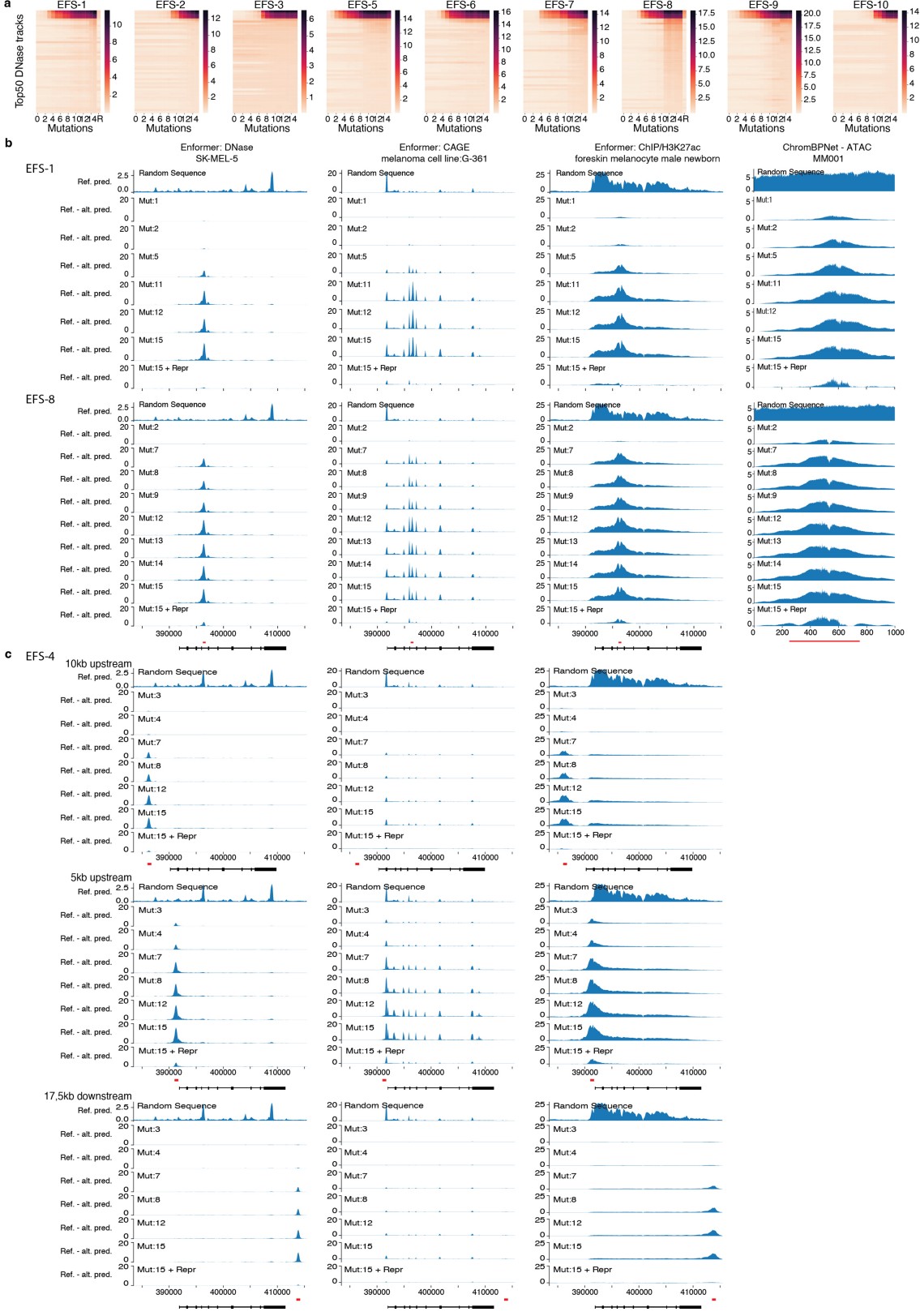

**Extended Data Fig. 7 | Human enhancer design by in silico evolution.**
**a**, Prediction scores for the top 50 DNase tracks for MEL EFS sequences. The four first DNAse tracks are: foreskin melanocyte male newborn, SK-MEL-5, foreskin melanocyte male newborn, SK-MEL-5. **b**, Enformer prediction tracks for three classes and ChromBPNet MM001 ATAC prediction tracks (right) for melanoma EFS-1 (top) and EFS-8 (bottom) sequences added in place of the *IRF4* enhancer. **c**, Enformer prediction tracks for three classes for melanoma EFS-4 sequences added 10 kb upstream, 5 kb upstream or 17.5 kb downstream of the *IRF4* enhancer. In **b**, **c**, top track: random sequence prediction score, other tracks: delta of mutated sequence prediction score vs random sequence prediction score.

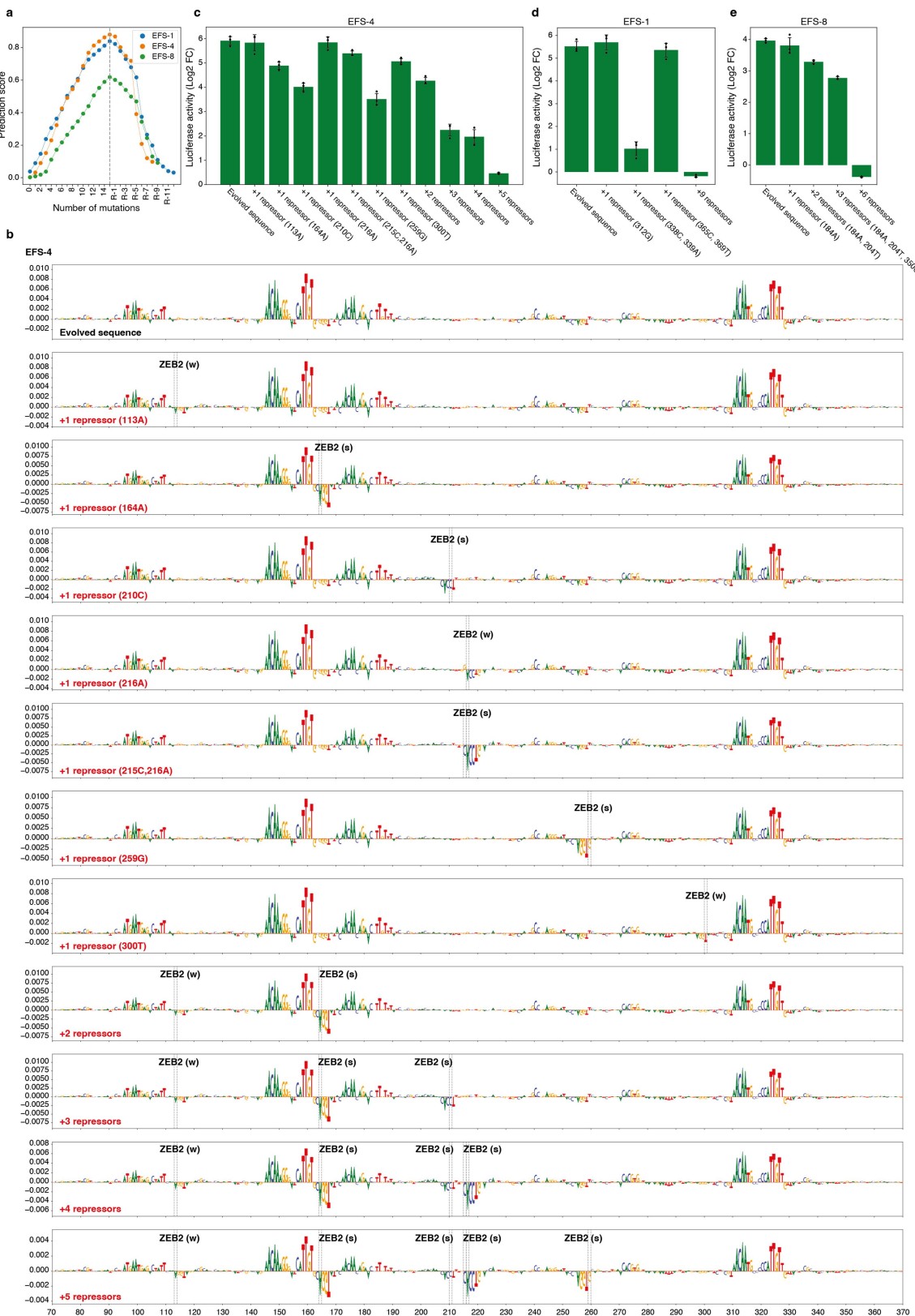

**Extended Data Fig. 8 | ZEB2 repression of in silico evolved MEL enhancers.**
**a**, Prediction scores for each mutational step and after the addition of repressor sites for 3 EFS sequences. **b**, Nucleotide contribution scores (DeepMEL2 MEL class) showing the creation of single or multiple repressor binding sites by single or double mutations in the EFS-4 sequence. **c**–**e**, In vivo enhancer activity of EFS-4 (**c**), EFS-1 (**d**) and EFS-8 (**e**) after the generation of repressor binding sites. ZEB2 motif annotation is indicated with strong (s) or weak (w) motif instances. The error bars in **c**–**e**, show the standard error of the mean (n = 3 biological replicates).

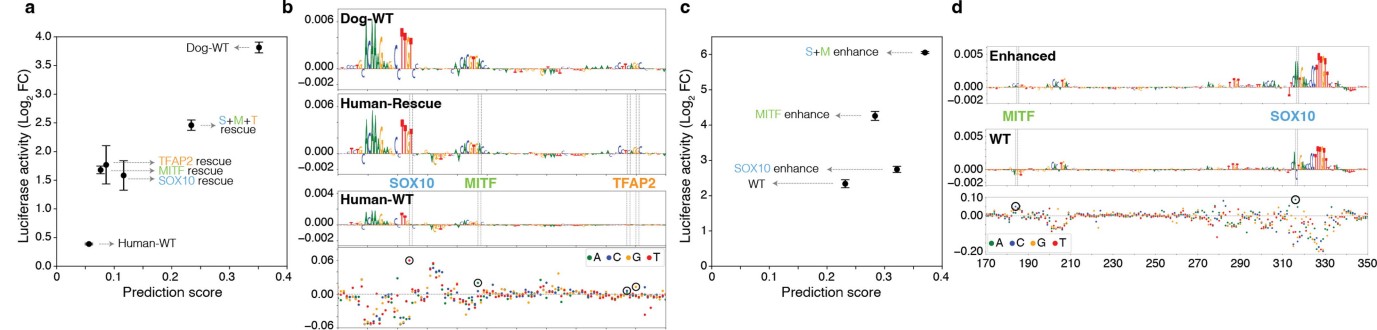

**Extended Data Fig. 9 | Human enhancer rescue.** In the fly brain, we applied in silico sequence evolution to create enhancers from genomic regions with high scores that did not show chromatin accessibility and could consequently be considered as 'near-enhancer' sequences. We extended this approach to MEL enhancers. We started from a human sequence that has no MEL enhancer activity, but its homologous sequence in the dog genome is accessible and active as MEL enhancer. We used DeepMEL to introduce 4 mutations that restored the activator binding sites in the human sequence, resulting in a rescue of the activity, as measured by luciferase activity. **a**, Dot plot showing the mean luciferase signal (log$_2$ fold-change (FC) over *Renilla*) versus prediction score for the MEL class of the WT human and dog genomic sequences and the rescued human sequences. **b**, Nucleotide contribution scores of the dog, human-rescued and human-WT sequences (top 3 rows) and in silico saturation mutagenesis assay of human-WT sequence (bottom). **c**, As a variation of this approach, we introduced two mutations in a weak MEL enhancer which resulted in a 10-fold increase in enhancer activity. Dot plot showing the mean luciferase signal (log$_2$ FC over *Renilla*) versus prediction score for the MEL class of the wild-type and enhanced enhancers. **d**, Nucleotide contribution scores of the wild-type (middle) and enhanced (top) enhancers and in silico saturation mutagenesis assay of wild-type enhancer (bottom). In **a**, **c**, the error bars show the standard error of the mean (n = 3 biological replicates). S, SOX10; M, MITF; T, TFAP2. In **b**,**d**, each dot on the saturation mutagenesis plot represents a single mutation and its effect on the prediction score (y axis). The position of the mutations is shown with dashed lines and circles.

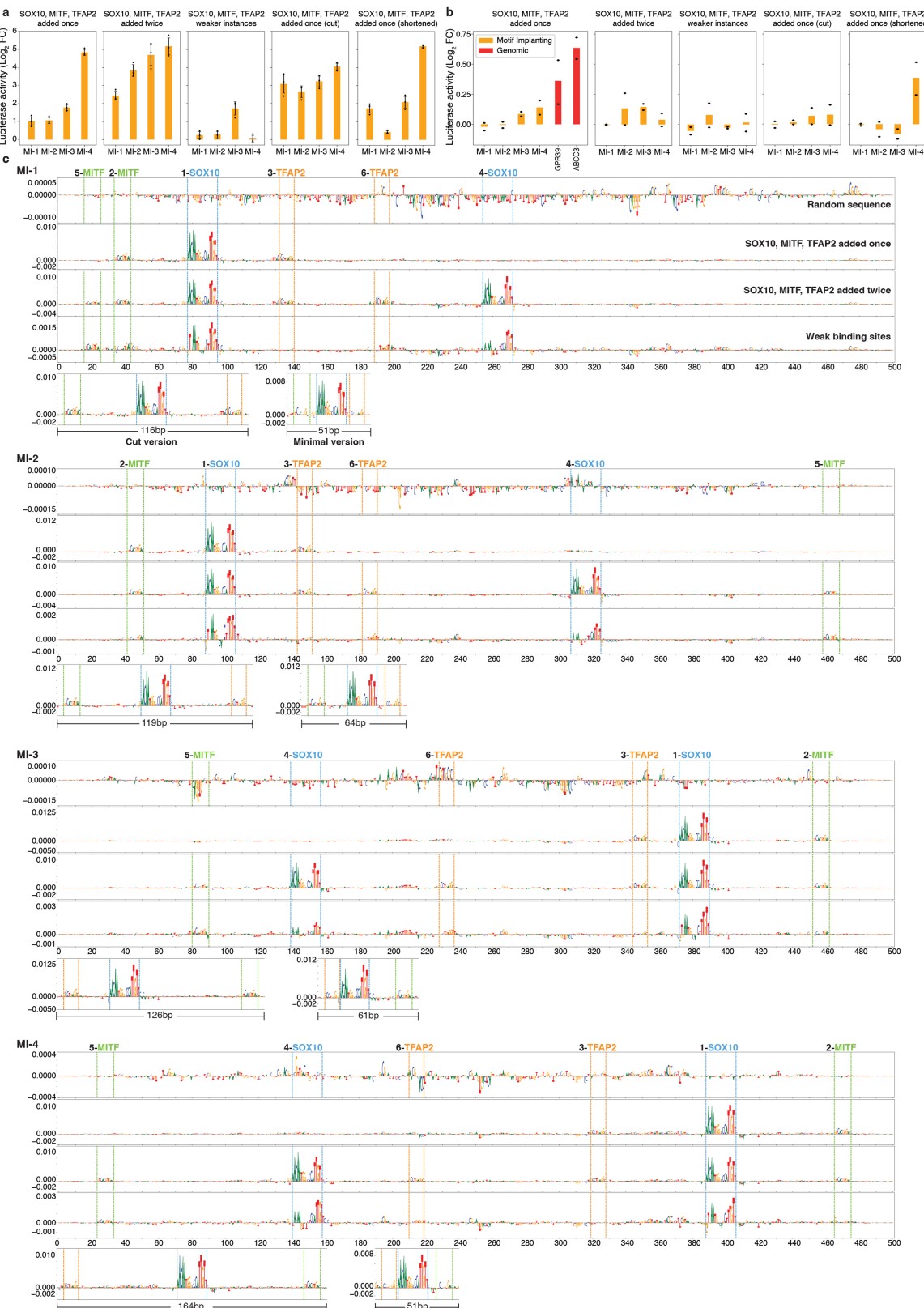

**Extended Data Fig. 10 | Human enhancer design by motif implantation.**
**a-b**, Bar plots show the mean luciferase signal (log$_2$ fold-change over *Renilla*) of the synthetic sequences, which were generated by motif implantation, tested in MM001 (**a**, MEL melanoma cell line, n = 3 biological replicates) and MM047 (**b**, MES melanoma cell line, n = 2 biological replicates). Values of 2 previously validated MES regions are displayed for MM047. The error bars in **a**, show the standard error of the mean. **c**, Nucleotide contribution scores of the selected synthetic sequences in their initial form (first row), after adding SOX10, MITF and TFAP2 motifs once (second row), after adding SOX10, MITF and TFAP2 motifs twice (third row), weaker-motif version of the third row after replacing implanted motifs with weaker sites (fourth row), cut version of the second row where only the part with the binding sites were taken (fifth row, left) and minimal version of the second row where MITF and TFAP2 are placed as close as possible to SOX10 (fifth row, right). The names of the motifs and their implantation order are indicated at the top. The position of the motifs is shown with dashed lines.

# Reporting Summary

## Statistics

For all statistical analyses, confirm that the following items are present in the figure legend, table legend, main text, or Methods section.

| n/a | Confirmed | |
|---|---|---|
| ☐ | ☒ | The exact sample size (*n*) for each experimental group/condition, given as a discrete number and unit of measurement |
| ☒ | ☐ | A statement on whether measurements were taken from distinct samples or whether the same sample was measured repeatedly |
| ☒ | ☐ | The statistical test(s) used AND whether they are one- or two-sided *Only common tests should be described solely by name; describe more complex techniques in the Methods section.* |
| ☒ | ☐ | A description of all covariates tested |
| ☒ | ☐ | A description of any assumptions or corrections, such as tests of normality and adjustment for multiple comparisons |
| ☐ | ☒ | A full description of the statistical parameters including central tendency (e.g. means) or other basic estimates (e.g. regression coefficient) AND variation (e.g. standard deviation) or associated estimates of uncertainty (e.g. confidence intervals) |
| ☒ | ☐ | For null hypothesis testing, the test statistic (e.g. *F*, *t*, *r*) with confidence intervals, effect sizes, degrees of freedom and *P* value noted *Give P values as exact values whenever suitable.* |
| ☒ | ☐ | For Bayesian analysis, information on the choice of priors and Markov chain Monte Carlo settings |
| ☒ | ☐ | For hierarchical and complex designs, identification of the appropriate level for tests and full reporting of outcomes |
| ☐ | ☒ | Estimates of effect sizes (e.g. Cohen's *d*, Pearson's *r*), indicating how they were calculated |

*Our web collection on statistics for biologists contains articles on many of the points above.*

## Software and code

Policy information about availability of computer code

| Data collection | Confocal images: ZEN blue 3.4.91 |
|---|---|
| Data analysis | Custom codes: https://doi.org/10.5281/zenodo.10184648 |

DL Python environment to use DeepMEL 1.0, DeepMEL2 1.0, and DeepFlyBrain 1.0:
python=3.7  tensorflow-gpu=1.15 numpy=1.19.5 matplotlib=3.1.1 shap=0.29.3 ipykernel=5.1.2 h5py=2.10.0

DL Python environment to train GAN models:
python=3.6  tensorflow-gpu=1.14.0 keras-gpu=2.2.4 numpy=1.16.2 matplotlib=3.1.1 shap=0.29.3 ipykernel=5.1.2

To perform motif analysis: TF-Modisco 0.5.5.4, Tomtom (MEME 5.5.1),  ClusterBuster 2022-04-21, BEDTools 2.30.0
To create higher-order background sequences: INCLUSive 3.2

To calculate statics: Scipy 1.6.0

To train ChromBPNet model: ChromBPNet 1.3-pre-release

ATAC-seq and ChIP-seq data analysis:
Demultiplexing with bcl2fastq 2.20
Adapter trimming with trimgalore 0.6.7
Mapping with bwa-mem2 2.2.1

Sorting with SAMtools 1.16.1
Deduplicating with SAMtools 1.16.1
Removing blacklist regions with SAMtools 1.16.1
Generating bigwig with deepTools 3.5.0
Peak calling with MACS2 2.1.2.1

For manuscripts utilizing custom algorithms or software that are central to the research but not yet described in published literature, software must be made available to editors and reviewers. We strongly encourage code deposition in a community repository (e.g. GitHub). See the Nature Portfolio guidelines for submitting code & software for further information.

## Data

Policy information about availability of data

All manuscripts must include a data availability statement. This statement should provide the following information, where applicable:
- Accession codes, unique identifiers, or web links for publicly available datasets
- A description of any restrictions on data availability
- For clinical datasets or third party data, please ensure that the statement adheres to our policy

Cloned Drosophila and human sequences were provided as Supplementary Tables. DeepMEL, DeepMEL2, and DeepFlyBrain deep learning model files were obtained from Kipoi (http://kipoi.org/models/DeepMEL, https://kipoi.org/models/DeepFlyBrain) with Zenodo record ids 3592129, 4590308, and 5153337. The fasta files used to train GAN models and the trained GAN models are available on Zenodo at https://doi.org/10.5281/zenodo.6701504. Custom genomes (hg38 and dm6) generated in this study are available on Zenodo at https://doi.org/10.5281/zenodo.10184648. Chromatin accessibility values in Kenyon Cells in adult Drosophila brains were obtained from GSE16369739. In vitro saturation mutagenesis on IRF4 data was obtained from https://kircherlab.bihealth.org/satMutMPRA/. Chromatin accessibility of Drosophila and transduced melanoma lines and ZEB2 ChIP-seq data generated for this study have been submitted to the NCBI Gene Expression Omnibus (GEO, https://www.ncbi.nlm.nih.gov/geo/) under accession number GSE240003.

## Research involving human participants, their data, or biological material

Policy information about studies with human participants or human data. See also policy information about sex, gender (identity/presentation), and sexual orientation and race, ethnicity and racism.

| | |
|---|---|
| Reporting on sex and gender | No human research participants involved |
| Reporting on race, ethnicity, or other socially relevant groupings | No human research participants involved |
| Population characteristics | No human research participants involved |
| Recruitment | No human research participants involved |
| Ethics oversight | No human research participants involved |

Note that full information on the approval of the study protocol must also be provided in the manuscript.

# Field-specific reporting

Please select the one below that is the best fit for your research. If you are not sure, read the appropriate sections before making your selection.

☒ Life sciences ☐ Behavioural & social sciences ☐ Ecological, evolutionary & environmental sciences

For a reference copy of the document with all sections, see nature.com/documents/nr-reporting-summary-flat.pdf

# Life sciences study design

All studies must disclose on these points even when the disclosure is negative.

| | |
|---|---|
| Sample size | The number of synthetic enhancers that were tested using transgenic flies was determined to be minimally 6 per cell type and it was bounded by the feasibility of the transgenic animal generation experiments. In total, 68 transgenic flies were generated.<br>The number of synthetic enhancers that were used with luciferase assays is determined to be minimally 10 per different category (in silico evolution, motif embedding, GAN, repressors, mutational steps). In total, 97 sequences were tested using luciferase assay. |
| Data exclusions | No data was excluded. |
| Replication | The same results were obtained from different replication experiments.<br>Multiple brains (at least 10) were stained and imaged for the fly experiments.<br>3 biological replicates were performed for the main luciferase experiments.<br>2 biological replicates were performed for the negative control luciferase experiments.<br>No biological replicates perfomed on ATAC-seq or ChIP-seq experiments. |
| Randomization | The initial random sequences (used for sequence evolution and motif implantation) were sampled from the sequence space that matches the |

| Randomization | GC content of the genomic sequences.<br>Flies fitting the gender(equal amount of male and female) and age (<10days) criteria were selected randomly for all experiments.<br>In this study, we didn't perform experiments that needed to be allocated into different groups. |
|---|---|
| Blinding | The investigators were blinded when performing cloning, transfection, antibody staining, and luciferase experiments by using enhancer IDs. |

# Reporting for specific materials, systems and methods

We require information from authors about some types of materials, experimental systems and methods used in many studies. Here, indicate whether each material, system or method listed is relevant to your study. If you are not sure if a list item applies to your research, read the appropriate section before selecting a response.

## Materials & experimental systems

| n/a | Involved in the study |
|---|---|
| ☐ | ☒ Antibodies |
| ☐ | ☒ Eukaryotic cell lines |
| ☒ | ☐ Palaeontology and archaeology |
| ☐ | ☒ Animals and other organisms |
| ☒ | ☐ Clinical data |
| ☒ | ☐ Dual use research of concern |
| ☒ | ☐ Plants |

## Methods

| n/a | Involved in the study |
|---|---|
| ☐ | ☒ ChIP-seq |
| ☒ | ☐ Flow cytometry |
| ☒ | ☐ MRI-based neuroimaging |

## Antibodies

| Antibodies used | 1 - Rabbit polyclonal anti-GFP (1:1000 dilution); Life Technologies CAT# A-6455; RRID: AB_221570<br>2 - Donkey polyclonal anti-rabbit Alexa Fluor 488 (1:500 dilution); Life Technologies CAT# A-21206; RRID: AB_2535792<br>3 - Rabbit anti-ZEB2; Bethyl CAT# A302-473A (1mg/ml and we used 5 micrograms for ChIP)<br>4 - Mouse anti-Dachshund (1:250 dilution); DSHB; CAT# dac1-1<br>5 - Alexa Fluor 647 goat anti-mouse IgG (1:500 dilution); Invitrogen, CAT# A-21235 |
|---|---|
| Validation | 1- References provided, statement on manufacturer's website: "This Antibody was verified by Relative expression to ensure that the antibody binds to the antigen stated.". Selected references out of 238: PMID 36067320, 35142344, 34908527, 34644579, 33932333, 33846330, 33463521, 33174166, 33112231, 32640222.<br>2- References provided, no statement on manufacturer's website. Selected references out of 6277: PMID 36067320, 35142344, 34908527, 34644579, 33932333, 33846330, 33463521, 33174166, 33112231, 32640222.<br>3- Testing and references provided, we performed ChIP-seq using ZEB2 antibody and the most enriched motif was the ZEB2 motif. No statement on the manufacturer's website. References: PMID 33614228, 20515682<br>4- References provided, statement on manufacturer's website: "The antibody reproduces the pattern observed by in situ hybridization with a dac cDNA probe (unpublished observations) and an enhancer trap insert in dac.". References: PMID 7821215, 17868668, 32781577, 18430931, 25670791, 8756723, 9845371, 24142104, 22874913, 34409041, 34322481, 33982759, 32738261, 32781577, 32184260, 31453329.<br>5- References provided, statement on the manufacturer's website: "The antibody "was used with a concentration of 2μg/mL.". Selected references out of 1448: PMID 35297981, 35017509, 33570489, 32878938, 32649914, 33659324, 32579612, 32317641, 37332603, 36879821, 36355348, 36649336, 34459871, 34605405, 33689682. |

## Eukaryotic cell lines

Policy information about cell lines and Sex and Gender in Research

| Cell line source(s) | MM001, MM047, and MM099 were obtained from Prof. Dr. Ghanem Ghanem with a Material Transfer Agreement.<br>HEK293T was obtained from ATCC (CAT# CRL-3216). |
|---|---|
| Authentication | We have used MM001, MM047, and MM099 in previous studies (Verfaillie et al., Nature Communications, 2015; Wouters et al., Nature Cell Biology 2020; Minnoye et al., Genome Research, 2020; Kalender-Atak et al., Genome Research 2021). We authenticated the cell lines by tracking their morphology overtime and by checking their genomic profile and mutations (Verfaillie et al., Kalender-Atak et al.), transcriptomic profile (Wouters et al.), and epigenomic profile (Verfaillie et al., Wouters et al., Kalender-Atak et al.).<br>HEK293T cells were only used for lentivirus production in this study, and the final products were tested and confirmed by sequencing. No authentication was needed for this cell line. |
| Mycoplasma contamination | Cell lines were tested for mycoplasma contamination, and were found negative. |
| Commonly misidentified lines<br>(See ICLAC register) | No commonly misidentified cell lines were used in this study. |

# Animals and other research organisms

Policy information about studies involving animals; ARRIVE guidelines recommended for reporting animal research, and Sex and Gender in Research

| | |
|---|---|
| Laboratory animals | Transgenic Drosophila melanogaster strains were used in this study. Young adult flies (<10-days-old) were used when performing antibody stainings. |
| Wild animals | No wild animals were used. |
| Reporting on sex | Sexes were equally mixed when performing antibody staining on adult Drosophila melanogaster brains. |
| Field-collected samples | No field-collected samples were used. |
| Ethics oversight | No approval required. |

Note that full information on the approval of the study protocol must also be provided in the manuscript.

# Plants

| | |
|---|---|
| Seed stocks | No plants were used. |
| Novel plant genotypes | No plants were used. |
| Authentication | No plants were used. |

# ChIP-seq

## Data deposition

☒ Confirm that both raw and final processed data have been deposited in a public database such as GEO.

☒ Confirm that you have deposited or provided access to graph files (e.g. BED files) for the called peaks.

| | |
|---|---|
| Data access links<br>*May remain private before publication.* | GSE240003 |
| Files in database submission | MM001_ZEB2_ChIP-seq<br>MM001_input_ChIP-seq |
| Genome browser session<br>(e.g. UCSC) | no longer applicable |

## Methodology

| | |
|---|---|
| Replicates | n=1 |
| Sequencing depth | ZEB2_ChIP-seq: 83410868<br>input_ChIP-seq: 168512695 |
| Antibodies | Rabbit anti-ZEB2; Bethyl CAT# A302-473A |
| Peak calling parameters | macs2 callpeak default parameters |
| Data quality | 31866 peaks are called with 5% FDR |
| Software | Demultiplexing with bcl2fastq 2.20<br>adapter trimming with trimgalore 0.6.7<br>mapping with bwa-mem2 2.2.1<br>sorting with SAMtools 1.16.1<br>deduplicating with SAMtools 1.16.1<br>removing blacklist regions with SAMtools 1.16.1<br>generating bigwig with deepTools 3.5.0 |

peak calling with MACS2 2.1.2.1

