## [Peer Review File · Nature]

Manuscript Title: Cell type directed design of synthetic enhancers

Reviewer Comments & Author Rebuttals

Reviewer Reports on the Initial Version:

Referees' comments:

Referee #1 (Remarks to the Author):

Taskiran and colleagues report a large-scale computational design of cell-type specific enhancers using Deep Learning in the manuscript titled “Cell type directed design of synthetic enhancers”. This study proposes the use of computationally driven *in silico* mutagenesis of DNA sequences guided by the DeepFlyBrain algorithm previously developed by the authors for design of Kenyon cell *Drosophila* enhancers (DeepMEL and DeepMEL2 algorithms were used for a parallel design of human melanoma cell enhancers). The authors investigate three types of sequence modifications—individual sequence mutations, motif implanting, and generative design—for establishing cell-line specific enhancer activity. They validate a handful of their enhancer predictions using GFP (*Drosophila*) and luciferase (human) reporter gene assays. The motivation behind generating synthetic enhancers is not to fuel any translational research, but to illustrate a ‘detailed tracing of enhancer features at nucleotide-level resolution’. This is an important area of research as biological manifestations of wild type enhancer constructs are largely unknown. Overall, the authors have provided a comprehensive *in silico* analysis of impact of changing sequence features on activator and repressor binding, which might underlie evolutionary pathways of enhancer gain and loss in vertebrate and invertebrate species. However, without sufficient experimental validation, the attempt to ‘decode the regulatory logic of enhancers’ would be incomplete.

Major concerns:

1. It should be clearly described in the manuscript that this study does not perform a design of cell-type specific enhancers *per se*. Instead, it effectively focuses on profiling the predictive characteristics of the DeepFlyBrain method (DeepMEL/2, in case of the human genome). As DeepFlyBrain is an accurate but far from ideal predictor of cell-type specific enhancers, profiling DeepFlyBrain predictions describes only a subclass of cell-type specific enhancers (possibly a subclass with simplistic motif compositions) and enhancer rules, and conclusions driven from this analysis do not necessarily describe either the “recipe” of cell-type specific enhancer design or the evolutionary trends underlying such enhancers.
2. Experimental validation of the synthesized Kenyon cell enhancers indicates rather inaccurate computational predictions (4/6 validation rate) and is limited in scope. It would be beneficial to increase the number of tested sequences from just a handful of arbitrary selected examples and to analyze the source of false positive predictions.
3. Given that the “evolutionary” design of the proposed model is driven by the assumption of the DeepFlyBrain scoring system being a reliable indicator of enhancer activity at each iteration of the enhancer design, this relatively low validation rate raises concerns of the applicability of the proposed “evolutionary trends” (repressor binding deactivation followed by acquisition of activator

sites) to the natural evolution of enhancer sequences. In particular, the experimental validation of the repressor binding deactivation (which is proposed as the foundation of enhancer activation / preprogramming) appears necessary. Please experimentally demonstrate that a) the binding of repressor proteins is affected by the first round of mutations and b) repressor deactivation has an expected impact on the enhancer activity.

4. The minimal number of mutations needed for cell-type specific enhancer formation is discussed in detail for *Drosophila* and human enhancers. However, this metric is based on a sequential selection of top DeepFlyBrain scoring nucleotide changes. The authors should demonstrate that the same number of multiple lower-scoring changes are not capable of achieving a stronger impact on the enhancer activity. While there might be a limit to combinatorial testing of all possible mutations, it should be feasible to quantify all possible 5-6 mutation combinations using a computational approach and to compare it to the same number of sequential top-scoring mutations. Also, detailing the impact of individual sequential mutations on the gradual change of enhancer function would strongly benefit from experimental annotation of enhancer activity at every step of enhancer reprogramming.

5. The paragraph starting on line 128 is confusing. If the premise of this method is based on “high prediction scores”, why are the selected elements with the “high prediction scores” not acting as enhancers and require additional mutations to turn into enhancers? Additionally, if the selected regions have high prediction scores, then do the six mutations correspond to positions that impart repression through chromatin inaccessibility?

6. Line 170. Would it be correct to assume that the ‘second cell type’ is KC? Did the augmented enhancer preserve its activity T1 intact? Mutations often coevolve; testing the original activity of augmented enhancer in its original cell type (T1) would be essential to corroborate the statement made in lines 157-159. In addition, though probably evident, expanding the legends of figure 3 (panels b, d and f) to include cell types would provide clarity.

7. Is it always necessary that mutations that abrogate the activity of an enhancer in a cell type will do so necessarily by creating a repressor binding site? To this end, it would also be pertinent to highlight the T1 specific repressors for which binding sites were generated (lines 180-181).

8. In section ‘Enhancer design by motif implantation’, it would be more informative to provide an assessment of occurrence and spatial arrangement of various activator binding sites in *Drosophila* cell lines (and similarly for human cell lines) or cite relevant literature.

9. Line 409, if I understand correctly, “for example” should be replaced with “furthermore/in addition”. In the same paragraph, the discussion of close-to-zero scores of random sequences does not necessarily imply that they contain multiple repressor binding sites (unless the authors illustrate the repressors and their binding site motifs). Further, to correctly understand the trajectory of sequence evolution, it would be profoundly more useful to assess the binding of these repressors (as well as activators) after each mutation to comprehensively understand the evolution of wild type enhancers. Such an analysis would also provide an understanding of 1) the correlation between DeepFlyBrain prediction scores/DeepExplainer contribution scores and in-vivo changes in enhancer activity w.r.t. binding sites, chromatin accessibility, etc., 2) co-evolving sites, if any, and 3) interplay of various players in the enhancer region, such as protein binding sites, contextual sites, chromatin accessibility defining sites, etc. It would also address the key motivation behind this work, i.e., providing “a thorough understanding of how enhancer activation is encoded in its DNA sequence” (line 54).

10. The foremost logic provided in this manuscript behind the emergence of enhancer sites in

random sequences is the conversion of repressor to activator binding sites. However, the approach used to label these sites in DNA fragments is not provided/cited anywhere in text, for e.g., in lines 110, 116, figures 2, S1. It will be useful to cite appropriate references and provide a brief description in Methods. Alongside, repressors/activators, if known, associated with sites may also be mentioned.

Minor concerns:

1. Inconsistent usage of terms. T1 class, T-neurons, T1-neurons, all refer to the same cell type. Similarly, γ -KC, gamma-KC, KC cells, KC class, and so on.
2. References 20 and 23 should be quoted on line 59 for enhancer mutational analysis.
3. Several phrases such as 'KC class', 'activity pattern', should be defined at the first mention.
4. Line 106 "we investigated the effect of the initial (random) sequence," is incomplete. Effect on what?
5. Quantify the word 'usually' in line 116.
6. Expand the abbreviation EFS
7. In line 271, how was the strength of binding sites quantified?
8. Line 272, 'that were predicted as KC enhancers..'. .
9. Line 412, cite reference/provide data.

Referee #2 (Remarks to the Author):

This manuscript entitled "cell type directed design of synthetic enhancers" by Taskiran et al presents 3 methods (previously established by other labs) that can aid in the design of functional enhancer sequences. They demonstrate the 3 methods using DeepFlyBrain and DeepMel (both of which have been established in previous papers by the corresponding author's lab). A small subset of predictions are validated experimentally. It is a solid demonstration that adds to the growing literature of the feasibility of using neural network predictions as a way to guide the design of functional sequences.

The strength of this paper is the validation with experiments. Nevertheless, as the manuscript currently stands, it has many more weaknesses that should be addressed in terms of clarity and depth of analysis to become a more impactful paper.

General concerns

- Credit is not given clearly. For each method, the authors should do a better job of giving credit as they have all been previously established. For instance, in silico evolution (Vaishnav et al. Nature, 2022), motif implantation (Koo et al. PLoS Comput Biol, 2021; Avsec et al, Nat Genetic, 2021), and GANs (Killoran et al, arXiv, 2017; Gupta and Zou, Nat Mach Intell, 2019). In addition, the authors should provide some additional context of what differences exist (if any) with the proposed method and established methods.

- There is no mention of other approaches that have found success at designing regulatory genomic

sequences. There are several other approaches that have designed functional genomic sequences, including input maximization (Bogard et al. Cell, 2019)) and deep exploration network (Linder et al. Cell Systems, 2020), which is similar in spirit to GANs. Each has performed experimental validation as well.

- The biological sequence design literature is much more well established for protein sequences, with many compelling examples. Although not a strict requirement, it would be nice to see the connections (if any).
- The depth of analysis is rather shallow in many areas, especially with GANs. Many of the claims are not supported quantitatively.
- Lack of reproducibility. The code availability is an html file of jupyter notebooks that supply basic examples of code utility. The authors should supply a more thorough set of `_usable_codes` -- both as a tutorial for other people that would like to use these computational tools and a more comprehensive set of code that reproduces the analysis (including figures) in this manuscript. Also, there is no way to run the code without the proper files, including the data (DeepFlyBrain_data.pkl) and models (json, hdf5). All data and code should be uploaded to zenodo (not just for partial GAN analysis).
- For each method, what are the characteristics of the sequences that did not yield significant expression experimentally? Why did they not work? Is it a poor regulatory code or poor sequence context?
- Thoughtful discussion of limitations. All of the predictions rely on deepflybrain or deepmel. Any biases they learn will be propagated to the enhancer design predictions. There is no discussion of the limitations of this model-based sequence design strategies.

Enhancer design by in silico evolution

- The authors should cite in silico mutagenesis (Alipanahi et al. 2015; Zhou & Troyanskaya, 2015, Kelley et al. 2016) and in silico sequence evolution (Vaishnav et al. Nature, 2022). Also, is there any difference between what was done in Vaishnav and here. If any, it should be clarified. If not, it should be explicitly stated that that method was used here.
- Many of the stated claims are anecdotal. For instance, short repressor binding sites are present in random sequences and they are the targets of deleterious mutations during the early rounds of in silico evolution. However, the scale of the attribution maps suggest that these short repressors in random sequences are very weak.
- What defines a repressor binding site? Are these sites strong or weak repressor binding sites or partial binding sites? Perhaps a PWM track could clarify this.
- What if a strong repressor binding site were present?
- suggestion: to support any claims quantitatively, one could be to look at the frequency of mutations at repressor sites vs other sites early during the evolution process (across a population of

test sequences).

- suggestion: One could implant strong repressor binding sites to see how the single nucleotide-based evolution cycles. This can be compared to weak repressor binding sites.

- The claim "this suggests that KC enhancers... can arise de novo in the genome with few mutations." is not strongly supported. The authors should emphasize the assumption that the region must be accessible in the first place. This model (i.e. DeepFlyBrain) does not understand temporal ordering of TF binding but rather a steady state assumption. Causation and correlation cannot be resolved by this (or similar sequence-function) model.

- There are other areas outside the annotation in Fig. 2f and 2j that have increased green intensity. What are these cell types? Do these also agree with predictions for other cell types?

- From the attribution maps, there seems to be a positional bias of motifs to be near the center of the sequence. If trained on peak centered data, then models can be biased to focus on motifs near center of sequence. In this scenario, models may then have blind spots at the 3' and 5' flanks. If important motifs or context were to be present here, they may be ignored by the model.

- suggestions: plot the positional distribution of (positive) motifs that increase function at the end.

- suggestion: An interventional experiment can also be performed where the motif positions are shifted to be closer to the 3' or 5' flanks. If the model predictions do not change much, then the model is robust to translational shifts and there likely is not a learned bias by the original DL model. If the predictions change significantly, then it could be that there is a positional bias or that important sequence context that was not captured by the attribution map is now missing.

- Among the final evolved sequences, how many motifs are present? What are the frequencies observed of each motif type? Do they have a specific order/grammar?

- For the attribution plots, how were motifs annotated on attribution plots? How was a repressor site identified? What are the motifs of repressors?

Enhancer design towards multiple cell type codes

- the enhancer CG15117 enhancer in T1 neurons was not predicted to be highly active, despite the authors claim that it is well characterized. The prediction is around 0.25. Does that make sense given it is well characterized?

- The statement "The nucleotide contribution scores before and after the in silico evolution show that the T1 enhancer code was barely touched". Although the nucleotide contribution scores did not change much, the prediction increases from 0.25 to 0.5. Warning: despite the literature in deep learning applications in genomics, independent nucleotide contributions are not explanations of predictions. This is well established in ML literature (across several papers, eg. Lipton, "The Mythos of Model Interpretability" arXiv, 2016) but has largely evaded the genomics field. Attribution maps should not be over-interpreted. They only provide a hypotheses at best for what a model has learned.

Enhancer design by motif implantation

- The authors should show the intermediate plots of motif scanning in supplemental figs. This can also reveal any positional bias learned by the model.
- How were motif instances determined? Were the flanks optimally chosen (as was done in DeepStarr)?
- The analysis is very similar to the in silico experiments in BpNet (Avsec et al. Nat Gen 2021) and GIA (Koo et al, PLoS Comp Biol, 2021), and deepStarr (de Almeida et al. Nat Gen 2022), which uses GIA. BpNet and DeepStarr also validate results experimentally. This application is slightly different (i.e. designing enhancers) but the overall goals are largely the same  identify motif code that drives high activity in respective system. The authors should do a better job of highlighting previous work, and ensuring that they distinguish the contributions (if any) of this approach versus previous approaches.
- The claim at the end of the section that just 3 binding sites with a minimal length is sufficient to serve as a functional enhancer is not fully supported. This is from an anecdotal example.
- Suggestion: If authors want to make this claim, they should implant this motif combination in different backgrounds and map out a distribution. This would test whether it is just the combination of motifs or whether the combination with sequence context is important.

Enhancer design by GANs

- Again, should reframe first sentence because it makes it seem like the GAN for DNA sequences is a different field. The authors should mention here that the GAN was a copy of an existing method with very much the same architecture. Credit isn't given until the methods. To a casual reader, this makes it seem like the method is novel. Also, another GAN was applied to DNA sequences previously, feedback GAN, though it was meant for protein coding regions, not regulatory genomics (Gupta and Zou, Nat Mach Intell, 2019).
- The GANs here are not well characterized beyond the positional GC bias in the sequences. What is the distribution of predictions of the GAN generated sequences? How does that compare with the predictions of the original genomic sequences?
- Fig. 5b shows correctly predicted sequences. Does this mean predictions were above a 50% threshold? How would random sequences that contain the GC-bias do here? This would be a better negative control than the Markov model which doesn't learn the positional GC-bias.
- To better characterize the generated sequences, the authors should do motif enrichment analysis of genomic sequences and GAN sequences to further characterize the number of motifs within GANs, their specificity/similarity to strong motifs (vs weak motifs), and the diversity of motifs learned.

- The attribution plots look like the values are much lower than genomic sequences. What is the distribution of attribution scores for positions with motifs in GAN generated sequences compared to genomic sequences that have high enhancer activity (i.e. high accessibility prediction)?

- The GANs generate softmax sequences -- continuous values and not one-hot. One hot sequences can be generated with the Gumbel-Softmax trick (Jang et al, arXiv, 2016) and be differentiable so that gradients can backpropagate to the generator. It seems more natural for discrete sequences. Why was a softmax chosen instead?

Referee #3 (Remarks to the Author):

In this study by Taskiran and colleagues, the power of deep learning from transcriptional regulatory elements is turned to generate novel, cell-type specific regulatory elements, following the Feynman dictum of "if you can build it, you can understand it". The aim of the work is to explore specific models for enhancer structure and function, gaining insights on possible evolution of enhancers, the distal regulatory elements that control expression of most genes in higher eukaryotes. Overall, this is a highly ambitious goal, and the authors make impressive progress in identifying the underpinnings of cell type expression using two systems, specific KC cells of the *Drosophila* brain, and "MEL" type melanoma gene expression. Both are supported by previous deep learning studies by the Aerts group, in which endogenous sets of differentially accessible chromatin and gene expression were carefully studied to train predictive models.

In the paper, the authors use three general approaches to evolve regulatory elements *in silico*, with *in vivo* validation of sample products. Exhaustive *in silico* mutagenesis, with deep learning models scoring the proposed elements at each step, are used to generate specific enhancers in less than 20 steps. A second approach assesses the predicted activity of background sequences into which binding site motifs of specific transcription factors are implanted, again with models scoring the expected output as positions are moved across the elements. A third approach uses generative adversarial networks, which have been used extensively in image analysis. Each approach succeeds to a certain extent in generating cell-type specific enhancers from random sequences, from genomic sequences that have sequence qualities of active enhancers but were scored as not accessible, or from enhancers that are active in a different cell type. The chromatin-based assessment for *Drosophila* indicates that many of the *in silico* evolved elements are functional in a highly cell-type specific manner (judged by GFP expression in KC cells, for example), while the MEL elements are assessed in transient transfections, a non-chromosomal setting.

Overall, the authors conclude from this work that deep learning provides the tools to efficiently identify cell-type specific elements. From the analysis of the *in silico* evolved enhancers, they conclude that the restrictions on generating a functional enhancer are modest; there is a certain amount of clustering of motifs, and in MEL elements some preferred binding geometries, but apparently many configurations are sufficient for the specificity. Prior difficulties in developing novel enhancers *ab initio* are attributed in part to the inability to pre-score likely winners (a benefit of deep learning models) as well as the observation here that many inactive random sequences have too many repressor binding sites, which must be eliminated. Generation of novel enhancers *in vivo*

are suggested to lie as close as a few mutations away, which puts a surprisingly low bar for remodeling of GRNs through creation of new regulatory elements.

The results of this study are compelling, and in silico wed properly with in vivo validation. Especially compelling is the ability to mine the information developed by deep learning to generate likely active elements, which would appear to be an important extension of the field. There are a number of areas that the authors might consider to strengthen the impact of this work, noted below:

1. The study is based on models trained on endogenous regulatory elements, or putative elements, and the predicted modules are termed “enhancers”, yet the context in which the activity of the binding sites are actually tested in the fly or cell are much more permissive, namely, within close range of the basal promoters. The finding that even a small ~50bp cluster of binding sites works as an enhancer suggests that we are actually observing the interaction of some regulatory sites in close proximity to targets in the promoter. The demands for gene activation may therefore be heavily shaped by the megadalton complexes of basal actors that can already set up a chromatin context suitable for the cell-type factors to work within. A more refined consideration of what is being identified as a functional element would be helpful in the interpretation. The low threshold to mutate into an active element may actually be higher if distal elements need to overcome chromatin barriers, or have interaction Kds great enough to maintain looping over a considerable distance, against entropic effects.

2. Some metazoan regulatory elements are capable of exhibiting a high degree of cell-type specificity using a very simple logic; if that factor is capable of driving gene expression by itself, and if the cell-type specificity is “computed” by enhancers that integrate developmental information and drive cell type expression of the factor, then a few binding sites for Pax proteins (for instance) will give eye-specific transcription. Many other elements that have been dissected don’t have such simple design. The sparkling enhancer analyzed by Barolo and colleagues shows a confounding degree of complexity, while the shaven-baby elements studied by Fuqua and Crocker exhibit extreme sensitivity to individual mutations, such that all sequences appear to influence the output. The authors should discuss evidence for whether the simple grammar identified here is perhaps representative of a certain class of elements, and how one would distinguish that if so. Overall, a broader consideration of types of enhancer structure/function would strengthen the Discussion.

3. The in silico evolution of the elements tested, as they climb their ways to activity, appear to be a gold mine of possible insights on action of enhancers. The authors share some general insights, such as replacement of repressors with activators, and some spacing requirements. It would be very useful to quantitate some of these (predicted) effects. For instance one notes that in some cases shown, existing binding motifs for repressors that are distal to actual mutations go from strongly repressive to not repressive at all as activator sites appear. E.g. Supp. Fig. 1 EFS-1 repressor site at 270; Fig. 2D repressor at 390 disappears. Perhaps the deep learning is indicating that repressor sites work in a cooperative fashion, and with the replacement of one site, the distal one is rendered inert? For this effect and others, it would be quite interesting to understand the trends that may drop out of the thousands of in silico experiments conducted.

4. The number of tested elements is small, but there are “successes” and “failures” – are there properties of each that one can draw on to understand the shortcomings of the models that are

driving evolutionary design?

5. Some aspects of the experimental procedures were unclear. For example, the fly elements are tested in the PH-Stinger vector (reference is needed), which is a P-element random insertion vector. Yet, the Materials and Methods indicates directed insertion, as if using a phiC31 system. The cloning indicates that elements were introduced into the vector using a lambda-based recombinase system, yet Stinger has restriction sites for cloning upstream of the -43 site of hsp70. Was this vector altered to allow for in vitro recombination, and if so, what is the distance of elements to basal promoter?

6. The evolved elements tested in MEL cells are more active than they are in MES cells, but it appears that even “native” enhancers are much weaker in MES cells, relative to Renilla. Are MES cells poorly transfected, or is the Renilla much more active in this cell type? Because of normalization, it is hard to know if the elements tested are robust in expression.

7. Some minor points:

Fig. 3 m, n – “Chromatin accessibility profile of this region” Presumably this is Pkc53e? Not clear from the lack of label in Fig. 3n.

Renilla is capitalized.

“prediction scores started to saturate [plateau?] after 20 mutations”

Author Rebuttals to Initial Comments:

Referees' comments:

Referee #1 (Remarks to the Author):

Taskiran and colleagues report a large-scale computational design of cell-type specific enhancers using Deep Learning in the manuscript titled “Cell type directed design of synthetic enhancers”. This study proposes the use of computationally driven *in silico* mutagenesis of DNA sequences guided by the DeepFlyBrain algorithm previously developed by the authors for design of Kenyon cell *Drosophila* enhancers (DeepMEL and DeepMEL2 algorithms were used for a parallel design of human melanoma cell enhancers). The authors investigate three types of sequence modifications—individual sequence mutations, motif implanting, and generative design—for establishing cell-line specific enhancer activity. They validate a handful of their enhancer predictions using GFP (*Drosophila*) and luciferase (human) reporter gene assays. The motivation behind generating synthetic enhancers is not to fuel any translational research, but to illustrate a ‘detailed tracing of enhancer features at nucleotide-level resolution’. This is an important area of research as biological manifestations of wild type enhancer constructs are largely unknown. Overall, the authors have provided a comprehensive *in silico* analysis of impact of changing sequence features on activator and repressor binding, which might underlie evolutionary pathways of enhancer gain and loss in vertebrate and invertebrate species. However, without sufficient experimental validation, the attempt to ‘decode the regulatory logic of enhancers’ would be incomplete.

We thank the reviewer for their constructive comments. For the revision we particularly focused on adding many more experimental validations, as we describe throughout the rebuttal.

Major concerns:

Remark 1.1: It should be clearly described in the manuscript that this study does not perform a design of cell-type specific enhancers per se. Instead, it effectively focuses on profiling the predictive characteristics of the DeepFlyBrain method (DeepMEL/2, in case of the human genome). As DeepFlyBrain is an accurate but far from ideal predictor of cell-type specific enhancers, profiling DeepFlyBrain predictions describes only a subclass of cell-type specific enhancers (possibly a subclass with simplistic motif compositions) and enhancer rules, and conclusions driven from this analysis do not necessarily describe either the “recipe” of cell-type specific enhancer design or the evolutionary trends underlying such enhancers.

Answer 1.1: Thank you for this important comment.

Generalization to other cell types. We had already studied cell types in *Drosophila* and human to specifically examine this aspect of generalizability of our approach. To further strengthen this in the revision, we now also designed enhancers for **perineurial glia** (PNG) (Fig R1.1a). Interestingly, we started the design process from the same random sequences as we used for Kenyon cells (Fig R1.1b,c). Glial cells, and also the T1 neurons that we use for the dual-code enhancers, have highly different motif compositions compared to the (most worked-out) Kenyon cells enhancers. Based on this comparison of different cell types, we do not find that the Kenyon cell enhancers are more simplistic.

*New data regarding PNG enhancer design and validation can be found in Figure 2e,f and Supplementary Figure 2e,f and is presented in the “Enhancer design by *in silico* evolution” section.*

Figure R1.1 (Figure 2d,e,f): **a**, In vivo enhancer activity of the cloned sequences. Top-middle: initial random sequence, top-left: random sequence after 10 mutations toward KC evolution, top-right: random sequence after 15 mutations toward PNG evolution, bottom: Three other random sequences after mutations toward PNG evolution. **b**, Nucleotide contribution scores of a selected random sequence in its initial form (top) and after KC-directed mutations (bottom). The position of the mutations are shown with dashed lines. The mutational order is written in between top and bottom plots together with the type of nucleotide substitutions. **c**, Nucleotide contribution scores of the same initially selected random sequence as **b** after PNG-directed mutations. The position of the mutations are shown with dashed lines.

Comparing complexity of genomic and synthetic enhancers. We performed a range of new analyses, designs, and experiments to unravel the complex motif compositions of human melanoma enhancers, particularly regarding the combination of activator and repressor motifs. Again, from these findings, we could not identify aspects that could classify *Drosophila* Kenyon cell enhancers as more simplistic compared to human melanoma enhancers. Kenyon cell enhancers are controlled by at least 3 repressor motifs, and are therefore more complex in regard to repressor architecture. For each cell type that we used in this work to design enhancers for, we identified 3-4 activator motifs and 1 or more repressor motifs (Fig R1.2).

Figure R1.2 (panel **a**: Figure 5a and Figure 6d; panel **b**: only in rebuttal (top) and Figure 2d (bottom); panel **c**: Figure 2e): Illustration of comparable complexity of genomic enhancers and synthetic enhancers for both human melanoma (**a**) and *Drosophila* KC (**b**) and PNG (**c**) enhancers.

Comparing complexity to other systems. Below, we highlight the complexity of enhancers from other cell types published in previous studies, illustrating comparable complexity with the enhancers in our chosen cell types, for example in the number of motifs and their arrangement (Fig. R1.3).

Ameen et al., 2022 (Fig.2c,d)

Almeida et al., 2022 (Fig.2a)

Kim et al., 2021 (Fig.3c)

Avsec et al., 2021 (Fig.4c)

Figure R1.3 (Only this rebuttal): Examples of enhancer complexity from previous studies investigating enhancer code in cardiogenesis (Ameen et al., 2022), *Drosophila melanogaster* S2 cells (Almeida et al., 2022), epidermal differentiation (Kim et al., 2021), and HepG2 (Avsec et al., 2021).

Comparison of DeepMEL, ChromBPNet, and Enformer. To go beyond our own (previously published) models DeepMEL and DeepFlyBrain, we also compared our design strategy with the Enformer model, finding strong agreement in predicted activity scores along the optimization path from a random sequence to a cell type specific enhancer. The Enformer model contains four classes that correspond to melanocytes or melanocytic melanoma, similar to the cell state that we used for human enhancer design. We scored all our designed sequences, starting from a random sequence, at every mutational step with the Enformer model (Fig R1.4a,c). Enformer, for the DNase track “melanocyte” or “SK-MEL-5”, shows continuous increases in prediction scores along our evolutionary path just as in our DeepMEL2 model. Interestingly, when we score our enhancers for all 643 DNase classes, only the melanocyte and melanoma classes show a significant score, illustrating the high cell type specificity of our synthetic enhancers (Fig R1.4b).

Note that we also score, after our 15 mutations, a sequence in which we added additional ZEB repressor sites. This addition shows a strong decrease of the prediction score of the synthetic enhancers due to these repressor sites. This aspect is very well recapitulated by Enformer as well (Fig R1.4).

Figure R1.4 (Figure 6g,i,j): **a**, Enformer prediction scores for three classes calculated after each mutational step and after the addition of repressor sites for all MEL EFS sequences. **b**, Prediction scores for the top 50 DNase tracks for EFS-4 sequences. The four first DNase tracks are: foreskin melanocyte male newborn, SK-MEL-5, foreskin melanocyte male newborn, SK-MEL-5. **c**, Enformer prediction tracks for three classes for melanoma EFS-4 sequences added in place of the IRF4 enhancer. Top track: random sequence prediction score, other tracks: delta of mutated sequence prediction score vs random sequence prediction score.

We trained a ChromBPNet model on MM001 and another ChromBPNet model on MM047. ChromBPNet models bulk ATAC-seq data per cell type, correcting for Tn5 biases. The MM001 (MEL) model agrees very well with our MEL-design process (Fig R1.5). When adding ZEB sites to the functional enhancer, the ChromBPNet model confirms, like DeepMEL and Enformer models, that these represent repressor sites and can abolish enhancer activity (Fig R1.5). This reduction of enhancer activity is confirmed by in vitro experiments (see further).

To prove that the enhancers are not biased to the DeepMEL model, we show that two additional enhancer models with completely different frameworks (Enformer and ChromBPNet) confirm our DeepMEL predictions (R1.4, R1.5). Based on the fast uptake of our methods by the field, we believe that our new version of the manuscript, with new case studies, new models, and improved discussion provide the reader with both nuance and excitement of using our methodology as a roadmap to design enhancers for other cell types.

Figure R1.5 (panel a: Figure 6h, panel b: only this rebuttal): **a**, ChromBPNet ATAC MM001 (MEL) and ATAC MM047 (MES) prediction scores for each mutational step and after the addition of repressor sites for all EFS sequences. **b**, ChromBPNet ATAC MM001 prediction tracks for EFS-4 sequences. Top track: random sequence prediction score, other tracks: delta of mutated sequence prediction score vs random sequence prediction score.

Remark 1.2: Experimental validation of the synthesized Kenyon cell enhancers indicates rather inaccurate computational predictions (4/6 validation rate) and is limited in scope. It would be beneficial to increase the number of tested sequences from just a handful of arbitrary selected examples and to analyze the source of false positive predictions.

Answer 1.2: We followed the advice of the reviewer and tested 68 sequences in total by generating 43 additional transgenic lines. Using these new lines we performed further validation experiments, and examined in detail the source of false positives. These new results are incorporated throughout the revised manuscript and below we provide an overview of the new experiments related to Kenyon cell enhancers:

- a) We performed 5 additional mutation steps (in silico evolution) on two negative and two weak enhancers from the first manuscript version to examine if they now become positive (under the assumption that the in silico evolution process did not yet reach a high enough score (Fig R1.4, R1.5)). All four are now positive (Fig R1.6). This suggests that the negatives were indeed not yet optimized to their highest end point. Interestingly, the additional mutations that make these enhancers active affect the same activator motifs across positive enhancers (i.e., either creating stronger sites, or additional sites). In addition to the previous enhancer fly lines, we designed 7 new Kenyon cell synthetic enhancers (in silico evolution with 15 mutations) (Fig. R1.7).

Figure R1.6 (Top panels: Supplementary Figure 1i, bottom panels: Figure 2d): In vivo enhancer activity of cloned EFS sequences with no or weak activity after 10 (top) and acquired activity after 15 (bottom) mutations.

Figure R1.7 (panel a top and panel b: Supplementary Figure 1k,h; panel a bottom: Figure 2d): **a**, In vivo enhancer activity of negative (top panels, Enhancer-GFP co-labeling with Dachshund (DAC) to stain Kenyon cells) and positive (bottom panels, Enhancer-GFP) enhancer sequences after 15 mutations. **b**, Positive enhancer EFS-13 after 15 mutations: combined image of Enhancer-GFP with DAC (top panel). The expected location of KC is shown with dashed circles.

- b) For three of our positive enhancers for either *Drosophila* Kenyon cells or melanoma, we tested the mutational steps along the design path to investigate how many mutational steps are required to go from a random sequence to a Kenyon cell/melanoma specific enhancer. Starting from the random sequence, we tested the activity of the sequences after every 2 mutations. We did this for *Drosophila* enhancers in vivo (each sequence required a new transgenic line) (Fig R1.8a), and for human melanoma enhancers using luciferase assays (Fig R1.8b). All *drosophila* enhancers started to show activity after 8 mutations and luciferase measurements in the MM001 cell line showed a gradual increase in activity after 7 mutations and displayed very high correlation with DeepMEL2 predictions.

Figure R1.8 (panel a: Supplementary Figure 1n; panel b: Figure 6f and Supplementary figure 7f): **a**, In vivo enhancer activity for three KC EFS sequences after every two mutations. **b**, Luciferase activity levels in MM001 (left) and correlation with DeepMEL2 prediction score (right) for three MEL EFS sequences after incremental mutation steps.

- c) In the “near-enhancer” sequence category we had tested three different genomic enhancers and rescued their activity with only six mutations. Of these, 2/3 became positive, while one remained negative. We now performed an additional five mutations in this negative sequence (FP3), and tested it again in a new transgenic fly. It now became positive (Fig R1.9a). Thus all three ‘near enhancers’ have now been turned into an active enhancer. We also tested one additional “near-enhancer” sequence (FP1) and rescued it with 6 mutations (Fig R1.9b). Note that the name of the rescued sequences has been changed compared to the original manuscript (e.g. FP2 is now FP3).

Figure R1.9 (panel a and b bottom: Figure 2g; panel b top: Supplementary Figure 2j); **a**, In vivo enhancer activity of the FP3 sequence after 6 (top) or 11 (bottom) mutations. **b**, In vivo enhancer activity of the WT FP1 sequence (right) and after 6 mutations (left) to rescue KC activity.

From all these experiments, 4/6 glial synthetic enhancers (66%), all genome-rescued enhancers (100%), and 10 (11) out of 13 Kenyon cell enhancers worked (76-85%). KC enhancer number 10 is slightly positive but weak and a-specific so we consider it negative (Fig. R1.7). Two other KC enhancers (number 8, 9) we consider as negative. All three non-GFP expressing enhancers (8-10) were investigated further. We did verify whether the negative sequences did not have any repressor site generated in the junction of the enhancer and the vector sequence, or if mutations were present in the DNA synthesis or cloning, but this was not the case. Next, we double-stained these lines with anti-Dachsund antibody that labels KCs and observe a lesion pattern in the KC region of these three cases (R1.7a EFS8-10 vs R1.7b). This distinct lesion pattern (EFS 8-10) compared to the positive enhancer (e.g., EFS 13) can be observed in the fly brains (Supplementary Figure 1k vs. Supplementary Figure 1h). Based on the lesion we hypothesize a GFP misexpression causing a defect to the KCs. Due to the lesion pattern, these 3 KC enhancers are not 100% certain to be negatives, we have too few non-GFP expressing enhancers to identify possible sequence features that may cause them to be negative (or cell-lethal). Looking into the expression of these enhancers in wandering larvae did not lead to a clear conclusion either and we consider this potential developmental or cell-lethal defect out of the scope of the paper.

Remark 1.3: Given that the “evolutionary” design of the proposed model is driven by the assumption of the DeepFlyBrain scoring system being a reliable indicator of enhancer activity at each iteration of the enhancer design, this relatively low validation rate raises concerns of the applicability of the proposed “evolutionary trends” (repressor binding deactivation followed by acquisition of activator sites) to the natural evolution of enhancer sequences. In particular, the experimental validation of the repressor binding deactivation (which is proposed as the foundation of enhancer activation / preprogramming) appears necessary. Please experimentally demonstrate that a) the binding of repressor proteins is affected by the first round of mutations and b) repressor deactivation has an expected impact on the enhancer activity.

Answer 1.3: We agree that it is very interesting to further unravel the relationship between the generation of activator motifs and the destruction of repressor motifs along the in silico evolution process, and to provide additional mechanistic detail. Note that we do not argue that in silico evolution (by a greedy search) reflects the natural evolution of enhancers, we clarify this point better to the reader now.

Repressor motifs in the first round of mutations. We assessed at which steps during the mutational rounds that the repressor motifs are destroyed. We did this using motif discovery across the 6,000 sequences, at each step

during *in silico* evolution. Classical motif enrichment using PWMs for the activator and repressor motifs showed preferential destruction of repressor motifs during the first rounds, both for *Drosophila* and human enhancers (Fig R1.10). In addition, DeepExplainer applied to all sequences combined with TF-MoDISco, again at each step, confirmed this finding.

Figure R1.10 (Panel a: Supplementary Figure 1g; panel b: Supplementary Figure 7c): Delta number of motifs in each mutational step for KCs (a) and for MEL (b). The TF-Modisco patterns are shown at the top of each plot. The patterns that are upside-down are the ones contributing negatively to the model's prediction and they are destroyed by the model on each step.

Impact of repressor sites on enhancer activity. To assess whether the destruction of repressor motifs have the expected impact on enhancer activity we performed new *in vivo* enhancer-reporter assays. Specifically, we introduced repressor binding sites, each created by a single or double mutation, without interfering with activator binding sites of designed and active enhancers.

We introduced repressor sites to the following enhancers, and tested them experimentally:

- Three Kenyon cell enhancers that were designed (and positive) by *in silico* evolution; the introduction of repressor sites causes the complete loss of activity (Fig R1.11a).
- Three near-enhancer sequences that were rescued by *in silico* evolution; the introduction of repressor sites causes the complete loss of activity (Fig R1.11a).
- One Kenyon cell enhancer that was created (and was positive) by motif embedding; the introduction of repressor sites causes the complete loss of activity (Fig R1.11b).
- A genomic melanoma enhancer, namely the IRF4 enhancer; the wild type enhancer is only moderately active but the introduction of repressor sites causes the complete loss of activity (Fig R1.11c).
- Three human melanoma enhancers designed by *in silico* evolution; the introduction of repressor sites causes the complete loss of activity (Fig R1.11d).

Thus, in every case, the activity of the enhancer is lost because of the introduced mutations that generate repressor sites. These experiments show that having activators on the correct location, distance, orientation, and combination is not enough to drive the expression, because when repressor binding sites are present in between the activator sites, or flanking the activator sites, there is still no activity. In conclusion, we confirmed that the repressor motifs have the expected impact on enhancer activity.

We also performed the reciprocal experiment, namely to destroy a repressor site in the IRF4 enhancer. The addition of ZEB sites abolished activity, while mutating the WT ZEB sites caused a significant increase in enhancer activity (Fig R1.11c).

Figure R1.11 (panel **a**: Figure 3b; panel **b**: Figure 4f,g; panel **c**: Figure 5d; panel **d**: Figure 6f and Supplementary Figure 7f): **a**, In vivo enhancer activity of the EFS-4 sequence before and after generation of repressive sites (top panels) and of five other originally active synthetic enhancers in which repressor sites were generated. **b**, In vivo enhancer activity of a KC enhancer generated by motif embedding before and after generation of repressive sites. **c**, Luciferase activity levels in MM001 of the WT IRF4 enhancer and mutational variants. **d**, Luciferase activity levels in MM001 of three MEL EFS sequences with incremental mutation steps and addition of repressor sites.

Repressor code is not a model artifact. To further confirm that the repressor code is not an artifact of our deep learning models, we compared DeepMEL predictions with Enformer predictions. The Enformer model contains classes that represent our MEL state, namely normal melanocytes and the SKMel5 melanoma cell line, which is also a MEL-state cell line. Experiments with Enformer confirm that when we introduce a repressor site into an active enhancer, the chromatin accessibility, H3K27ac, and the target gene expression is decreased (Fig R1.4). Note that also the predicted ChIP-seq signal for ZEB2 (in HEK293 cells) is increased according to Enformer (Fig. R1.12).

Figure R1.12 (on this rebuttal): Enformer prediction tracks for the ZEB2 ChIP-seq class for the IRF4 enhancer (red). Top track: WT sequence prediction score, other tracks: delta of mutated sequence prediction score vs WT sequence prediction score.

The prediction of H3K27Ac and gene expression by the Enformer model, using our MEL cell state, is a powerful approach to compare genomic enhancers with synthetic enhancers, and to also assess the role of the repressor and activator sites in synthetic enhancers. We replaced the IRF4 MEL enhancer with our synthetic MEL enhancers and observed that the predicted decrease of IRF4 gene expression, by deleting the enhancer, could be rescued by our synthetic enhancers (Fig R1.1.3c). This rescue could then be canceled again by introducing repressor sites.

We also trained a ChromBPNet model (Brennan et al., 2022) on several of our human melanoma cell lines ATAC-seq data. The ChromBPNet model trained on MEL lines (such as MM001) showed exactly the same activators and repressor and their effect on chromatin accessibility as a further validation (Fig R1.5 and Fig R1.13).

Figure R1.13 (Supplementary Figure 6a): Nucleotide contribution scores for the IRF4 enhancer (first track), delta prediction score for in silico saturation mutagenesis with DeepMEL2 (second track), delta log₂FC for in vitro saturation mutagenesis for SKMEL-28 (third track) and delta prediction score for in silico saturation mutagenesis with ChromBPNet MM001 ATAC model (fourth track).

Identification of ZEB2 as candidate repressor and confirmation of ZEB2 binding by ChIP-seq. An important addition in the revision is that we now identified the transcriptional repressor that binds to the candidate ZEB binding sites. Using RNA-seq data from our earlier work (Wouters et al., 2020) we identified ZEB2 as the most likely candidate. We performed ChIP-seq against ZEB2 in human melanoma cells (MM001, MEL state). The top 3,000 ChIP-seq peaks are enriched for the ZEB1/2 motif, confirming the quality of the ChIP-seq data. Using this new data set, we found that genomic regions with ZEB2 binding sites are less accessible (as our model predicts), and genomic

regions with multiple ZEB sites are completely closed (Fig. R1.14). This suggests that the ZEB2 repressor TF, when bound, prevents activator TFs from binding on the enhancers.

Figure R1.14 (panels **a** and **b**: Figure 5f,g; panel **c**: Supplementary Figure 6c): **a**, ZEB2 ChIP-seq signal (x-axis), SOX10 ChIP-seq signal (y-axis), and ATAC-seq signal of the top 3,000 ZEB2 ChIP-seq regions in MM001. High ChIP-seq peaks for ZEB2 (x-axis) are not accessible; high SOX10 binding (y-axis) can overcome weak ZEB2 binding, as these regions are accessible. High SOX10 and low ZEB2 binding are most accessible. **b**, ATAC-seq (top) and ZEB2 ChIP-seq (bottom) tracks in MM001 highlight the chr16:90027296-90037796 region (red bar). **c**, Nucleotide contribution score (top) and in silico saturation mutagenesis (bottom) of ZEB2 ChIP-seq high region (chr16:90027296-90037796) for the MEL class.

Interestingly, from these ChIP-seq data we identified a peak for ZEB2 on the IRF4 enhancer, an enhancer with a clear predicted ZEB site, overlapping with the ChIP-seq peak for the activator SOX10 (Fig. R1.15a). This balance between activator and repressor binding may be the cause of the relatively low activity of this enhancer in a luciferase assay. Indeed, the DeepMEL model predicts that the destruction of this ZEB site increases the activity. We validated this by mutating the ZEB2 site in this enhancer, and observed a strong increase in luciferase activity (3-fold). Reciprocally, multiple nucleotides in this enhancer are predicted to decrease enhancer activity when mutated through the creation of a ZEB site (Fig. R1.15b,c,d). We also validated this prediction via a luciferase assay and observed a further decrease of enhancer activity when multiple ZEB sites are present.

Figure R1.15 (Figure 5a,b,c,d,e): **a**, MM001 ATAC-seq (top), SOX10 ChIP-seq (middle), and ZEB2 ChIP-seq (bottom) tracks of the IRF4 gene. The IRF4 enhancer location is highlighted in red. **b**, Luciferase activity levels in MM001 of the WT IRF4 enhancer and mutational variantes. **c**, Nucleotide contribution score (top) and in silico saturation mutagenesis (middle) of the IRF4 enhancer for DeepMEL2 human melanocyte-like melanoma class. In vitro saturation mutagenesis in SK-MEL-28 (bottom). Mutations resulting in a drop of both prediction and activity, and that generate a ZEB2 binding site are highlighted with a black circle. **d**, Scatter plot comparing in vitro (x-axis) and in silico (y-axis) saturation mutagenesis values of the IRF4 enhancer. **e**, Nucleotide contribution scores of newly generated ZEB2 motifs from mutations highlighted in panel **c**.

ATAC-seq on integrated synthetic enhancers: accessibility decreases due to repressor sites. To assess whether ZEB2 is also bound to synthetic enhancers with ZEB repressor sites, which cannot be tested on episomal constructs, we randomly integrated sequences into the MM001 genome using lentiviral vectors. We inserted random (negative control) sequences, synthetic enhancers that are active by luciferase, and the same synthetic enhancers with additional ZEB sites inserted. We then performed ATAC-seq on these lines (Fig. R1.16). This shows that random sequences are not accessible, enhancers with only activator sites are accessible, and the enhancers with both activators and repressor sites are not accessible. This experiment further confirms that synthetic enhancers behave like genomic enhancers.

Figure R1.16 (Figure 6e): MM001 ATAC-seq profile of 3 integrated lentiviral EFS reporters in their initial random form (top), after in silico evolution (middle) and after generation of repressive sites in the in silico evolved sequence (bottom). Red dashed lines indicate boundaries of the enhancer.

Remark 1.4: The minimal number of mutations needed for cell-type specific enhancer formation is discussed in detail for *Drosophila* and human enhancers. However, this metric is based on a sequential selection of top DeepFlyBrain scoring nucleotide changes. The authors should demonstrate that the same number of multiple lower-scoring changes are not capable of achieving a stronger impact on the enhancer activity. While there might be a limit to combinatorial testing of all possible mutations, it should be feasible to quantify all possible 5-6 mutation combinations using a computational approach and to compare it to the same number of sequential top-scoring mutations. Also, detailing the impact of individual sequential mutations on the gradual change of enhancer function would strongly benefit from experimental annotation of enhancer activity at every step of enhancer reprogramming.

Answer 1.4: In the revised manuscript, we have added additional analyses regarding mutation selections. When we choose randomly selected mutations (random drift) instead of the mutation with the top delta score (directed sequence evolution), the prediction score after many mutations does not change and stays close to zero (Fig. R1.17).

Figure R1.17 (Figure 2a and Figure 6a): **a**, Prediction score distribution of the sequences for the γ -KC class ($n = 6,000$) after each mutation. The box plots show the median (center line), interquartile range (box limits), and 5th and 95th percentile range (whiskers) for KC directed mutations (blue) or random drift mutations (orange). **b**, Prediction score distribution of the sequences for the MEL class ($n = 4,000$) after each mutation. The box plots show the median (center line), interquartile range (box limits), and 5th and 95th percentile range (whiskers) for KC directed mutations (blue) or random drift mutations (orange).

Changing the optimization algorithm. Quantifying all possible 5-6 mutation combinations as suggested by the reviewer is not possible with the computational power and memory we have. In each mutational step we generate 1500 sequences with a single mutation (500bp x 3 possible nucleotides to mutate), which takes 1 second to generate the sequences and calculate the prediction scores. If we would explore every possible direction for 5 steps, this would require around $7.5e+15$ sequences/calculations and it would take around 160.000 years to calculate (only for just one random sequence), not mentioning the required memory. So the search space even for a single random sequence is enormous. This further demonstrates that our state space search is efficient: starting from any location in this sequence search space (random sequence), by just 10 mutations, we can find the path that converts the random sequence into a functional enhancer in just 10 seconds. We consider this de facto enhancer design.

Nevertheless, we find the reviewer's comment interesting to explore - in the context of alternative state space searches compared to our greedy search. How does the enhancer prediction score behave when following different paths? Instead of choosing the best mutation in each step, what would happen if we go for the second or third best mutation? To test this, starting from a random sequence, we chose the top 20 mutations in each step and followed the path for 5 mutations. This took a relatively shorter time (less than a day) and starting from 1 random sequence, we obtained 3.2 million paths. This analysis showed that there are different paths that create binding sites at different locations. In some of the paths, the prediction score was higher compared to the path where we only chose the top scoring mutation (Fig. R1.18). This illustrates further that our evolutionary process determines a path in the state space towards a local optimum, and it is interesting for the reader to think about alternative state space searches.

Figure R1.18 (Supplementary Figure 2,a,b,c,d): **a**, Prediction score distribution for 3 million sequences. Blue line represents the path that was taken by the greedy algorithm. **b**, Zoomed-in version of panel **a** to the sequences that have higher prediction score than 0.25. This shows that there can be sequences scoring higher than the original path when the best mutation is not selected on each step. **c**, Prediction score of evolved sequences by greedy algorithm (EFS-4) vs the best of 3 million sequences on each mutational step. **d**, Comparison of the selected mutations. Both paths create an Ey binding site. Greedy algorithm creates Onecut binding site at the downstream of Ey, while the top-scoring path creates Mef2 binding site, which increases the prediction score further compared to the EFS-4 path.

Experimental assessment of different evolutionary steps. The second question of the reviewer concerns the experimental assessment of enhancer activity at different steps during the optimization process. To address this we have experimentally tested the activity of several enhancer designs at multiple mutational steps:

- Using three of our KC positive synthetic enhancers, we tested the activity of the sequence *in vivo*, for every two mutations including the initial random sequence (0M, 2M, 4M, 6M, 8M). We, thus, generated 15 new transgenic flies for this experiment. As seen in Figure R1.8a, each of these three KC enhancers becomes positive at mutational step 8. In conclusion, our approach succeeded in transforming a random sequence into a functional KC enhancer in 8 mutational steps.
- Using 3 of our MEL positive synthetic enhancers, we performed similar experiments testing every other mutational step by performing luciferase assay on 21 new constructs (Fig. R1.8b). Luciferase measurements in the MM001 cell line showed a gradual increase in activity after 7 mutations and displayed very high correlation with DeepMEL2 predictions.
- Using Enformer and training a ChromBPNet model to predict the activity and accessibility of the evolving sequence at each step *in silico*: both alternative models are in very strong agreement with DeepMEL, in terms of increases in prediction scores along the evolutionary process (Fig. R1.4, Fig. R1.5).

Remark 1.5: The paragraph starting on line 128 is confusing. If the premise of this method is based on “high prediction scores”, why are the selected elements with the “high prediction scores” not acting as enhancers and require additional mutations to turn into enhancers?

Answer 1.5: This shows that the model is not 100% accurate; a minority of genomic regions have a high prediction score despite showing no, or low, chromatin accessibility signal. Note that these ‘high prediction scores’ are lower compared to the true positive regions (the highest prediction scores are all accessible) (Fig. R1.19). We exploited these false positive predictions to identify near-enhancer sequences, and could show that with relatively few mutations, these regions can become active enhancers.

We updated the text to make it more clear as well as the discussion to mention about the pitfalls of our (or any) deep learning models.

Figure R1.19 (Supplementary Figure 2g): Comparison between γ -KC prediction score and mean γ -KC accessibility for the binned fly genome regions. The selected regions with high prediction and low accessibility are highlighted with blue, orange, green, and red dots.

Additionally, if the selected regions have high prediction scores, then do the six mutations correspond to positions that impart repression through chromatin inaccessibility?

Indeed, in one case, the mutations destroy the flanking repressor binding sites and also create activator sites. We now highlight and annotate the effect of each mutation on the figures (Fig. R1.20).

Figure R1.20 (Figure 2h): Nucleotide contribution scores of a selected genomic sequence in its initial form (top) and after 6 iterations (bottom). The position of the mutations are shown with dashed lines. The mutational order is written in between top and bottom plots together with the type of nucleotide substitutions.

Remark 1.6: Line 170. Would it be correct to assume that the ‘second cell type’ is KC? Did the augmented enhancer preserve its activity T1 intact? Mutations often coevolve; testing the original activity of augmented enhancer in its original cell type (T1) would be essential to corroborate the statement made in lines 157-159. In addition, though probably evident, expanding the legends of figure 3 (panels b, d and f) to include cell types would provide clarity.

Answer 1.6: Thank you for bringing this to our attention. It is correct that the second cell type is KC. To resolve the question around T1: we have done several experiments to cross in driver lines for the Tm1 and T1 cell types, but we did not achieve unambiguous results. We therefore decided to consider this as a semi-failed experiment, and we write in the text that the KC cell type was successfully added to the T1 enhancer, but that the T1 cell type may have been altered. The additional double enhancer we generated (amon) showed correct T4 expression before and after introduction of the mutations.

Captions of Figure 3 and Supplementary Figure 3 have been expanded to clarify cell type names.

Remark 1.7: Is it always necessary that mutations that abrogate the activity of an enhancer in a cell type will do so necessarily by creating a repressor binding site? To this end, it would also be pertinent to highlight the T1 specific repressors for which binding sites were generated (lines 180-181).

Answer 1.7: We show that transcription factor binding sites are the main driver of enhancer activity. The mutations that decrease the activity of enhancers do not only create repressor binding sites, but they can also destroy or decrease the affinity of activator binding sites.

Now we highlight which binding sites are destroyed and which ones are created that correspond to transcription factors expressed in the T1 cell type (Fig. R1.21).

Figure R1.21 (Supplementary Figure 3g): Nucleotide contribution scores of Pkc53e WT sequence and after 9 mutations for T2 (top), T1 (middle) and γ -KC (bottom). The position of the mutations are shown with dashed lines. The mutational order is written in-between top and bottom plots.

To further demonstrate the mutations that decrease the activity of an enhancer (either by destroying activators or creating repressors), we compared *in silico* and *in vitro* saturation mutagenesis of the IRF4 enhancer. We highlighted mutations resulting in the generation of a repressor site that were both predicted by DeepMEL2 and measured *in vitro* (Fig. R1.15b).

Remark 1.8: In section ‘Enhancer design by motif implantation’, it would be more informative to provide an assessment of occurrence and spatial arrangement of various activator binding sites in *Drosophila* cell lines (and similarly for human cell lines) or cite relevant literature.

Answer 1.8: We thank the reviewer for the suggestion. Note that we do not use *Drosophila* cell lines in our work, we only study enhancers *in vivo* in the *Drosophila* brain.

Investigating the occurrence and spatial arrangement of motifs in genomic enhancers was performed extensively in our previous DeepFlyBrain and DeepMEL/2 papers, so we had not included them here. Now we cite these papers and also performed additional analysis of motifs in genomic enhancers and compared them with the synthetic enhancers (Fig. R1.22).

Figure R1.22 (Supplementary Figure 1f and Supplementary Figure 7b): Average number of motif hits at each mutational step compared to Genomic enhancers for KCs (a) and for MEL (b).

Remark 1.9: Line 409, if I understand correctly, “for example” should be replaced with “furthermore/in addition”. In the same paragraph, the discussion of close-to-zero scores of random sequences does not necessarily imply that they contain multiple repressor binding sites (unless the authors illustrate the repressors and their binding site motifs). Further, to correctly understand the trajectory of sequence evolution, it would be profoundly more useful to assess the binding of these repressors (as well as activators) after each mutation to comprehensively understand the evolution of wild type enhancers. Such an analysis would also provide an understanding of 1) the correlation between DeepFlyBrain prediction scores/DeepExplainer contribution scores and in-vivo changes in enhancer activity w.r.t. binding sites, chromatin accessibility, etc., 2) co-evolving sites, if any, and 3) interplay of various players in the enhancer region, such as protein binding sites, contextual sites, chromatin accessibility defining sites, etc. It would also address the key motivation behind this work, i.e., providing “a thorough understanding of how enhancer activation is encoded in its DNA sequence” (line 54).

Answer 1.9: We agree with the reviewer that close-to-zero scores of random sequences does not always mean that they contain multiple repressor sites. It could also be that they do not have any activator sites. We clarified this in the text.

To assess the binding of these repressors and activators after each mutation, as suggested by the reviewer, we now performed multiple additional analyses.

- a) We illustrate the evolution of sequences by showing intermediate mutational steps, with annotated activator and repressor sites combined with the activity measurements.

Figure R1.23 (Supplementary Figure 8): Nucleotide contribution scores of EFS-4 at different mutational steps; 0 (random sequence), 3, 4, 7, 8, 12, 15, 15+Repressors.

- b) We experimentally test both fly enhancers in vivo (a transgenic line for each test) and human enhancers in vitro (luciferase assays), at multiple steps during the in silico evolution. As seen in Figure R1.8a, each of these three Kenyon cell enhancers becomes positive at mutational step 8. In conclusion, the threshold of transforming a random sequence into a functional Kenyon cell enhancers is set at 8 mutational steps.
- c) We performed ChIP-seq against the key repressor of the MEL enhancers in human, namely ZEB2 and provide important evidence on the balance between activator and repressor motifs (Fig. R1.14). In our view, this is a crucial addition to the revision that further explains the changes in activity upon creating or destroying a repressor site during the design process. Combined with ATAC-seq on these genome-integrated enhancers, we now provide multiple lines of evidence that our synthetic enhancers behave in the same (predictable) way as genomic enhancers.

- d) We assess the evolution of sequences, at intermediate mutational steps, using two alternative deep learning models, namely Enformer and a newly trained ChromBPNet model (Fig. R1.4 and Fig. R1.5). These models also recapitulate the genomic *and* synthetic enhancer logic, along the design trajectory.
- e) We extend our model predictions from chromatin accessibility to include also predictions of TF binding (ChIP-seq), predictions of histone modifications and predictions of gene expression, making use of the Enformer model (Fig. R1.4). These analyses show that synthetic enhancers can rescue genomic enhancers in the Enformer model.
- f) We could abolish the activity of our newly created enhancers simply by introducing mutations that create repressor sites, but without touching activators binding sites. We experimentally tested the activity of these sequences, together with their accessibility using ATAC-seq, and with model predictions (DeepMEL/2, DeepFlyBrain, ChromBPNet, Enformer). For each enhancer (genomic or synthetic) that is active, it can be destroyed relatively easily (both in predictions and experimentally) by adding repressor sites in between the activator sites.

All these experiments added many extra layers to provide a thorough understanding of how enhancer activation and repression is encoded in an enhancer's DNA sequence.

Remark 1.10: The foremost logic provided in this manuscript behind the emergence of enhancer sites in random sequences is the conversion of repressor to activator binding sites. However, the approach used to label these sites in DNA fragments is not provided/cited anywhere in text, for e.g., in lines 110, 116, figures 2, S1. It will be useful to cite appropriate references and provide a brief description in Methods. Alongside, repressors/activators, if known, associated with sites may also be mentioned.

Answer 1.10: We updated the figures, texts, and methods (PWM and TF-Modisco based) with the labeling of these activator and repressor sites together with the literature about the factors we identified as repressors in this study. We describe the approach to label sites in the Methods.

Note that enhancer emergence is not always the result of the conversion of repressors to activators. It can also be achieved independently by creating an activator site in a sequence without a repressor site, or by destroying a repressor without converting it to an activator. Nevertheless, because of the greedy algorithm, the mutations that destroy repressors and create activators at the same time are the ones that are often selected by the algorithm since they increase the prediction score more compared to a mutation that only destroys a repressor or only creates an activator site. This is one of the interesting observations in our study, but in our view not 'the foremost logic'. We consider this finding as important as our other novelties, namely the process of AI-driven sequence design, the combination of two cell type codes in a single enhancer; and the design of minimal enhancers.

Minor concerns:

1. Inconsistent usage of terms. T1 class, T-neurons, T1-neurons, all refer to the same cell type. Similarly, γ -KC, gamma-KC, KC cells, KC class, and so on.
2. References 20 and 23 should be quoted on line 59 for enhancer mutational analysis.
3. Several phrases such as 'KC class', 'activity pattern', should be defined at the first mention.
4. Line 106 "we investigated the effect of the initial (random) sequence," is incomplete. Effect on what?
5. Quantify the word 'usually' in line 116.
6. Expand the abbreviation EFS
7. In line 271, how was the strength of binding sites quantified?
8. Line 272, "that were predicted as KC enhancers..".
9. Line 412, cite reference/provide data.

All these minor concerns have been addressed in the text. Particularly, for point 5 we now show a quantification of the repressor sites present in the initial sequence which was added as Supplementary Figure 1f (Fig. R1.22a).

Referee #2 (Remarks to the Author):

This manuscript entitled "cell type directed design of synthetic enhancers" by Taskiran et al presents 3 methods (previously established by other labs) that can aid in the design of functional enhancer sequences. They demonstrate the 3 methods using DeepFlyBrain and DeepMel (both of which have been established in previous papers by the corresponding author's lab). A small subset of predictions are validated experimentally. It is a solid demonstration that adds to the growing literature of the feasibility of using neural network predictions as a way to guide the design of functional sequences.

The strength of this paper is the validation with experiments. Nevertheless, as the manuscript currently stands, it has many more weaknesses that should be addressed in terms of clarity and depth of analysis to become a more impactful paper.

General concerns:

Remark 2.1: Credit is not given clearly. For each method, the authors should do a better job of giving credit as they have all been previously established. For instance, in silico evolution (Vaishnav et al. Nature, 2022), motif implantation (Koo et al. PLoS Comput Biol, 2021; Avsec et al, Nat Genetic, 2021), and GANs (Killoran et al, arXiv, 2017; Gupta and Zou, Nat Mach Intell, 2019). In addition, the authors should provide some additional context of what differences exist (if any) with the proposed method and established methods.

Answer 2.1: Thank you for these suggestions. In the revised version, we expand the literature review and explain differences compared to our enhancer design. We clarify better that we do not claim to have invented the underlying methodology of in silico mutagenesis, motif embedding or GAN *per se*. Rather, we implemented and combined these methods with deep learning enhancer models to design synthetic Drosophila and human enhancers that are cell type specific, we experimentally test dozens of synthetic enhancers in vivo, and then exploit the design process to unravel enhancer logic, with novel findings, such as those related to activator and repressor motifs.

- In silico evolution: the reference to Vaishnav was already cited in our original version. It was used to modulate the activity of yeast promoters from native regulatory DNA, while we expanded this to create human and Drosophila enhancers from random scratch that are cell type specific, and to experimentally test their activity in vivo, which is novel. Using this approach in combination and comparison with other design methods is also new. Furthermore, our work is original in demonstrating how to scrutinise the logic of enhancers by following the 'path' of enhancer design. Importantly, we are also the first to use in silico evolution to prune enhancers (remove a cell type), to augment enhancers (add a cell type), and to rescue near-enhancers in the genome. We are also first in using in silico evolution to destroy a functional enhancer by generating repressor sites without disturbing the activator sites.
- Motif implantation: embedding motifs in sequences is a classical method that has been used for two decades to evaluate motif prediction methods (e.g., PWM scanning), or more recently sequence interpretation frameworks in the context of deep learning. Indeed, also Koo et al use motif embedding in the context of RNA sequences (RNA-protein interactions). Avsec et al. used in their BpNet paper motif embedding to test for the dependency of two motifs, as scored and interpreted by their BpNet model. We use motif embedding to design enhancers by adding one motif after the other, each time choosing the best embedding position as dictated by the model. Again, we are the first to use motif embedding to create synthetic enhancers that achieve in vivo success rates similar to the success rates of endogenous enhancers. *We now cite these suggested papers as examples where motif embedding was used before.*
- GANs: the use of GANs by Killoran and by Gupta and Zou focused on the methodological aspects but it was unknown whether such enhancers would actually be functional, or whether they would be cell type specific in vivo. To our knowledge, we are the first to show that.

We agree with the reviewer that these earlier studies are important to discuss and refer to. However, in our view our work is highly novel, because from these earlier studies, even though some of the concepts were laid out, it was not obvious or trivial to assume/deduct that enhancer CNNs could actually be used to efficiently design - and characterize in depth - functional enhancers that are active in a cell type specific manner and that work in vivo. It was certainly not

obvious to predict the success rate of correct activities for synthetically designed enhancers, given the low success rate in the pre-CNN era, and without testing a large number of enhancers experimentally and in vivo.

Remark 2.2: There is no mention of other approaches that have found success at designing regulatory genomic sequences. There are several other approaches that have designed functional genomic sequences, including input maximization (Bogard et al. Cell, 2019) and deep exploration network (Linder et al. Cell Systems, 2020), which is similar in spirit to GANs. Each has performed experimental validation as well.

Answer 2.2: In the revised version of our manuscript, we expand the literature review including the approaches that have performed designing regulatory genomic/synthetic sequences.

Remark 2.3: The biological sequence design literature is much more well established for protein sequences, with many compelling examples. Although not a strict requirement, it would be nice to see the connections (if any).

Answer 2.3: We thank the reviewer for this suggestion. It would indeed be interesting to investigate the connections between protein sequence design and our approaches. However, this is not the main focus of our study and would be more appropriate to a separate in-depth review.

Remark 2.4: The depth of analysis is rather shallow in many areas, especially with GANs. Many of the claims are not supported quantitatively.

Answer 2.4: We agree that the GAN-designed enhancers received fewer in depth follow up compared to in silico evolution and motif embedding. The reason for this is that the GAN design is more a black box model, from which it is more difficult to derive new rules. The insight we obtained from the other two strategies is far greater, and we could follow up many of these predictions with additional experimental validations, as outlined in our answers to the other questions. Indeed, in the revised manuscript we improved our manuscript substantially with much deeper analyses and insights.

Nevertheless, we find it important to include the GAN strategy as comparison, and focus for those enhancers on (1) the success rates; (2) how the GAN has learned the GC content bias; (3) how the prediction scores on GAN generated sequences change during each iteration by scoring them with DeepMEL and DeepFlyBrain; and (4) how motif composition of GAN generated sequences changes on each iteration compared to the genomic enhancers.

Remark 2.5: Lack of reproducibility. The code availability is an html file of jupyter notebooks that supply basic examples of code utility. The authors should supply a more thorough set of _usable_ codes -- both as a tutorial for other people that would like to use these computational tools and a more comprehensive set of code that reproduces the analysis (including figures) in this manuscript. Also, there is no way to run the code without the proper files, including the data (DeepFlyBrain_data.pkl) and models (json, hdf5). All data and code should be uploaded to zenodo (not just for partial GAN analysis).

Answer 2.5: Now we provide the models (json and hdf5), data (pkl files), intermediate files, as well as the code as notebooks in ipynb format to reproduce all the figures. The code is provided in Supplementary Code and the data can be requested here: <https://zenodo.org/record/8224621>

Remark 2.6: For each method, what are the characteristics of the sequences that did not yield significant expression experimentally? Why did they not work? Is it a poor regulatory code or poor sequence context?

Answer 2.6: This remark being very similar to Remark 1.2 from reviewer 1, we provide the same answer below: We followed the advice of the reviewer and tested 68 sequences in total by generating 43 additional transgenic lines. Using these new lines we performed further validation experiments, and examined in detail the source of false positives. These new results are incorporated throughout the revised manuscript and below we provide an overview of the new experiments related to Kenyon cell enhancers:

- d) We performed 5 additional mutation steps (in silico evolution) on two negative and two weak enhancers from the first manuscript version to examine if they now become positive (under the assumption that the in silico

evolution process did not yet reach a high enough score (Fig R1.4, R1.5). All four are now positive (Fig R1.6). This suggests that the negatives were indeed not yet optimized to their highest end point. Interestingly, the additional mutations that make these enhancers active affect the same activator motifs across positive enhancers (i.e., either creating stronger sites, or additional sites). In addition to the previous enhancer fly lines, we designed 7 new Kenyon cell synthetic enhancers (in silico evolution with 15 mutations) (Fig. R1.7).

Figure R1.6 (Top panels: Supplementary Figure 1i, bottom panels: Figure 2d): In vivo enhancer activity of cloned EFS sequences with no or weak activity after 10 (top) and acquired activity after 15 (bottom) mutations.

Figure R1.7 (panel a top and panel b: Supplementary Figure 1k,h; panel a bottom: Figure 2d): **a**, In vivo enhancer activity of negative (top panels, Enhancer-GFP co-labeling with Dachshund (DAC) to stain Kenyon cells) and positive (bottom panels, Enhancer-GFP) enhancer sequences after 15 mutations. **b**, Positive enhancer EFS-13 after 15 mutations: combined image of Enhancer-GFP with DAC (top panel). The expected location of KC is shown with dashed circles.

e) For three of our positive enhancers for either *Drosophila* Kenyon cells or melanoma, we tested the mutational steps along the design path to investigate how many mutational steps are required to go from a random sequence to a Kenyon cell/melanoma specific enhancer. Starting from the random sequence, we tested the activity of the sequences after every 2 mutations. We did this for *Drosophila* enhancers in vivo (each sequence required a new transgenic line) (Fig R1.8a), and for human melanoma enhancers using luciferase assays (Fig R1.8b). All *drosophila* enhancers started to show activity after 8 mutations and luciferase measurements in the MM001 cell line showed a gradual increase in activity after 7 mutations and displayed very high correlation with DeepMEL2 predictions.

Figure R1.8 (panel a: Supplementary Figure 1n; panel b: Figure 6f and Supplementary figure 7f): **a**, In vivo enhancer activity for three KC EFS sequences after every two mutations. **b**, Luciferase activity levels in MM001 (left) and correlation with DeepMEL2 prediction score (right) for three MEL EFS sequences after incremental mutation steps.

f) In the “near-enhancer” sequence category we had tested three different genomic enhancers and rescued their activity with only six mutations. Of these, 2/3 became positive, while one remained negative. We now performed an additional five mutations in this negative sequence (FP3), and tested it again in a new transgenic fly. It now became positive (Fig R1.9a). Thus all three ‘near enhancers’ have now been turned into an active enhancer. We also tested one additional “near-enhancer” sequence (FP1) and rescued it with 6 mutations (Fig R1.9b). Note that the name of the rescued sequences has been changed compared to the original manuscript (e.g. FP2 is now FP3).

Figure R1.9 (panel a and b bottom: Figure 2g; panel b top: Supplementary Figure 2j); **a**, In vivo enhancer activity of the FP3 sequence after 6 (top) or 11 (bottom) mutations. **b**, In vivo enhancer activity of the WT FP1 sequence (right) and after 6 mutations (left) to rescue KC activity.

From all these experiments, 4/6 glial synthetic enhancers (66%), all genome-rescued enhancers (100%), and 10 (11) out of 13 Kenyon cell enhancers worked (76-85%). KC enhancer number 10 is slightly positive but weak and a-specific so we consider it negative (Fig. R1.7). Two other KC enhancers (number 8, 9) we consider as negative. All three non-GFP expressing enhancers (8-10) were investigated further. We did verify whether the negative sequences did not have any repressor site generated in the junction of the enhancer and the vector sequence, or if mutations were present in the DNA synthesis or cloning, but this was not the case. Next, we double-stained these lines with anti-Dachsund antibody that labels KCs and observe a lesion pattern in the KC region of these three cases (R1.7a EFS8-10 vs R1.7b). This distinct lesion pattern (EFS 8-10) compared to the positive enhancer (e.g., EFS 13) can be observed in the fly brains (Supplementary Figure 1k vs. Supplementary Figure 1h). Based on the lesion we hypothesize a GFP misexpression causing a defect to the KCs. Due to the lesion pattern, these 3 KC enhancers are not 100% certain to be negatives, we have too few non-GFP expressing enhancers to identify possible sequence features that may cause them to be negative (or cell-lethal). Looking into the expression of these enhancers in wandering larvae did not lead to a clear conclusion either and we consider this potential developmental or cell-lethal defect out of the scope of the paper.

Remark 2.6: Thoughtful discussion of limitations. All of the predictions rely on deepflybrain or deepmel. Any biases they learn will be propagated to the enhancer design predictions. There is no discussion of the limitations of this model-based sequence design strategies.

Answer 2.5: We thank the reviewer for the suggestion. We also find it important to investigate whether our predictions are mere reflections of the model, or whether they represent generic and correct enhancer biology. Besides additional experimental validations, which confirm that our model predictions reflect bona fide regulatory logic, we now included additional, and independently trained deep learning models (with different architectures and different training data) in comparison to DeepMEL, namely Enformer and a newly trained ChromBPNet model. Enformer was trained by the authors on thousands of ENCODE tracks and did not include any of the data used for DeepMEL training. ChromBPNet models we trained ourselves on a single ATAC-seq data set of one of our samples (two separate models, one for MM001 and one for MM047). These models are thus not learning how to contrast MEL versus MES enhancers. Using these models we compared (interpreted) the motif content of DeepMEL-generated synthetic enhancers and found a very strong agreement in interpretation. In other words, Enformer and ChromBPNet predict, and interpret, the DeepMEL-based synthetic enhancers as genuine enhancers (Fig. R2.1 and Fig. R2.2). Both activator and repressor motifs could be confirmed in genomic enhancers, and in synthetic enhancers (Fig.

R2.3). We also used these alternative models to assess our synthetic design 'path' from random sequences to full-blown enhancers, and confirmed that both Enformer and ChromBPNNet prediction scores correlate with DeepMEL scores along the design path.

Hence, there are limitations to our approach, but importantly, we show that using alternative models with very different architectures, training data, and training regimes, we arrive at the same interpretation.

In addition, we added to the discussion, as suggested by the reviewer, the remaining limitations of our design strategies and of our models, including the fact that our models are limited to identifying repressor motifs that decrease chromatin; our limitation in explaining subtle variation in GFP output between synthetic enhancers (which may represent unseen sequence features); and our focus on enhancers for differentiated cell types (as compared to developmental enhancers).

Figure R2.1 (Figure 6g,i,j): **a**, Enformer prediction scores for three classes calculated after each mutational step and after the addition of repressor sites for all MEL EFS sequences. **b**, Prediction scores for the top 50 DNase tracks for EFS-4 sequences. The four first DNase tracks are: foreskin melanocyte male newborn, SK-MEL-5, foreskin melanocyte male newborn, SK-MEL-5. **c**, Enformer prediction tracks for three classes for EFS-4 sequences added in place of the IRF4 enhancer. Top track: random sequence prediction score, other tracks: delta of mutated sequence prediction score vs random sequence prediction score.

Figure R2.2 (panel a: Figure 6h, panel b: only this rebuttal): **a**, ChromBPNet ATAC MM001 (MEL) and ATAC MM047 (MES) prediction scores for each mutational step and after the addition of repressor sites for all EFS sequences. **b**, ChromBPNet ATAC MM001 prediction tracks for EFS-4 sequences. Top track: random sequence prediction score, other tracks: delta of mutated sequence prediction score vs random sequence prediction score.

Figure R2.3 (Supplementary Figure 6a): Nucleotide contribution scores for the IRF4 enhancer (first track), delta prediction score for in silico saturation mutagenesis with DeepMEL2 (second track), delta log₂FC for in vitro saturation mutagenesis for SKMEL-28 (third track) and delta prediction score for in silico saturation mutagenesis with ChromBPNet MM001 ATAC model (fourth track).

Concerns regarding enhancer design by in silico evolution:

Remark 2.8: The authors should cite in silico mutagenesis (Alipanahi et al. 2015; Zhou & Troyanskaya, 2015, Kelley et al. 2016) and in silico sequence evolution (Vaishnav et al. Nature, 2022). Also, is there any difference between what was done in Vaishnav and here. If any, it should be clarified. If not, it should be explicitly stated that that method was used here.

Answer 2.8: In silico mutagenesis is not the first time used by or invented by us, and our in silico evolution is similar to Vaishnav et al., 2022, where they used this method to design yeast promoters from already existing genomic sequences. Note that we introduce variations to this method by evolving sequences towards a second cell type, while keeping the score for the first cell type constant; evolving ‘away’ one cell type from a dual-code enhancer; or evolving a functional enhancer towards a repressed enhancer without touching the activator sites. We also investigated alternative state space searches besides the greedy search algorithm.

We updated the text and added the suggested citations.

Remark 2.9: Many of the stated claims are anecdotal. For instance, short repressor binding sites are present in random sequences and they are the targets of deleterious mutations during the early rounds of in silico evolution. However, the scale of the attribution maps suggest that these short repressors in random sequences are very weak.

Answer 2.9:

Short repressor motifs. The statement of short repressor binding sites in random sequences is an important finding, and we provide additional experiments and analyses in the revision that further support and strengthen this statement. These additional data include luciferase assays with random sequences (they are not active), with sequences at different steps along the design path (they become gradually active), and active synthetic enhancers where we added repressor sites (without touching the activator sites), which then become inactive (100% success rate) (Fig. R2.4). For these experiments we used both human enhancers in cell culture, and Drosophila enhancers for which we created new transgenic lines.

Figure R2.4 (panel **a**: Figure 3b; panel **b**: Figure 4f,g; panel **c**: Figure 5d; panel **d**: Figure 6f and Supplementary Figure 7f): **a**, In vivo enhancer activity of the EFS-4 sequence before and after generation of repressive sites (top panels) and of five other originally active synthetic enhancers in which repressor sites were generated. **b**, In vivo enhancer activity of a KC enhancer generated by motif embedding before and after generation of repressive sites. **c**, Luciferase activity levels in MM001 of the WT IRF4 enhancer and mutational variants. **d**, Luciferase activity levels in MM001 of three MEL EFS sequences with incremental mutation steps and addition of repressor sites.

Attribution scores of repressor motifs. The score in the attribution maps for the repressor sites seem weak because these scores are calculated based on the enhancer prediction score. Since the prediction scores for the random sequences are close to zero, the contribution scores on the attribution maps are inevitably very small. But this doesn't mean that these binding sites are weak. In the figure below we show an example of a genomic region that has many ZEB2 sites, which also has high ZEB2 ChIP signal and no accessibility (Fig. R2.5). When we look at the attribution map, since the prediction score is zero or close to zero, the contribution scores are also very small. Yet, our ChIP-seq data confirms that they are bound by ZEB2, and the ATAC-seq data confirms that the region is not accessible.

This comment is important for the reader though, and we clarify this better in the text.

Figure R2.5 (panels **a** and **b**: Figure 5f,g; panel **c**: Supplementary Figure 6c): **a**, ZEB2 ChIP-seq signal (x-axis), SOX10 ChIP-seq signal (y-axis), and ATAC-seq signal of the top 3,000 ZEB2 ChIP-seq regions in MM001. High ChIP-seq peaks for ZEB2 (x-axis) are not accessible; high SOX10 binding (y-axis) can overcome weak ZEB2 binding, as these regions are accessible. High SOX10 and low ZEB2 binding are most accessible. **b**, ATAC-seq (top) and ZEB2 ChIP-seq (bottom) tracks in MM001 highlight the chr16:90027296-90037796 region (red bar). **c**, Nucleotide contribution score (top) and in silico saturation mutagenesis (bottom) of ZEB2 ChIP-seq high region (chr16:90027296-90037796) for the MEL class.

We also have examples of ZEB2 binding sites with high attribution scores (Fig. R2.6). Since these regions have activator binding sites resulting in high prediction scores, now the relative attribution scores of ZEB2 sites are also high.

Figure R2.6 (only in rebuttal): DeepMEL2 (MEL class) nucleotide contribution score of one the designed sequences after applying the mutations to generate repressor sites at three different locations.

Remark 2.10: What defines a repressor binding site? Are these sites strong or weak repressor binding sites or partial binding sites? Perhaps a PWM track could clarify this.

Answer 2.10: Repressor binding sites are defined by their negative contribution to the prediction of chromatin accessibility and the presence of such sites decreases the prediction score, decreases chromatin accessibility, and decreases enhancer activity. Our revision experiments focused largely on these repressor sites, their balance with activator sites, and experimental validations to confirm these predictions. These results are shown throughout the

rebuttal and revised version. As suggested by the reviewer we also used PWM scoring, to annotate how weak (w) or strong (s) they are (Fig. R2.7). We thank the reviewer for this suggestion.

Figure R2.7 (panel a: Supplementary Figure 1g; panel b: Figure 2b): **a**, TF binding motifs were identified by using TF-Modisco. The repressor patterns are shown upside-down as they contribute negatively. **b**, Nucleotide contribution scores (DeepFlyBrain) together with the strong(s) / weak(w) annotations of the identified PWMs.

In earlier work we had identified Mamo as a repressor TF that binds to the repressor sites in KC enhancers (Janssens et al., 2022). In this work we now identify ZEB2 as a repressor that binds to melanoma enhancers and validate this finding experimentally by ZEB2 ChIP-seq. We also include many new enhancer design constructs with experimental testing, both in *Drosophila* and human, where we introduce repressor sites in working enhancers (not overlapping with activator sites), and each time they completely abolish the activity of the enhancer (Fig. R2.4). We furthermore use additional deep learning models (ChromBPNet and Enformer) that recapitulate the ZEB repressor site predictions (Fig. R2.1 and Fig. R2.2).

Remark 2.11: What if a strong repressor binding site were present?

- suggestion: to support any claims quantitatively, one could be to look at the frequency of mutations at repressor sites vs other sites early during the evolution process (across a population of test sequences).
- suggestion: One could implant strong repressor binding sites to see how the single nucleotide-based evolution cycles. This can be compared to weak repressor binding sites.

Answer 2.11: Now we provide the number of activators and repressors on each sequence during the evolution process. It shows that the repressor binding sites are destroyed by mutations while activator binding sites are generated. GATC and AAGA were destroyed more during the first round mutations for KCs, and it was ZEB2 for MEL.

Figure R2.8 (Panel a: Supplementary Figure 1g; panel b: Supplementary Figure 7c): Delta number of motifs in each mutational step for KCs (a) and for MEL (b). The TF-Modisco patterns are shown at the top of each plot. The patterns that are upside-down are the ones contributing negatively to the model's prediction and they are destroyed by the model on each step.

In order to understand the effect of repressor binding sites with regards to their position, distance to activators, number, and affinity level, we performed the following experiments:

- We created repressor binding sites on the fully evolved synthetic sequences (also on the IRF4 enhancer) by choosing single or double mutations in between activator binding sites without disrupting any of the activator binding sites (Fig. R2.9a,b). We observed that it is very easy to create repressors with a single mutation.
- By creating single repressor binding sites on different locations with single mutations, we saw that the effect of the repressor can vary based on its location relative to the activators binding sites.
- We created a weak binding site with a single mutation, it didn't decrease the activity of the enhancer. However, if we make it stronger with an additional mutation, the activity of the enhancer decreases.
- We added repressor binding sites one-by-one and it showed that increasing the number of repressor binding sites on an enhancer, decreases the activity of the enhancer further. After 5-6 repressor binding sites creation, the enhancer activity is completely gone.

Figure R2.9 (Supplementary Figure 10a,b,c,d,e): **a**, Prediction scores for each mutational step and after the addition of repressor sites for 3 EFS sequences. **b**, Nucleotide contribution scores (DeepMEL2 MEL class) showing the creation of single or multiple repressor binding sites by single or double mutations in the EFS-4 sequence. **c-e**, In vivo enhancer activity of EFS-4 (**c**), EFS-1 (**d**), and EFS-8 (**e**) after the generation of repressor binding sites.

Remark 2.12: The claim "this suggests that KC enhancers... can arise de novo in the genome with few mutations." is not strongly supported. The authors should emphasize the assumption that the region must be accessible in the first place.

Answer 2.12:

ATAC-seq on human integrated enhancers. We respectfully disagree with this interpretation. Our interpretation, and our computational and experimental data, strongly suggests that with an optimal constellation of TF binding sites, some of which include TFs that act as pioneer TFs or recruit chromatin modifiers, an enhancer can become accessible due to (cooperative) TF binding. This view is also commonly used in other studies (Shlyueva et al., 2014; Catarino et al., 2018). We now show evidence for this experimentally by performing ATAC-seq on integrated synthetic sequences in the genome: when a random sequence is integrated, it is not accessible, but when a synthetically evolved enhancer sequence is integrated, it becomes accessible (Fig. R2.10). Likewise, we reason that a near-enhancer in the genome, upon a few mutations, can become an active enhancer (and thus accessible), by recruiting the TF (and co-factor) complement.

Figure R2.10 (Figure 6e): MM001 ATAC-seq profile of 3 integrated lentiviral EFS reporters in their initial random form (top), after in silico evolution (middle) and after generation of repressive sites in the in silico evolved sequence (bottom). Red dashed lines indicate boundaries of the enhancer.

Near-enhancer sequences are not accessible in the genome. The regions we focused on are those that are particularly not accessible in Kenyon cells but have a high prediction score. To make this clearer, we now provide single-cell chromatin accessibility profiles (in addition to the scatter plot) of these regions *before* the mutations (Fig. R2.11).

Figure R2.11 (Supplementary Figure 2g,h): **a**, Comparison between γ -KC prediction score and mean γ -KC accessibility for the binned fly genome regions. The selected regions with high prediction and low accessibility are highlighted with blue, orange, green, and red dots. **b**, γ -KC ATAC-seq profile of the four selected regions. The exact location of the regions is indicated with dashed lines.

One additional region was rescued. We also provide another example for this rescue experiment, for a 4th region for which the genomic sequence is not active, but the rescued sequence becomes positive with 6 mutations. This makes 4 clear examples of this phenomenon (arising of de novo enhancers with few mutations) using 8 transgenic animals (Fig. R2.12).

Figure R2.12 (Figure 2g): In vivo enhancer activity of the WT FP1 sequence (top left) and after 6 mutations (top right) to rescue KC activity, and of the FP2 (bottom left), FP3 (bottom center) and FP4 (bottom right) after 6, 10 and 6 mutations respectively.

Synthetic fly enhancers are accessible in the genome. For the *Drosophila* enhancers, we integrate the synthetic enhancers in the genome. Hence, when we measured GFP activity in Kenyon Cells, we worked under the assumption that they are also accessible. We now experimentally verified whether this is the case by performing ATAC-seq on 6 transgenic lines for 6 different synthetic enhancers (Fig. R2.13). All 6 regions have effectively an ATAC-seq peak confirming their accessibility. Since we know exactly which mutations were created, and they are

linked to TF binding sites, these data strongly support our model and conclusion. Based on these additional experiments, we propose to maintain this claim, as we believe it is supported by our data.

Figure R2.13 (Supplementary Figure 1): *Drosophila* adult brain bulk-ATAC-seq profile of 6 transgenic flies that have the designed enhancers integrated. The chromatin accessibility profile of the integrated enhancers (left) and two control regions *gish* enhancer (middle) and *Appl* enhancer (right) are shown.

Remark 2.13: This model (i.e. DeepFlyBrain) does not understand temporal ordering of TF binding but rather a steady state assumption. Causation and correlation cannot be resolved by this (or similar sequence-function) model.

Answer 2.13: We agree that temporal ordering of TF binding is difficult to disentangle and we consider that question to be outside the scope of this work. We identify enhancer logic with optimal TF binding site combinations that work cooperatively. It may indeed be interesting in the future to perform kinetic and biochemical studies to investigate whether Eyeless needs to bind before Mef2, and so on. For synthetic enhancer design we achieve novel and highly useful results without taking the order of binding into account.

Note that we do have an order of binding site *creation* during the design process, but this order, in our view, is not necessarily related to the order of TF binding (Fig. R2.8b). Nevertheless, since we see in the first wave of binding site more appearances of SOX10 sites compared to MITF sites, we believe this is due to the fact that SOX10 binding sites have a greater impact on chromatin accessibility compared to MITF (this is also in agreement with our previous publication; Minnoye et al., 2020), one could argue that SOX10 binds first (and has higher influence to displace nucleosomes). Our data however does not allow us to make these conclusions directly, and to avoid speculation about hierarchy, we use the conservative interpretation of cooperativity.

Remark 2.14: There are other areas outside the annotation in Fig. 2f and 2j that have increased green intensity. What are these cell types? Do these also agree with predictions for other cell types?

Answer 2.14: In the time of first submission, we were thinking that these signals were coming from the vector and that the α -specific GFP signal in photoreceptor cells was caused by the minimal promoter, because we observed the same signal for all genomic enhancers as well as for negative control sequences (e.g., random sequences). During the revision we realized through additional experiments that there is a bleed-through from another confocal excitement range (555) while the signal we were interested in is in GFP channel (488). The linker in our vector construct contained residues of mini-white which we use to identify the flies with inserted mutation. This linker, however, causes a 555 excitability that is present in all flies that have an integrated vector in their genome (i.e., enhancer flies). We therefore repeated all our confocal imaging, and we restricted the wavelength 410-545 to get the

signal only from GFP range. Now all the background signal is gone and our images now clearly show that our designed enhancers are highly specific to Kenyon Cells.

Remark 2.15: From the attribution maps, there seems to be a positional bias of motifs to be near the center of the sequence. If trained on peak centered data, then models can be biased to focus on motifs near center of sequence. In this scenario, models may then have blind spots at the 3' and 5' flanks. If important motifs or context were to be present here, they may be ignored by the model.

- suggestions: plot the positional distribution of (positive) motifs that increase function at the end.

- suggestion: An interventional experiment can also be performed where the motif positions are shifted to be closer to the 3' or 5' flanks. If the model predictions do not change much, then the model is robust to translational shifts and there likely is not a learned bias by the original DL model. If the predictions change significantly, then it could be that there is a positional bias or that important sequence context that was not captured by the attribution map is now missing.

Answer 2.15: From our earlier papers on DeepMEL and DeepFlyBrain, as well as in publications from other labs (e.g., DeepSTARR, BPNNet, ChromBPNNet, PRINT, ...) we found that enhancer logic is usually encoded in relatively short sequences of ~200-250bp similar to the distance between the farthest mutations generated by our in silico evolution strategy (Fig. R2.14a). Nevertheless, our models use 500bp input sequence, and TF binding sites are enriched within the central nucleosome that is under the ATAC-seq peak summit. Therefore, motif occurrences are enriched within the center ~250bp. This 'bias' is thus representing biology, especially since our random sequences are GC-biased similarly to accessible genomic regions. We confirmed that our model is not biased by comparing DeepExplainer predictions to two other models, namely ChromBPNNet and Enformer (Fig. R2.1 and Fig. R2.2). These two alternative models use a different receptive field. Both on genomic enhancers as well as synthetic enhancers we found high agreement with DeepMEL/DeepFlyBrain models. We conclude that our models are not technically, but rather biologically biased, and that this is expected behaviour.

Also note that, when we simulate the incorporation of a 200bp enhancer in a 500bp sequence space by choosing random locations, it will occur more towards the center, this is statistically inevitable as can be seen on the simulation figure (Fig. R2.14b). When we check the location of selected mutations during the in silico evolution (Fig. R2.14c), or the location of implanted binding sites (Fig. R2.14d), it shows a very similar profile to the simulation we performed (Fig. R2.14b).

Figure R2.14 (panels a and c: Supplementary Figure 1d,e; panel b and d: only this rebuttal: a, Distribution of distances ($n = 6,000$) between farthest mutations on each sequence after 10 iterative mutations. The orange line

shows the median. **b**, Simulation of 200 bp space selection to create enhancer on 500 bp sequence space ($n = 1,000$) After selecting random 200 bp locations, the sum of selected locations are shown on the y-axis. **c**, Location of the generated mutations across the random sequences when performing in silico evolution ($n = 6,000$ sequences). **d**, Distribution of locations of Ey, Mef2, Onecut, and Sr after motif implanting experiments ($n = 2,000$ sequences).

Remark 2.16: Among the final evolved sequences, how many motifs are present? What are the frequencies observed of each motif type? Do they have a specific order/grammar?

Answer 2.16: In the revised manuscript, we now perform comprehensive motif analysis by using both PWMs and TF-Modisco based annotations (Fig. R2.8), we plot the number of sites in the final evolved sequences (and in the intermediate steps), and highlight the identified binding sites on the figures.

- *Variation of Cbust based annotation by using TF-Modisco patterns over each mutational step of in silico evolution is now displayed in Supplementary Figure 1g (Drosophila) and Supplementary Figure 7c (human).*
- *Plots of the number of binding sites through the in silico evolution process has been added in Supplementary Figure 1f (Drosophila) and Supplementary Figure 7b (human) and for GAN generated sequences in Supplementary Figure 7b (Drosophila) and Supplementary Figure 13b (human).*
- *All DeepExplainer nucleotide contribution tracks in main figures have been annotated with binding site names We provide easy-to-use notebooks where readers can easily check which sequences have which binding sites at which mutational steps and how strong the binding sites are.*

Remark 2.17: For the attribution plots, how were motifs annotated on attribution plots? How was a repressor site identified? What are the motifs of repressors?

Answer 2.17: Now we perform comprehensive motif analysis by using both PWMs and TF-Modisco based annotations and highlight the identified binding sites on the figures.

Concerns regarding enhancer design towards multiple cell type codes:

Remark 2.18: the enhancer CG15117 enhancer in T1 neurons was not predicted to be highly active, despite the authors claim that it is well characterized. The prediction is around 0.25. Does that make sense given it is well characterized?

Answer 2.18: Apologies for the lack of explanation on this point. We had studied that enhancer in our previous publication in which we trained DeepFlyBrain (Janssens, Taskiran, Aibar et al 2022). We decided to bring back the analyses performed on this CG15117 enhancer.

Indeed, the sequence may seem to have a low prediction score (0.25). During the training of the model, the labels we use are binary rather than the height of the ATAC-peak. After the training, if we plot the distribution of the prediction scores on the test set, we saw that the average prediction score was around 0.25, so our threshold to choose True-Positive sequences was 0.25. So we consider this sequence as positive and this prediction score is not used to claim that the sequence is well characterized. By well-characterized, we meant the code, binding sites, its target gene, single-cell accessibility, and in-vivo activity of this genomic sequence from (Janssens, Taskiran, Aibar et al., 2022).

This section of the text has been rewritten to avoid misunderstanding.

Remark 2.19: The statement "The nucleotide contribution scores before and after the in silico evolution show that the T1 enhancer code was barely touched". Although the nucleotide contribution scores did not change much, the prediction increases from 0.25 to 0.5. Warning: despite the literature in deep learning applications in genomics, independent nucleotide contributions are not explanations of predictions. This is well established in ML literature (across several papers, eg. Lipton, "The Mythos of Model Interpretability" arXiv, 2016) but has largely evaded the genomics field. Attribution maps should not be over-interpreted. They only provide a hypotheses at best for what a model has learned.

Answer 2.19: We thank the reviewer for this observation. We re-analysed this case and we agree with the reviewer that this particular double-code-example does not yield a perfect result. In the revised version we now consider this

as a failed example. The code of T1 is changed slightly, the prediction score changed, and the GFP pattern of the in vivo reporter, to show the expression pattern of the enhancer in the optic lobe neuron is also changed. From the two examples we tried, we thus consider one successful and one unsuccessful.

Regarding the comment about over-interpretation of the attribution maps: we agree with the reviewer that everything we do is based on the model's prediction and its interpretation power, and imperfections in the model may lead to wrong conclusions. For this reason we included a very large amount of experimental validations with 68 transgenic animals, 97 luciferase assays, ChIP-seq, ATAC-seq, and lenti-viral integration of human enhancers. Furthermore, we confirmed that the attribution maps of DeepMEL and DeepFlyBrain are in agreement with those from Enformer and from newly trained ChromBPnet models.

Concerns regarding enhancer design by motif implantation:

Remark 2.20: The authors should show the intermediate plots of motif scanning in supplemental figs. This can also reveal any positional bias learned by the model.

Answer 2.20: *We now show these plots in the Supplementary data file (Fig. R2.15).*

Figure R2.15 (Supplementary File 1): **a**, Nucleotide contribution score (DeepFlyBrain KC class) of the original random sequence (1st row) and after each motif addition (4th, 6th, 8th row). In silico saturation mutagenesis of the original random sequence (2nd row) after motif implanting (9th row). Delta prediction score of motif implanted over

the whole enhancer for Ey (3rd row), Mef2 (5th row), and Onecut (7th row). Selected implanting location is indicated by a red dashed line. Implanted motifs are indicated by grey dashed lines.

Remark 2.21: How were motif instances determined? Were the flanks optimally chosen (as was done in DeepStarr)?

Answer 2.21: We selected the motif instances that maximize the TF-Modisco pattern score (Fig R2.16). These are the instances with the highest affinity (strongest binding sites).

Figure R2.16 (Only this rebuttal): TFMoDISCO track (top) and consensus sequence used for motif implanting (bottom).

The flanks of the motifs, as done in DeepStarr, represent only 1 bp, next to a 5bp motif (GATAA), making it de facto a 6bp motif (GATAAg). Indeed, multiple PWMs from PWM databases contain the GATAAg PWM representing a variant of the GATA factor recognition sequence. Therefore, calling the 6th position a flank is actually a semantic issue (one could call it a flank, or a part of the motif if using a 1bp-longer PWM). In fact, the DeepStarr authors concluded that using the 6bp full motif performs better. We take this possibility fully into account and consider the entire motif with all informative bases. The convolutional filters in the CNN are 20bp wide, and always allow for 'flanking' bps to be included. Also TFMoDISCO uses long patterns (35bp). In conclusion, without restricting to any 'core' motif, we choose the motif instances based on canonical and tf-modisco patterns.

We clarified how motifs were selected in the Methods.

Remark 2.22: The analysis is very similar to the in silico experiments in BPNNet (Avsec et al. Nat Gen 2021) and GIA (Koo et al, PLoS Comp Biol, 2021), and deepStarr (de Almeida et al. Nat Gen 2022), which uses GIA. BPNNet and DeepStarr also validate results experimentally. This application is slightly different (i.e. designing enhancers) but the overall goals are largely the same  identify motif code that drives high activity in respective system. The authors should do a better job of highlighting previous work, and ensuring that they distinguish the contributions (if any) of this approach versus previous approaches.

Answer 2.22: Thank you for this comment, we now expand the literature review and discussion to include and credit all these important earlier works on enhancer modeling and enhancer decoding, and we highlight better our key novelties.

- **Modeling and enhancer decoding** (with experimental validation): BPNNet, DeepSTARR, and our own publications on DeepMEL and DeepFlyBrain are all somehow comparable: to present deep learning models trained on enhancer accessibility (DeepMEL/DeepFlyBrain), enhancer activity (DeepSTARR), or TF binding (BPNNet) data. Our work and BPNNet was focused more on cell type specific enhancers, while DeepSTARR focused on developmental versus housekeeping enhancers. GIA is also comparable, but models RNA

binding proteins rather than TFs. In the current work, we re-use earlier models and we don't claim novelty on the enhancer modeling or DeepExplainer aspects. Some of the analyses we do using these models are indeed similar to earlier works (e.g., the use of DeepExplainer), but this reflects the current state-of-the-art and is an established method/tool rather than a novel finding.

- **Enhancer design** using deep learning models: we are, to our knowledge, the first to design cell type specific enhancers guided by deep learning models. We do this with three strategies and compare them. We are the first to test the designed enhancers both in vitro (human) and in vivo (Drosophila). We provide the first in vivo evidence that designed enhancers are cell type specific, with a success rate that is comparable or even better than the success rates of genomic enhancer testing. The most similar work is perhaps Vaishnav et al. Nature, 2022, where in silico evolution was performed to study the evolvability of yeast promoters (thus not from random sequences, and no design for cell type specificity). We strongly believe that we go significantly beyond this work. The DeepSTARR paper shows no enhancer design, rather they score with the DeepSTARR model 1 billion random sequences and select the highest scoring ones. Even if that approach is considered as design, the DeepSTARR enhancers are functional in S2 cells but, in that work, there is no notion of cell type specificity.
- **New insight into enhancer biology.** We are, to our knowledge, the first to exploit the enhancer design process to gain insight into the Metazoan enhancer code. This reveals important biological insights that, to our knowledge, have not been described in as much detail as we do, including the contribution and balance of activator and repressor motifs. Using experimental follow-up we provide solid evidence of how repressor motifs can abolish enhancer activity, and how their presence in random sequences can hamper enhancer design if they are not removed.
- **Advanced design.** We are the first to take the enhancer design process to more advanced designs, and we show, (1) that a minimal enhancer of only 49bp can be designed that is fully cell type specific; (2) that enhancers active in multiple cell types can be pruned, so that only its activity is narrowed to a single cell type; that enhancers active in one cell type can be augmented to become active in a 2nd cell types. We believe these are profound findings.
- **Design path confirmed by other models.** In the revised version we also bring forward two different enhancer models, namely ChromBPNet and Enformer. To our knowledge, it is also the first time that model predictions and DeepExplainer predictions for motif architecture are compared between different modeling approaches to verify potential model biases. The agreement between the models shows that our models are not biased, but more importantly, that we have reached a very deep explanation (with experimental validation) of the enhancer code for the systems we study. A key finding is that these alternative models also agree on our synthetic sequences, providing additional evidence that they represent true enhancers (multiple models and experimental evidence). We believe that our results and the confidence we achieve is profound and provides significantly novel information for the community.

Remark 2.23: The claim at the end of the section that just 3 binding sites with a minimal length is sufficient to serve as a functional enhancer is not fully supported. This is from an anecdotal example.

- Suggestion: If authors want to make this claim, they should implant this motif combination in different backgrounds and map out a distribution. This would test whether it is just the combination of motifs or whether the combination with sequence context is important.

Answer 2.23: Just the combination of motifs, within the 49 minimal enhancer, is sufficient for the enhancer activity as shown by the in vivo experiment, but when this short sequence is placed within a larger sequence, then the background plays a role, as expected. We assume the reviewer refers to the context *within* these 49bp, meaning the few leading, trailing, and spacing base pairs between the motifs.

These spacing nucleotides are indeed important too, because if they would be deleted (bringing the sites too close to each other), or if they are converted into repressor sites, then the prediction score is strongly decreased. In other words, the absence of repressive context is important. This is in line with our other results, for example our finding that random sequences often contain repressor motifs (this is also a type of context, in which activator motifs are implanted).

We followed the reviewer's suggestion and implanted this motif combination in 1 million different backgrounds and mapped out a distribution (Fig. R2.17a). When we visualize the nucleotide contribution scores of sequences with lower prediction score, they indeed show repressor sites (Fig. R2.17a).

Figure R2.17 (Supplementary data): **a**, Distribution of prediction score for the 49 bp short enhancer with 1,000,000 random background. Dashed lines indicate prediction scores for the 1st (orange), 50,000th (green), 200,000th (red), 500,000th (purple), and 999,900th (brown) worse background. **b**, Nucleotide contribution score (top) and in silico saturation mutagenesis (bottom) of the 1st, 50,000th, 200,000th, and 500,000th worse background.

Next, we performed a validation experiment using new transgenic fly lines, whereby we introduced repressor sites inside and outside the minimal enhancer (i.e., context) without touching any of the three activator sites, and as predicted, this completely abolished the enhancer activity *in vivo* (Fig. R2.18). This clearly shows that even if the activator sites are present, in the context of repressor sites, the enhancer activity can be suppressed.

Figure R2.18 (Figure 4e,f,g): **a**, Nucleotide contribution scores of a selected random sequence in its initial form (first track) and after Ey, Mef2, and Onecut implantations (second track). The position of the motifs are shown with dashed lines. Delta prediction score for in silico saturation mutagenesis (third track). Black circles: selected mutations to generate repressor sites. Nucleotide contribution scores after generation of repressor sites (fourth track). The position of the mutations are shown with dashed lines. **b-c**, In vivo enhancer activity of the cloned 500 bp sequence with Ey, Mef2, and Onecut implantations (**b**) and after generation of repressor sites (**c**).

Even though our original minimal enhancer was selected based on its high score, we could still find sequences with even higher activity compared to our original design (Fig. R2.19).

Figure R2.19 (Supplementary data): **a**, Distribution of prediction score for the 49 bp short enhancer with 1,000,000 random background. Dashed lines indicate prediction scores for the 1st (orange), 50,000th (green), 200,000th (red), 500,000th (purple), and 999,900th (brown) worst background. **b**, Nucleotide contribution score (top) and in silico saturation mutagenesis (bottom) of the 999,900th worst background.

This experiment actually allowed us to revisit the question about flanking nucleotides. One cut for example has a slight preference of A/T flanks (Fig. R2.20). This also provides information that one could consider as 'context' or as part of the motif (i.e., semantics). We do believe that this is important information for the reader.

Figure R2.20 (Supplementary Figure 4a): Preferred nucleotides flanking implanted motifs ($n = 2,000$).

Concerns regarding enhancer design by GANs:

Remark 2.24: Again, should reframe first sentence because it makes it seem like the GAN for DNA sequences is a different field. The authors should mention here that the GAN was a copy of an existing method with very much the same architecture. Credit isn't given until the methods. To a casual reader, this makes it seem like the method is novel. Also, another GAN was applied to DNA sequences previously, feedback GAN, though it was meant for protein coding regions, not regulatory genomics (Gupta and Zou, Nat Mach Intell, 2019).

Answer 2.24: We already referred to previous work of GANs on in silico enhancer design by Killoran et al. 2017. Now in the revised version, we expand the literature review including additional citations including Gupta & Zou as suggested. We do not claim that we came up with a brand-new GAN architecture. We indeed use the architecture of the paper the reviewer mentions even though the code of that paper is not available, and we describe this in the Methods section.

Remark 2.25: The GANs here are not well characterized beyond the positional GC bias in the sequences. What is the distribution of predictions of the GAN generated sequences? How does that compare with the predictions of the original genomic sequences?

Answer 2.25: We calculated the distribution of predictions of the GAN generated sequences and compared it to the predictions of the original genomic sequences (Fig. R2.21). This was a good suggestion and is informative for the reader.

Figure R2.21 (Supplementary Figure 5a and Supplementary Figure 13a): Prediction score distribution of the generated sequences sampled at different iterations of the GAN training (in blue), original genomic sequences that are used to train the GAN models (red), and background sequences that are generated with different order of k-mer compositions (gray) for KC (a), for MEL (b).

We also performed an additional quantitative analysis to test which activator and repressor motifs are included/removed by the GAN generated sequences during the training process, and compared that to the motif fractions in genomic enhancers (Fig. R2.22). We see that over iterations, the SOX10 and MITF motifs are increasing while the presence of ZEB repressor sites is decreasing.

Figure R2.22 (Supplementary Figure 5b and Supplementary Figure 13b): Average number of motif hits at each GAN iteration compared to Genomic enhancers for KCs (a) and for MEL (b).

In our view, the GAN models can also generate sequences but the steps of generation or how it is established is not as explainable as the first two approaches. That is the reason we did not explore this strategy as deep as the other two. Now we highlight this also in the discussion.

Remark 2.26: Fig. 5b shows correctly predicted sequences. Does this mean predictions were above a 50% threshold? How would random sequences that contain the GC-bias do here? This would be a better negative control than the Markov model which doesn't learn the positional GC-bias.

Answer 2.26: Our random sequences are already GC-bias corrected based on genomic sequences (Fig. R2.23a). Now we explain this better in the text. The prediction scores of GC-bias corrected random sequences are close to zero and are shown throughout the manuscript, some examples are below showing distributions of GC corrected random sequences (e.g., the random sequences before motif implantation, or the random sequences before the in silico evolution process) (Fig. R2.23b,c).

Now we show actual prediction score distributions for the GAN generated sequences (Fig. R2.21).

Figure R2.23 (panel **a**: Supplementary Figure 1a; panel **b**: Figure 2a; panel **c**: Figure 4a): **a**, Distribution of GC-content in GC-adjusted random sequences (green) and genomic regions (red). **b**, Prediction score distribution of the sequences for the γ -KC class ($n = 6,000$) after each mutation. The box plots show the median (center line), interquartile range (box limits), and 5th and 95th percentile range (whiskers) for KC directed mutations (blue) or random drift mutations (orange). **c**, Prediction score distribution of the sequences for the γ -KC class ($n = 2,000$) after each motif implantation and after 15 mutations. Abbreviations are used for Ey (E), Mef2 (M), Onecut (O), and Sr (S). The box plots show the median (center line), interquartile range (box limits), and 5th and 95th percentile range (whiskers). In **b** and **c**, black arrows indicate initial random sequence values.

Remark 2.27: To better characterize the generated sequences, the authors should do motif enrichment analysis of genomic sequences and GAN sequences to further characterize the number of motifs within GANs, their specificity/similarity to strong motifs (vs weak motifs), and the diversity of motifs learned.

Answer 2.27: We thank the reviewer for this suggestion. We have performed comprehensive motif analysis by using both PWMs and TF-Modisco based annotations (Fig. R2.22) and highlighted the identified binding sites on the nucleotide contribution tracks.

Remark 2.28: The attribution plots look like the values are much lower than genomic sequences. What is the distribution of attribution scores for positions with motifs in GAN generated sequences compared to genomic sequences that have high enhancer activity (i.e. high accessibility prediction)?

Answer 2.28: Attribution scores are scaled with the prediction scores and their range doesn't represent actual biology. The magnitude of the attribution scores highly correlates with the prediction scores as can be seen clearly

with the mutational steps or on the evolved sequences (Fig. R2.24). Since the GAN generated sequences have lower prediction scores compared to the genomic sequences, they have lower attribution scores.

Figure R2.24 (only this rebuttal): Prediction score versus attribution score comparison. Prediction score distribution at each step of mutations (Left). Mean attribution score distribution at each step of mutations (Middle). Prediction scores versus mean attribution scores at the mutation 10 (Right).

Remark 2.29: The GANs generate softmax sequences -- continuous values and not one-hot. One hot sequences can be generated with the Gumbel-Softmax trick (Jang et al, arXiv, 2016) and be differentiable so that gradients can backpropagate to the generator. It seems more natural for discrete sequences. Why was a softmax chosen instead?

Answer 2.29: This is a good point. We followed the architecture from an earlier study (Killoran et al., 2017) based on their 'Model and Dataset Details' section and they used softmax activation. However, indeed, using the Gumbel-Softmax trick would make much more sense in this case.

Referee #3 (Remarks to the Author):

In this study by Taskiran and colleagues, the power of deep learning from transcriptional regulatory elements is turned to generate novel, cell-type specific regulatory elements, following the Feynman dictum of “if you can build it, you can understand it”. The aim of the work is to explore specific models for enhancer structure and function, gaining insights on possible evolution of enhancers, the distal regulatory elements that control expression of most genes in higher eukaryotes. Overall, this is a highly ambitious goal, and the authors make impressive progress in identifying the underpinnings of cell type expression using two systems, specific KC cells of the *Drosophila* brain, and “MEL” type melanoma gene expression. Both are supported by previous deep learning studies by the Aerts group, in which endogenous sets of differentially accessible chromatin and gene expression were carefully studied to train predictive models.

In the paper, the authors use three general approaches to evolve regulatory elements *in silico*, with *in vivo* validation of sample products. Exhaustive *in silico* mutagenesis, with deep learning models scoring the proposed elements at each step, are used to generate specific enhancers in less than 20 steps. A second approach assesses the predicted activity of background sequences into which binding site motifs of specific transcription factors are implanted, again with models scoring the expected output as positions are moved across the elements. A third approach uses generative adversarial networks, which have been used extensively in image analysis. Each approach succeeds to a certain extent in generating cell-type specific enhancers from random sequences, from genomic sequences that have sequence qualities of active enhancers but were scored as not accessible, or from enhancers that are active in a different cell type. The chromatin-based assessment for *Drosophila* indicates that many of the *in silico* evolved elements are functional in a highly cell-type specific manner (judged by GFP expression in KC cells, for example), while the MEL elements are assessed in transient transfections, a non-chromosomal setting.

Overall, the authors conclude from this work that deep learning provides the tools to efficiently identify cell-type specific elements. From the analysis of the *in silico* evolved enhancers, they conclude that the restrictions on generating a functional enhancer are modest; there is a certain amount of clustering of motifs, and in MEL elements some preferred binding geometries, but apparently many configurations are sufficient for the specificity. Prior difficulties in developing novel enhancers *ab initio* are attributed in part to the inability to pre-score likely winners (a benefit of deep learning models) as well as the observation here that many inactive random sequences have too many repressor binding sites, which must be eliminated. Generation of novel enhancers *in vivo* are suggested to lie as close as a few mutations away, which puts a surprisingly low bar for remodeling of GRNs through creation of new regulatory elements.

The results of this study are compelling, and *in silico* wed properly with *in vivo* validation. Especially compelling is the ability to mine the information developed by deep learning to generate likely active elements, which would appear to be an important extension of the field. There are a number of areas that the authors might consider to strengthen the impact of this work, noted below:

Remark 3.1: The study is based on models trained on endogenous regulatory elements, or putative elements, and the predicted modules are termed “enhancers”, yet the context in which the activity of the binding sites are actually tested in the fly or cell are much more permissive, namely, within close range of the basal promoters. The finding that even a small ~50bp cluster of binding sites works as an enhancer suggests that we are actually observing the interaction of some regulatory sites in close proximity to targets in the promoter. The demands for gene activation may therefore be heavily shaped by the megadalton complexes of basal actors that can already set up a chromatin context suitable for the cell-type factors to work within. A more refined consideration of what is being identified as a functional element would be helpful in the interpretation. The low threshold to mutate into an active element may actually be higher if distal elements need to overcome chromatin barriers, or have interaction Kds great enough to maintain looping over a considerable distance, against entropic effects.

Answer 3.1: This is a thought-provoking comment. We agree with the reviewer that all the tested synthetic and genomic sequences are tested in a reporter cassette, with a minimal promoter at close proximity (98 bp). We would like to note though that our setup for enhancer validation does not differ from (hundreds of) studies that test the

activity of distal genomic enhancers by cloning them in a reporter vector just upstream of a minimal promoter. We believe it is fair to follow the same procedure for synthetic enhancers, and consequently to use the term enhancer (in the classical definition an enhancer is independent of location and orientation). Our conclusions are based on in vivo (and in vitro for human) reporter assays, assays that are generally accepted in the field to test enhancer function. Our purpose is to create synthetic enhancers that behave like genomic enhancers, and test them using approaches used for genomic enhancers, including distal genomic enhancers. Genomic enhancers that are distal, recapitulate their function in a reporter assay, being cloned directly upstream of a minimal promoter. What is crucial is that our in vivo tested enhancers are specifically active in a given cell type, such as Kenyon Cells, where they are designed for. This means that the megadalton complex at the promoter is not sufficient to activate the reporter gene in any other cell type in the brain. It will only activate the reporter when a correct enhancer is present. Note also that not all our designed enhancers are functional. We believe that this cell type specific control of transcription (spatial and temporal) warrants the use of the term enhancer.

The question whether a distal enhancer, when placed more proximal, would become more promiscuous, is interesting by itself (and relevant for genomic enhancers), but we consider this question not specifically related to synthetic enhancers. In fact, we are not aware of previous studies where it has been shown that when distal enhancers are placed more proximal, they would become more promiscuous. Nevertheless, it is an intriguing question, yet to answer it would require many genomic CRISPR deletions and insertions, or very large transgenes, which would not be achievable within a reasonable timeframe for a revision.

We could nevertheless address this question with a new analysis where we were able to simulate this setting. To this end we made use of the Enformer model, which has a large (200kb) receptive field and has been shown to recapitulate relatively distal (e.g., 50kb) enhancers. In addition, Enformer has been trained to predict the gene expression at the TSS (using CAGE data). We made use of this model by performing in silico CRISPR experiments: we replaced distal genomic enhancers with synthetic enhancers and measured the changes in the gene expression prediction during sequence evolution compared to expression of the gene with its genomic enhancer. We observed that our enhancers, when they are placed distal to their target gene TSS, behave as expected in terms of ATAC or DHS predictions, and even in terms of gene expression predictions. We believe this is an important result and included it in the manuscript.

We first used a control experiment, with a genomic enhancer that we manipulated, to test whether Enformer correctly captures the MEL enhancer logic, in perturbation experiments (Fig. R3.1). Indeed, when MITF or SOX10 sites are removed, the Enformer-predicted DNase signal is strongly reduced (delta score is negative). When ZEB repressor sites are removed, the predicted DHS signal increases (delta score is positive), and when ZEB repressor sites are added, DHS is reduced. We could recapitulate these perturbations using predicted H3K27Ac ChIP-seq data, and using CAGE data on the TSS that is distal to this intronic enhancer.

Figure R3.1 (Figure 5h): **a**, Enformer prediction tracks for three classes for the IRF4 enhancer. Top track: WT IRF4 enhancer prediction score, other tracks: delta of mutated IRF4 enhancer prediction score vs WT IRF4 enhancer prediction score.

Next we introduced a random sequence at this location, replacing the IRF4 genomic enhancer, as well as the different steps during the evolution process (Fig. R3.2). The DNase predictions by Enformer recapitulate very well our predictions and luciferase experiments, but now from a distal location.

Figure R3.2 (Figure 6j): **a**, Enformer prediction tracks for three classes for melanoma EFS-4 sequences added in place of the IRF4 enhancer. Top track: random sequence prediction score, other tracks: delta of mutated sequence prediction score vs random sequence prediction score.

Next, we moved this enhancer to a proximal location, an upstream distal location, and a downstream location, which all show the same evolutionary trend in DNase predictions (Fig. R3.3).

Figure R3.3 (Supplementary Figure 9c): Enformer prediction tracks for three classes for melanoma EFS-4 sequences added 10kb upstream (a), 5 kb upstream (b) or 17.5kb downstream (c) of the IRF4 enhancer. Top track: random sequence prediction score, other tracks: delta of mutated sequence prediction score vs random sequence prediction score.

Remark 3.2: Some metazoan regulatory elements are capable of exhibiting a high degree of cell-type specificity using a very simple logic; if that factor is capable of driving gene expression by itself, and if the cell-type specificity is “computed” by enhancers that integrate developmental information and drive cell type expression of the factor, then a few binding sites for Pax proteins (for instance) will give eye-specific transcription. Many other elements that have been dissected don’t have such simple design. The sparkling enhancer analyzed by Barolo and colleagues shows a confounding degree of complexity, while the shaven-baby elements studied by Fuqua and Crocker exhibit extreme sensitivity to individual mutations, such that all sequences appear to influence the output. The authors should discuss evidence for whether the simple grammar identified here is perhaps representative of a certain class of elements, and how one would distinguish that if so. Overall, a broader consideration of types of enhancer structure/function would strengthen the Discussion.

Answer 3.2: Thank you for this important comment. The experiments we perform show that both in human and fly, even in completely different systems (melanocytes/melanoma vs neurons & glia), the components that make an enhancer an enhancer are the transcription factor binding sites, activator versus repressor sites, their combination, distance to each other, their affinity, number, and orientation. The complexity is increased when an enhancer has

multiple and/or overlapping binding sites where their corresponding TFs are all expressed; and when the enhancer is active in multiple cell types (and activated by a different combination of TFs).

Arrangement of regulatory sites. The sparkling enhancer (called *spa*) analyzed by Barolo and colleagues contains indeed a high degree of complexity, with five Su(H) repressor sites (the TF can become an activator upon Notch signaling); three Pnt/Yan sites; and three Lozenge sites. Even though it is complex, the components of this enhancer are similar to the enhancers we describe, namely a combination of multiple activator and repressor sites. The authors attempted to create a synthetic *spa* enhancer by placing these sites in a random background sequence, using different spacings, but could not achieve a functional enhancer. It is plausible that with a deep learning model such an enhancer could be created using the principles we lay out in our work, but this would require a new deep learning model and appropriate training set tailored to the embryo/epidermis, which is outside the scope of this work. The authors found that deletions that affect 'contextual' sequence, between the functional sites, affect enhancer activity, but they could explain this by the changes these deletions caused in the distance between two sites. All these principles (activator sites, repressor sites, distance between sites, etc) are present in our models and synthetic enhancers (we show distance preferences in our motif implantation approach). The authors conclude that TF binding sites alone are not sufficient but **“additional essential patterning information is supplied by the arrangement of regulatory sites”**. We fully agree with this, and our work is perfectly in line with this work. We show that the deep learning models capture these requirements for arrangement of motifs. This is probably the most important conclusion in our work, that deep learning models allow for synthetic enhancer design, taking all these constraints into account, while uninformed random arrangements of motif combinations have much lower success rate.

Highly dense enhancers. It is certainly possible that enhancers with high density of activator and repressor sites show changes in activity upon most mutations, as in the example of the shavenbaby enhancer by Fuqua & Crocker. In fact, we show a minimal enhancer that is very dense and would be affected by nearly any mutation. We also show dual-code enhancers that would be affected by a high number of mutations. Finally, we show that repressor sites have a strong impact on enhancer activity and these can be generated relatively easily. For example, several dozen mutations in the IRF4 enhancer create a ZEB repressor site. Also short activator motifs can often be generated with single mutations.

Enhancer output beyond cell type specificity. An important consideration when comparing with other studies and other systems is to ask what exactly is measured: we measure cell type specific enhancer activity, and we created enhancers that have cell type specific activity. The work by Fuqua & Crocker on the shavenbaby enhancer measures not only activity, but includes detailed timing and spatial variation. A mutation in the enhancer that has a mild effect on the spatiotemporal output, was considered (rightfully) as functional. Interestingly, among our collection of successfully designed (positive) enhancers in Kenyon cells, we also observe a variety of weak and strong GFP levels, with more broad and more narrow patterns. It is thus possible that, in future work, improved models can be trained that capture such variation and have an increased resolution to measure 'enhancer output', and that this would lead to additional nucleotide importances, beyond the nucleotides we identified in this work. Our work nevertheless captures a 'core' cell type specific code that is necessary and sufficient to make the enhancer active in that specific cell type. We agree that this nuance is very important and that was not discussed in the previous version.

In conclusion, our data suggest that even when "complexity" is high (e.g., highly dense information, many packed TF binding sites), we still consider that the enhancer logic is interpretable by deep learning models (and that mutations can be interpreted by gain or loss of activator and repressor sites), at least when these models are trained on high-resolution training data that reflects the resolution of the enhancer assays. This is in agreement with the sparkling enhancer by Barolo, and, interestingly, in the Fuqua & Crocker publication, all the experimental evidence and follow-up analyses of enhancer variation focused on TF binding sites, while no other interpretations (of how mutations affect enhancer activity) were provided or experimentally proven in that publication. We think that the shaven-baby elements also fall in the category of complex enhancers that underlie precise levels and very specific spatiotemporal activity. The complexity of the sequences may need to be increased by the number of TF binding sites (similar to the *spa* enhancer). We also show an example with high density of motif information, namely our 49bp enhancer is very sensitive to individual mutations since it is densely packed. For a more loosely packed enhancer with no overlapping binding sites, such enhancers will be less sensitive to individual (random) mutations. We could confirm that by comparing our models with ChromBPnet and Enformer models, that are all in agreement on which are

the contributing nucleotides in an enhancer, while these models are trained on different training data and have different architectures. Finally note that every genomic enhancer (and every synthetic enhancer) has its own story (idiosyncrasy) and that the code is not overly strict. We can place Ey, Mef2, and Onecut sites next to each other as well as with more distance. Both configurations drive cell-type specific gene expression. We believe that for the systems we study our data is robust and convincing, but we also agree that nuance is always required, so we discuss this matter with openness and in relation to the works mentioned by the reviewer.

Remark 3.3: The in silico evolution of the elements tested, as they climb their ways to activity, appear to be a gold mine of possible insights on action of enhancers. The authors share some general insights, such as replacement of repressors with activators, and some spacing requirements. It would be very useful to quantitate some of these (predicted) effects. For instance one notes that in some cases shown, existing binding motifs for repressors that are distal to actual mutations go from strongly repressive to not repressive at all as activator sites appear. E.g. Supp. Fig. 1 EFS-1 repressor site at 270; Fig. 2D repressor at 390 disappears. Perhaps the deep learning is indicating that repressor sites work in a cooperative fashion, and with the replacement of one site, the distal one is rendered inert? For this effect and others, it would be quite interesting to understand the trends that may drop out of the thousands of in silico experiments conducted.

Answer 3.3: We agree that the paths of in silico evolution from random sequences to full blown enhancers is thrilling to investigate. In the revised version we added many new analyses to further study the role of, and cooperativity between, activator and repressor sites.

When we measure the activity of the synthetic enhancers in each mutational step, we see that a single strong activator binding site cannot drive the expression by itself. But when the number of sites is increased, an additional strong TF binding site increases the expression much further. Likewise, when we add repressor binding sites one by one to active synthetic enhancers without disrupting any activator binding site, we observed that increasing the number of repressors or creating repressors at a close proximity to activators, decreases the activity further. From these two observations we conclude that:

1. There is a balance between activators and repressors. The number of binding sites in each category fine-tunes the output activity.
2. When a repressor binding site is in a close proximity to an activator binding site, it decreases the activity of the sequence much more compared to a repressor that is distant to activator binding sites.

We could confirm these findings by different models (ChromBPNet and Enformer) and by experimental validations, both in human and mouse. For human we performed an important experiment, namely ZEB2 ChIP-seq, in the same melanoma cell line where we performed the ATAC-seq (training data) and luciferase experiments. We observed a ZEB2 ChIP-seq peak on the IRF4 enhancer, where also a SOX10 ChIP-seq peaks is present - in other words, both the repressor and activator can bind the same enhancer (our data does not provide information on whether this binding is on the same allele, at the same time) (Fig. R3.4a). Based on this intriguing finding, we mutated the ZEB2 site in the IRF4 enhancer, and indeed, activity is increased (Fig. R3.4b). When additional ZEB2 sites are added, activity is abolished. When the activator sites are mutated, activity is also abolished.

Figure R3.4 (Figure 5a,b,c,d,e): **a**, MM001 ATAC-seq (top), SOX10 ChIP-seq (middle), and ZEB2 ChIP-seq (bottom) tracks of the IRF4 gene. The IRF4 enhancer location is highlighted in red. **b**, Luciferase activity levels in MM001 of the WT IRF4 enhancer and mutational variants. **c**, Nucleotide contribution score (top) and in silico saturation mutagenesis (middle) of the IRF4 enhancer for DeepMEL2 human melanocyte-like melanoma class. In vitro saturation mutagenesis in SK-MEL-28 (bottom). Mutations resulting in a drop of both prediction and activity, and that generate a ZEB2 binding site are highlighted with a black circle. **d**, Scatter plot comparing in vitro (x-axis) and in silico (y-axis) saturation mutagenesis values of the IRF4 enhancer. **e**, Nucleotide contribution scores of newly generated ZEB2 motifs from mutations highlighted in panel **c**.

Repressor-vs-activator presence that we found in the design process could thus be confirmed experimentally. In a next experiment, we added repressor sites *after* the in silico evolution process, without touching activator sites in the functional synthetic enhancers. Both in human and *Drosophila*, this completely abolished the activity (as predicted by the model) (Fig. R3.5). Thus, even a perfect constellation of activator sites can be suppressed by the presence of (several) repressor sites.

Figure R3.5 (panel a: Figure 3b; panel b: Figure 4f,g; panel c: Figure 5d; panel d: Figure 6f and Supplementary Figure 7f): **a**, In vivo enhancer activity of the EFS-4 sequence before and after generation of repressive sites (top panels) and of five other originally active synthetic enhancers in which repressor sites were generated. **b**, In vivo enhancer activity of a KC enhancer generated by motif embedding before and after generation of repressive sites. **c**, Luciferase activity levels in MM001 of the WT IRF4 enhancer and mutational variants. **d**, Luciferase activity levels in MM001 of three MEL EFS sequences with incremental mutation steps and addition of repressor sites.

Another set of experiments we performed is to show each step of the evolutionary process, and to experimentally test multiple sequences at different evolutionary steps (Fig. R3.6).

Figure R3.6 (panel a: Supplementary Figure 1n; panel b: Figure 6f and Supplementary figure 7f): **a**, In vivo enhancer activity for three KC EFS sequences after every two mutations. **b**, Luciferase activity levels in MM001 (left) and correlation with prediction score (right) for three MEL EFS sequences after incremental mutation steps.

Remark 3.4: The number of tested elements is small, but there are “successes” and “failures” – are there properties of each that one can draw on to understand the shortcomings of the models that are driving evolutionary design?

Answer 3.4: This remark being very similar to Remark 1.2 from reviewer 1, we provide the same answer below: We followed the advice of the reviewer and tested 68 sequences in total by generating 43 additional transgenic lines. Using these new lines we performed further validation experiments, and examined in detail the source of false positives. These new results are incorporated throughout the revised manuscript and below we provide an overview of the new experiments related to Kenyon cell enhancers:

- g) We performed 5 additional mutation steps (in silico evolution) on two negative and two weak enhancers from the first manuscript version to examine if they now become positive (under the assumption that the in silico evolution process did not yet reach a high enough score (Fig R1.4, R1.5)). All four are now positive (Fig R1.6). This suggests that the negatives were indeed not yet optimized to their highest end point. Interestingly, the additional mutations that make these enhancers active affect the same activator motifs across positive enhancers (i.e., either creating stronger sites, or additional sites). In addition to the previous enhancer fly lines, we designed 7 new Kenyon cell synthetic enhancers (in silico evolution with 15 mutations) (Fig. R1.7).

Figure R1.6 (Top panels: Supplementary Figure 1i, bottom panels: Figure 2d): In vivo enhancer activity of cloned EFS sequences with no or weak activity after 10 (top) and acquired activity after 15 (bottom) mutations.

Figure R1.7 (panel a top and panel b: Supplementary Figure 1k,h; panel a bottom: Figure 2d): **a**, In vivo enhancer activity of negative (top panels, Enhancer-GFP co-labeling with Dachshund (DAC) to stain Kenyon cells) and positive (bottom panels, Enhancer-GFP) enhancer sequences after 15 mutations. **b**, Positive enhancer EFS-13 after 15 mutations: combined image of Enhancer-GFP with DAC (top panel). The expected location of KC is shown with dashed circles.

- h) For three of our positive enhancers for either *Drosophila* Kenyon cells or melanoma, we tested the mutational steps along the design path to investigate how many mutational steps are required to go from a random sequence to a Kenyon cell/melanoma specific enhancer. Starting from the random sequence, we tested the activity of the sequences after every 2 mutations. We did this for *Drosophila* enhancers in vivo (each sequence required a new transgenic line) (Fig R1.8a), and for human melanoma enhancers using luciferase assays (Fig R1.8b). All *drosophila* enhancers started to show activity after 8 mutations and luciferase measurements in the MM001 cell line showed a gradual increase in activity after 7 mutations and displayed very high correlation with DeepMEL2 predictions.

Figure R1.8 (panel **a**: Supplementary Figure 1n; panel **b**: Figure 6f and Supplementary figure 7f): **a**, In vivo enhancer activity for three KC EFS sequences after every two mutations. **b**, Luciferase activity levels in MM001 (left) and correlation with DeepMEL2 prediction score (right) for three MEL EFS sequences after incremental mutation steps.

- i) In the “near-enhancer” sequence category we had tested three different genomic enhancers and rescued their activity with only six mutations. Of these, 2/3 became positive, while one remained negative. We now performed an additional five mutations in this negative sequence (FP3), and tested it again in a new transgenic fly. It now became positive (Fig R1.9a). Thus all three ‘near enhancers’ have now been turned into an active enhancer. We also tested one additional “near-enhancer” sequence (FP1) and rescued it with 6 mutations (Fig R1.9b). Note that the name of the rescued sequences has been changed compared to the original manuscript (e.g. FP2 is now FP3).

Figure R1.9 (panel a and b bottom: Figure 2g; panel b top: Supplementary Figure 2j); **a**, In vivo enhancer activity of the FP3 sequence after 6 (top) or 11 (bottom) mutations. **b**, In vivo enhancer activity of the WT FP1 sequence (right) and after 6 mutations (left) to rescue KC activity.

From all these experiments, 4/6 glial synthetic enhancers (66%), all genome-rescued enhancers (100%), and 10 (11) out of 13 Kenyon cell enhancers worked (76-85%). KC enhancer number 10 is slightly positive but weak and a-specific so we consider it negative (Fig. R1.7). Two other KC enhancers (number 8, 9) we consider as negative. All three non-GFP expressing enhancers (8-10) were investigated further. We did verify whether the negative sequences did not have any repressor site generated in the junction of the enhancer and the vector sequence, or if mutations were present in the DNA synthesis or cloning, but this was not the case. Next, we double-stained these lines with anti-Dachsund antibody that labels KCs and observe a lesion pattern in the KC region of these three cases (R1.7a EFS8-10 vs R1.7b). This distinct lesion pattern (EFS 8-10) compared to the positive enhancer (e.g., EFS 13) can be observed in the fly brains (Supplementary Figure 1k vs. Supplementary Figure 1h). Based on the lesion we hypothesize a GFP misexpression causing a defect to the KCs. Due to the lesion pattern, these 3 KC enhancers are not 100% certain to be negatives, we have too few non-GFP expressing enhancers to identify possible sequence features that may cause them to be negative (or cell-lethal). Looking into the expression of these enhancers in wandering larvae did not lead to a clear conclusion either and we consider this potential developmental or cell-lethal defect out of the scope of the paper.

Remark 3.5: Some aspects of the experimental procedures were unclear. For example, the fly elements are tested in the PH-Stinger vector (reference is needed), which is a P-element random insertion vector. Yet, the Materials and Methods indicates directed insertion, as if using a phiC31 system. The cloning indicates that elements were introduced into the vector using a lambda-based recombinase system, yet Stinger has restriction sites for cloning upstream of the -43 site of hsp70. Was this vector altered to allow for in vitro recombination, and if so, what is the distance of elements to basal promoter?

Answer 3.5: In Aerts et al., PLoS Biology 2010 we described a new plasmid, based on pHStinger, where we replaced the MCS with a Gateway cassette, and introduced an attB site for phiC31 integration into the fly genome. We now refer to this in the Methods.

Remark 3.6: The evolved elements tested in MEL cells are more active than they are in MES cells, but it appears that even “native” enhancers are much weaker in MES cells, relative to Renilla. Are MES cells poorly transfected, or is the Renilla much more active in this cell type? Because of normalization, it is hard to know if the elements tested are robust in expression.

Answer 3.6: MES cells are indeed more difficult to transfect and display a lower level of activity. The same observation was made with MM099 (another MES cell line) in a previous study (Mauduit et al., eLife 2021). The positive control enhancers we used have been selected from this study. These genomic enhancers have been first identified via massively parallel reporter assay (MPRA) and then validated with luciferase assay where they showed a significant increase in activity compared to negative control sequences. MPRA relied on nucleofection, instead of lipid based transfection, which resulted in a higher transfection efficiency and more similar values across cell lines. Nucleofection was not used for the luciferase assay due to the large number of individual conditions and the low number of cells that each sample contains. To further investigate the specificity of the MEL EFS sequences, which was the original intent of the luciferase assay in MM047, we used the Enformer model. The model provided strong prediction scores for each sequence exclusively in melanocytes and SKMEL-5 DNase tracks (another MEL cell line) (Fig. R3.7).

Figure R.3.7 (Supplementary Figure 9a): Prediction scores for the top 50 DNase tracks for all MEL EFS sequences. The four first DNase tracks are: foreskin melanocyte male newborn, SK-MEL-5, foreskin melanocyte male newborn, SK-MEL-5.

Remark 3.7: Some minor points:

- Fig. 3 m, n – “Chromatin accessibility profile of this region” Presumably this is Pkc53e? Not clear from the lack of label in Fig. 3n.
- Renilla is capitalized.
- “prediction scores started to saturate [plateau?] after 20 mutations”

Answer 3.7: Thank you for pointing out these details. They have all been addressed in the main text of the manuscript.

Reviewer Reports on the First Revision:

Referees' comments:

Referee #1 (Remarks to the Author):

I commend the authors for the work performed in diligently addressing my concerns as well as those of the other reviewers. I find that all of my concerns have been satisfactorily addressed, with the exception of the analysis of combinatorial random mutations. The lack of appropriate in-house computational resources, including memory limitations, can be readily resolved through the utilization of cloud computing. If investigating all five possible mutations is still unfeasible, exploring four or even three mutations should not be as complex.

Referee #2 (Remarks to the Author):

In the revised manuscript, the narrative appears to have expanded considerably. Streamlining it would greatly enhance its clarity and impact. While the range of experiments is commendable, it would be advantageous for the authors to refine their over-reaching claims and ensure that their stated discoveries are in line with the supporting evidence provided.

Considering the evidence, a reevaluation of the paper's central message is suggested. I believe that the revised focus could instead highlight that a DNN trained on genome-wide functional assays (eg. ATAC-seq) can serve as an effective oracle to score the function of new enhancer sequences. Previously, the use of trained DNNs as oracles for the prediction of new sequences was validated experimentally in different contexts, including yeast promoter, TF binding syntax, variant effects via MPRA, enhancers in episomal settings in drosophila, plus more. This study explores many other biological systems in an in vivo setting. The success of DNN oracles implies that they learn good approximations of the underlying sequence-function relationship for enhancer activity, learning cis-regulatory features across natural genetic variation, which is currently under debate with some in the field claiming that it is necessary that the training data contains higher levels of genetic variation to learn more generalizable cis-regulatory representations. If this were the main scope of the paper, I believe it is quite significant and helps add another data point to the benefits of DNNs in regulatory genomics. The design methods are not advances, they are simple local search algorithms that were previously established, albeit in different systems but still for regulatory DNA. Moreover, the cell type specificity aspect to this is overblown as there is no outright terms in the objective function that suppresses off-target cell types, it is simply what previous groups have done in the past – focus on one task. In addition, the other motif analyses that are (in my opinion) not compelling could still be a part of the paper but should be downplayed as just observational statistics to gain some insights into what the DNN considers to be highly functional enhancers -- unless the claims are backed by follow up experiments that directly support the claims and not just indirect support.

Below I provide bullets on how I interpret the paper (based on the presented evidence):

- DNNs trained to predict enhancer activity (or chromatin accessibility) from sequence are useful as scoring functions to design functional enhancers. By using this DNN as a so-called oracle, ie. a scoring

function, the main task of enhancer design becomes a sequence search algorithm problem. They tested 2 local search algorithms that previously existed, including a nucleotide-level greedy search (namely sequence evolution) and motif-level greedy search (i.e. motif embedding). They also tested a previously established GAN (previously established in one arxiv and tested by a different group in another publication, which was not cited; Zrimec et al. Nature Commun 2022), which provides a deep generative approach to sample more widely in sequence space instead of local search. Local searches have optimization guarantees to a local maximum, but no guarantee to a global maximum. The GAN in principle has the potential to sample very different regions of sequence space but this GAN was not conditional and so the generated sequences are not necessarily based on function. Instead, the training sequences for the GAN were biased to represent functional sequences. This work shows that local optima can yield highly functional enhancers! So, the search for global optima may be overblown or not necessary in terms of enhancer function as it pertains to a direct enhancing of a nearby minimal promoter. But this was not mentioned.

- Through an established model interpretability method, called deepexplainer, they perform motif analysis to observe how the distribution of motifs changes throughout the sequence optimization. They find an interesting observation that repressors get removed while activators get created. Despite the terms model interpretability or explanations, deepexplainer does not provide a full explanation of model predictions -- it provides independent nucleotide contributions -- not motif level contribution scores. Even though this is commonly done in deep learning papers in genomics, at least in the past, there is a hazard of over-interpreting extended patterns (like motifs) that emerge from (additive) attribution maps. Nevertheless they can be quite informative to generate hypotheses as part of an observational analysis. Importance of motifs and their interactions with other motifs in the same sequence can be quantified with carefully designed in silico perturbation experiments, and these sorts of analyses are becoming increasingly popular for model interpretability over the last several years in genomics (Avsec et al. Nat Gen 2021; Koo et al. PLoS Comp Biol 2021; deAlmeida et al. Nat Gen 2021; Toneyan et al. Nat Mach Intell 2022; Karollus et al. Genom Biol 2023; + many more). But this was not explored here.

- Nevertheless the experiments do validate that the generated sequences with putative motif patterns lead to functional activity in a cell type. The problem is this corresponds to the sequence and not necessarily the motif "explanations" that is hypothesized. Claims that are made of the motifs should be proven by placing the motif combination into different background sequences and seeing if those are functional. Otherwise, the tone should reflect that it is not a discovery but an observation from attribution analysis.

- The GAN was largely unable to capture the regulatory grammars that make up strong enhancers, but it was still better than a poorly chosen random baseline (because the model's positional bias was not considered in the null model). However, since generation occasionally leads to high activity by chance, an inefficient, yet effective scheme was to use the DNN scoring function to subselect the rare sequences that yield high functional predictions (i.e. tail behavior of the distribution). This of course can also be done with random sequence generation, at a lower efficiency. This would not be considered a good generator in any machine learning field. Again, it is the oracle scoring function that enables the identification of functional enhancers, i.e. what the authors call enhancer design.

A broader set of critics:

- Despite making it a point in the `_first_` review that code and data was not provided, there is only an illusion that meaningful code is provided in the revision. The provided code is for plotting figures and

even then, it still does not contain any of the necessary files to actually run the notebooks. This is unacceptable. In fact, even if the intermediate files are provided, the code would still be broken as there are missing functions. For instance, the notebook FLY_KC_Repressors.ipynb calls `utils.model_load` but that function doesn't exist in `utils.py`. The zenodo link to code w/ intermediate files is restricted and inaccessible to the reviewer. This is unacceptable!

- The main contribution is supposed to be a set of sequence design methods that can be used to design cell type specific enhancers, but I have deep concerns that this is not a general solution. Despite the proof of concept for the designed enhancers validated *in vivo*, the objective function only considers the cell type of interest and it does not explicitly repress activity in other cell types during the optimization process. This can happen if the cell type chosen has enhancers that use a unique set of motifs. In other scenarios, off-target effects can occur. The benefit here is the experimental validation. The fact that it worked for the chosen cell types might say something about the orthogonality of motifs in enhancers across the cell types explored by DeepFlyBrain. But this requires additional experiments to assess if this is general. I have doubts that this holds across cell types that have overlapping TFs employed in enhancers. Other approaches that consider cell type specific enhancers directly deal with this effect by including a minimization term for off-target cell types (Gosai et al. *bioRxiv* 2023). In any case, the claims of novelty about cell type specificity should be toned down as cell type specific enhancer design means targeting a cell type of interest and not other cell types. This basic approach does not solve the general problem of cell type specific enhancer design and does not push the field forward from where it already was.

- Many general claims of motif grammars are based on observational motif analysis of attribution maps. I would not have had many objections a few years ago. But the standards of model explanations have improved over the past several years and cursory motif analysis solely based on observational analysis should be supplemented with interventional analysis with *in silico* experiments as done in: Avsec et al. *Nat Gen* 2021; Koo et al. *PLoS Comp Biol* 2021; deAlmeida et al. *Nat Gen* 2021; Toneyan et al. *Nat Mach Intell* 2022; Karollus et al. *Genom Biol* 2023; + many more. The over interpretation of attribution maps is a hazard, they come with many insights but also have intrinsic pathologies that are well known in the broader ML community and it is still taking time to reach the broader genomics community.

- The validation of the designed sequences *in vivo*, which contain the putative motifs, demonstrates that the (whole) designed sequence is functional but only indirectly supports motif-level claims. Validation of motif-level claims could be supported with additional experiments both *in silico* (and/or *in vivo*). Ideally, a set of motifs would be implanted in different random sequences (that are sampled from a good null distribution; see other comments below for this). Then the average activity across contexts could inform the motif's effect size. These experiments can be done *in silico* with model predictions to prove that the model has indeed learned these features or *in vivo* to provide supporting evidence that this reflects real biology.

- For motif embeddings, a good control is to implant numerous activator motifs (from the same set) in random locations. I would suggest implanting more than one of each motif, perhaps up to two or three rounds and measure the design quality based on the oracle predictions. My suspicion is that it would also promote a high cell-type specific function, but perhaps would be less efficient than the proposed greedy search.

- The GAN analysis is a negative result. The predictions of the GAN-generated sequences seem to be substantially lower than the predictions of motif embedding or sequence evolution. However, it remains difficult to draw any strong conclusions from this as the GAN hyperparameters were not

thoroughly optimized -- so is the GAN methodology flawed or is it just that a good setting was not found? The presented comparison with poorly chosen baseline model is not demonstration that the GAN works. Since strong statements cannot be made either way, it might be better to remove GANs from the story to simplify an increasingly complex story. The GAN requiring an oracle scoring function means that it is just slightly more efficient at occasionally generating a functional sequence (i.e. from tail behavior of the generated sequence distribution) than a "poorly" chosen null model. Also, here is another example of GAN for DNA regulatory sequences: Zrimec, Jan, et al. "Controlling gene expression with deep generative design of regulatory DNA." *Nature communications* 13.1 (2022): 5099.

- The lack of consideration for the ML field focused on designing biological sequences makes this paper read like it is less thoughtful about how it fits into the broader field. Moreover, there are serious drawbacks and limitations of using supervised DNNs as oracles, which are well established (eg. this recent manuscript, "Is novelty predictable?" by Fanning and Listgarten, but there are many more over the past several years). This manuscript reads as an over-confidence in the previously established DNNs which are used as oracles for sequence design, in this case DeepFlyBrain and DeepMel. However, it is well known that the reliability of model predictions changes under covariate shifts from the training distribution, that any interpretability is strictly through lens of DNN and may not be a reflection of the full biological process. An argument against this could be that it just works according to some validation. But the cases that the designed sequences failed may reflect pathological regions of the DNN-inferred sequence-function landscape. This was not explored.

Other concerns:

- The baseline of 5th order Markov is clearly not the right null model given the positional GC bias that DeepFlyBrain learns. A more suitable null should be random sequences that match the GC content per position.
- "Note that besides the greedy search that selects the best mutation at each iteration, alternative state space searches can be employed that may yield, for the same number of mutations, enhancers with higher scores, albeit with higher computation cost " There is no description of this method and it is written so vaguely that I have no idea what alternative search was used.
- abstract: "we could accurately define the enhancer code as the optimal strength, combination, and relative distance of TF activator motifs, and the absence of TF repressor motifs"  the validation performed does not prove optimality, just functionality.
- The work can sell itself without over-the-top language that is currently used throughout. Abstract: "well-trained models are reaching a level of understanding that may be close to complete" This should be removed and the other superlatives throughout should be minimized. (Examples of how sequence-based DNNs are failing and not close to complete: Karollus et al. *Genome Biol* 2023; Huang et al. *bioRxiv* 2023; Sasse et al. *bioRxiv* 2023)

Minor comments:

- For enhancer design towards multiple cell type codes: how were the mutations in the sequence evolution chosen? The description in the main text and the methods are exactly the same and no clear description of the implementation or the objective function is given.
- It would be helpful to see the attribution maps on the same scales, especially when claims are made based on comparing different attribution maps. (eg. Fig 4e among many others).
- Fig 4e axis labels are not there.

Referee #3 (Remarks to the Author):

In this revised manuscript, Taskiran et al. have undertaken a number of revisions to address topics that were raised in the first round of review of this manuscript. The manuscript now includes significantly more validated enhancers, including activity of elements that featured intermediate scores during *in silico* evolution. An expanded discussion of previous related studies better integrates this work into the field. Expanded ATAC-seq analysis demonstrates the concordance between activity and chromatin open-ness *in vivo*. ZEB2 is identified as the likely repressor involved in IRF4 enhancer regulation in melanocytes. Regarding conclusions for numbers or arrangements of motifs in active elements, the manuscript now provides convenience summaries of numbers and types of motifs, although subtleties of arrangement are not identified.

With regard to comments from Reviewer 3, the authors addressed the question of whether the *cis* regulatory elements that were developed indeed act as enhancers by testing activity *in silico*. These models predict that IRF4 elements should perform similarly in proximal or distal locations. (The authors were unaware of studies that indicate more promiscuous activity for minimal elements at a promoter position; for Dorsal and Twist activators, Szymanski & Levine 1995 describes one example of this frequently-observed phenomenon; the minimal MSU element lacks activity distally, and only synergizes with an existing promoter-proximal MSU when placed in an adjacent, not distal, position).

In the context of whether the computational approaches here will succeed with only “simple” enhancer design, the paper now discusses similarities of the fly and human findings, leading to the conclusion that these functional properties are likely broadly applicable. At the same time, they note that their assays lack the fine spacial, temporal and quantitative resolution of Fuqua et al. for *svb*, thus there may be additional properties still to identify computationally.

The revised manuscript also better develops insights into the likely balance between repressors and activators at active elements, providing more experimental evidence for assemblies that are dominantly regulated by repressors.

Two reviewers asked about lessons from prediction “failures”, and new data in this paper shows that such inactive elements are just a few mutations away from activity *in vivo*, indicating the inherent potential of the evolved elements.

Finally, a number of technical points regarding computational approaches, vector design and small writing issues are addressed. Overall, this manuscript describes a significant advance in our understanding of the potential and limitations to identification and *ab initio* generation of tissue-specific regulatory elements (likely to act as enhancers), and furthermore provides a rich dataset of *in silico* evolved *de novo* or genome-derived sequences to undergird studies of enhancer function. By applying these methods to two very different systems, the authors have provided evidence for the applicability of deep learning for *cis* regulatory elements in metazoans, and their findings will be of high interest to researchers in computational and systems biology, development, and evolution.

Author Rebuttals to First Revision:

Referees' comments:

Referee #1 (Remarks to the Author):

Remark 1.1: I commend the authors for the work performed in diligently addressing my concerns as well as those of the other reviewers. I find that all of my concerns have been satisfactorily addressed, with the exception of the analysis of combinatorial random mutations. The lack of appropriate in-house computational resources, including memory limitations, can be readily resolved through the utilization of cloud computing. If investigating all five possible mutations is still unfeasible, exploring four or even three mutations should not be as complex.

Answer 1.1: Thank you, we are very pleased that the Reviewer was satisfied by our revised manuscript.

Regarding the comment about combinatorial random mutations:

- In Figure 2a, we added for the first revision random mutations at each incremental step.
- We had also included in the first revision another type of optimization search (Supplementary Figure 2a,b,c): when the top 20 scoring sequences with one mutation are selected in each step (rather than only the best one), then after 5 steps we found a number of sequences with even higher scores compared to the greedy-search result. We already discussed this point in the previous revision; namely that alternative state space search algorithms or brute force searches can also yield high-scoring sequences.
- Related to this analysis, we now also tested the same number of sequences, namely 3.2 million, with 5 random mutations, but without following the selection process. When compared to the optimized sequence (greedy search), the scores of all these randomly selected sequences with 5 mutations are very low (below 0.06) (Figure R1.1).

Figure R1.1 (Only in rebuttal): Prediction score distribution for 3.2 million sequences after 5 consecutive random mutations (left) or after selection of the top 20 mutations for 5 consecutive mutations (right). Blue line represents the path that was taken by the greedy algorithm.

- Finally, as suggested by the reviewer, we now tested all possible 3 mutations by using 1,000 CPU cores, 13.5 terabytes of RAM, and 200 gigabytes of physical memory (to only keep the prediction scores) in total. All possible 3 mutations of a random sequence resulted in 0.5 billion sequences of which only 7 reached a prediction score equal to, or higher than our greedy search approach with the best sequence scoring only 0.057 (Fig. R1.2). Thus, it is possible to generate sequences with comparable or slightly better prediction

scores with this method but at a much higher computational cost that becomes unaffordable when we increase the number of mutations from 3 to 10-15, and if we increase the number of initial random sequences to 6000.

Figure R1.2 (Only in rebuttal): Prediction score distribution for 0.5 billion sequences with 3 mutations (left) and the same plot with different x-axis scale (zoomed-in, right). Blue line represents the path that was taken by the greedy algorithm with 3 mutations. Black line represents the prediction score of the initial random sequence.

Referee #2 (Remarks to the Author):

Remark 2.1: In the revised manuscript, the narrative appears to have expanded considerably. Streamlining it would greatly enhance its clarity and impact. While the range of experiments is commendable, it would be advantageous for the authors to refine their over-reaching claims and ensure that their stated discoveries are in line with the supporting evidence provided.

Considering the evidence, a reevaluation of the paper's central message is suggested. I believe that the revised focus could instead **highlight that a DNN trained on genome-wide functional assays (eg. ATAC-seq) can serve as an effective oracle to score the function of new enhancer sequences**. Previously, the use of trained DNNs as oracles for the prediction of new sequences was validated experimentally in different contexts, including yeast promoter, TF binding syntax, variant effects via MPRA, enhancers in episomal settings in drosophila, plus more. This study explores many other biological systems in an in vivo setting. The success of DNN oracles implies that they learn good approximations of the underlying sequence-function relationship for enhancer activity, learning cis-regulatory features across natural genetic variation, which is currently under debate with some in the field claiming that it is necessary that the training data contains higher levels of genetic variation to learn more generalizable cis-regulatory representations. If this were the main scope of the paper, I believe it is quite significant and helps add another data point to the benefits of DNNs in regulatory genomics.

Answer 2.1: Thank you for the positive words. We agree that the main scope is to use a trained DNN as an effective oracle to design functional enhancers, and that this is the core message of our paper.

We also find our follow-up analyses important, where we track individual mutations during in silico evolution (or motif embedding), to study the biology of enhancers (for example, finding how repressor motifs need to be avoided when designing enhancers, as shown by in vitro and in vivo validations). Both Reviewer 1 and Reviewer 3 are appreciative of these follow-up experiments, and the extended narrative in the revised manuscript was largely the result of addressing the reviewer's comments from the first review. We hope that we can keep these findings in the paper. We followed the suggestions to be careful in over-shooting by distinguishing those interpretations for which we have direct evidence from those that are more observational (see answers below, mainly from Remark 2.6, 2.10, and 2.19).

Remark 2.2: The design methods are not advances, they are simple local search algorithms that were previously established, albeit in different systems but still for regulatory DNA. Moreover, the cell type specificity aspect to this is overblown as **there is no outright terms in the objective function that suppresses off-target cell types**, it is simply what previous groups have done in the past – focus on one task. In addition, **the other motif analyses that are (in my opinion) not compelling** could still be a part of the paper but should be downplayed as just observational statistics to gain some insights into what the DNN considers to be highly functional enhancers -- unless the claims are backed by follow up experiments that `_directly_` support the claims and not just indirect support.

Answer 2.2: Thank you for summarising the comments clearly. We address them in detail below where they are separately mentioned. The three main comments are:

- **The design methods are not advances: we agree.** We already discussed this in our previous rebuttal - the in silico evolution and the motif embedding methods do not constitute the novelty of our work. They are simple and previously established for other purposes, yet the outcome is novel.
- **The objective function we use does not explicitly suppress off-target cell types: we agree.** We better explain in the answer to Remark 2.9 how we (implicitly) suppress off-target cell types without using such an objective function. This has to do with the particular design of our training set (cell type specific ATAC-seq peaks) and model.
- **Motif-level interpretation of enhancer function:** we understand the critique and have adapted the interpretation of our results. We now use motif claims only when we have direct evidence, and otherwise use them as observational statistics. We recapitulate the instances where we have direct evidence in our answer to Remark 2.6.
 - This point is also related to the comments about DeepExplainer-based motif-level predictions (Remarks 2.5, 2.10, 2.11). We provide a **new supplementary note** with an overview of

complementary analyses and experimental validations of DeepExplainer-based motif-level interpretations. We also review how we validated the DeepExplainer-based motif-level predictions in our DeepMEL2 and DeeFlyBrain publications. We realized that for the reviewer and the reader of the paper, such information may be useful, beyond simply citing our earlier work. Finally, we added to that supplementary note a new comparison of DeepExplainer predictions by DeepMEL2 and ChromBPNet, and compared them to in silico mutagenesis by DeepMEL2, ChromBPNet and Enformer.

Below I provide bullets on how I interpret the paper (based on the presented evidence):

Remark 2.3: DNNs trained to predict enhancer activity (or chromatin accessibility) from sequence are useful as scoring functions to design functional enhancers. By using this DNN as a so-called oracle, ie. a scoring function, the main task of enhancer design becomes a sequence search algorithm problem.

Answer 2.3: Indeed, our work shows that DNNs trained to predict enhancer activity can be used in combination with simple search algorithms to yield functional enhancers with high success rate.

Remark 2.4: They tested 2 local search algorithms that previously existed, including a nucleotide-level greedy search (namely sequence evolution) and motif-level greedy search (i.e. motif embedding). They also tested a previously established GAN (previously established in one arxiv and tested by a different group in another publication, which was not cited; Zrimec et al. Nature Commun 2022), which provides a deep generative approach to sample more widely in sequence space instead of local search. Local searches have optimization guarantees to a local maximum, but no guarantee to a global maximum. The GAN in principle has the potential to sample very different regions of sequence space but this GAN was not conditional and so the generated sequences are not necessarily based on function. Instead, the training sequences for the GAN were biased to represent functional sequences. This work shows that local optima can yield highly functional enhancers! So, the search for global optima may be overblown or not necessary in terms of enhancer function as it pertains to a direct enhancing of a nearby minimal promoter. But this was not mentioned.

Answer 2.4: We agree that we find local optima and that they can yield highly functional enhancers. *We now better discuss this, as compared to global optima. Reference to Zrimec et al. has been added.*

Remark 2.5: Through an established model interpretability method, called deepexplainer, they perform motif analysis to observe how the distribution of motifs changes throughout the sequence optimization. They find an interesting observation that repressors get removed while activators get created. Despite the terms model interpretability or explanations, deepexplainer does not provide a full explanation of model predictions -- it provides independent nucleotide contributions -- not motif level contribution scores. Even though this is commonly done in deep learning papers in genomics, at least in the past, there is a hazard of over-interpreting extended patterns (like motifs) that emerge from (additive) attribution maps. Nevertheless they can be quite informative to generate hypotheses as part of an observational analysis. Importance of motifs and their interactions with other motifs in the same sequence can be quantified with carefully designed in silico perturbation experiments, and these sorts of analyses are becoming increasingly popular for model interpretability over the last several years in genomics (Avsec et al. Nat Gen 2021; Koo et al. PLoS Comp Biol 2021; deAlmeida et al. Nat Gen 2021; Toneyan et al. Nat Mach Intell 2022; Karollus et al. Genom Biol 2023; + many more). But this was not explored here.

Answer 2.5: We fully agree that DeepExplainer does not provide a full explanation. It is not the ground-truth and is not always trustable. As the reviewer writes, they can be "quite informative to generate hypotheses as part of an observational analysis", and this is how we also approached DeepExplainer. Indeed, we complement and test these hypotheses using a variety of complementary approaches (both in silico and experimentally), which we list for the reviewer below in a table. In situations where we only use DeepExplainer (observational), we make sure to write carefully and not over-interpret.

Our design strategies are independent from DeepExplainer scores. We use saturation mutagenesis and motif implantations and follow the prediction scores (not DeepExplainer results) to design enhancers, which are all purely in silico perturbation experiments.

Note also that many of the validations of DeepExplainer-suggested (and PWM confirmed) motifs from DeepFlyBrain and DeepMEL on genomic enhancers were presented in the respective publications that describe and validate these models, including Cut&Tag and DamID experiments as explained further below under Answer 2.6. The main focus of the current paper is enhancer design rather than motif interaction analysis. We have performed many such perturbation experiments in our previous publications by using the same deep learning models that we use in this study to design enhancers. In the current work, we find the same motif predictions from DeepExplainer (+ PWM) for synthetic enhancers. We agree that we should refer back to our earlier publications on genomic enhancers more often, when describing results on synthetic enhancers.

DeepExplainer validation approaches	Description
Motif annotation E.g.: Figure 2 	As requested in the first review, we explain how we annotate the motifs, namely by using an independent scoring method (Cluster-Buster, a HMM) with a position weight matrix. The motifs are still observational, however, they exactly match with the 'code table' of genomic Kenyon Cell enhancers from [Janssens et al. Nature, 2022], namely Ey+Mef2+Onecut+Sr as activators and Mamo (and others, unknown) as repressors. These were already validated by Cut&Tag and DamID in [Janssens et al. Nature, 2022]. Code table for KC from [Janssens et al. Nature, 2022]:  Glial enhancers motifs	This is observational (DeepExplainer plus PWM-based

E.g.: Figure 2

motif annotation).

However, when we compare the putative motif predictions with the code table of glia from [Janssens et al. Nature, 2022], we can clearly distinguish ttk motifs as repressors (removed by the design process), and ct+repo sites generated as candidate activators. Even though they are predictions, they are exactly the same motifs as found in genomic enhancers. Code table for glia from [Janssens et al. Nature, 2022]:

In silico saturation mutagenesis
E.g.: Figure 3

DeepExplainer + motif annotation (PWM based) is complemented here with in silico mutagenesis (in silico perturbation experiments), similar to the cited studies by the reviewer. The activator motifs suggested by DeepExplainer agree with decreases in prediction when these positions are mutated.

In vivo testing of repressor motifs
E.g.: Figure 3

The suggested repressor motifs by DeepExplainer are annotated using PWM-based scoring, and the indicated repressor sites were tested in vivo, showing that the enhancer is no longer functional.

Agreement with motif embedding
E.g.: Figure 4

DeepExplainer is supported by the in silico perturbation experiment, namely the motif embedding: exactly at the positions where the Mef2, Ey, and Onecut motifs are implanted, the DeepExplainer predictions are positive for the correct nucleotides comprising the motif.

In vitro saturation mutagenesis
E.g.: Figure 5

Like the *Drosophila* results in Figure 2, here DeepExplainer with PWM-based motif annotation is complemented with in silico mutagenesis. These data are then compared with experimental data, namely in vitro mutagenesis as determined by a massively parallel reporter assay (Kircher et al., 2019), showing that mainly mutations inside the annotated activator motifs cause decrease in activity; or isolated mutations that generate ZEB repressor sites.

Single mutations generate motifs similar to the ones observed in genomic enhancers
E.g.: Supplementary figures

DeepExplainer predictions are observational. They are used to show how the design via sequence evolution changes the enhancer step by step. The motifs are observational, but they are exactly the same (Ey, Mef2, Onecut) when generated by a few single mutations, as compared to genomic enhancers.

DeepExplainer and ISM comparison with other models
E.g.: Supplementary Note 1 (new)

What is also highly supportive of the DeepExplainer predictions provided by DeepMEL, is that an independently trained model, ChromBPNet, provides highly similar explanations. In addition, when our synthetic sequences are given to independent models (ChromBPNet and Enformer), they corroborate the predicted activity and the predicted motifs (both by DeepExplainer and ISM). If we mutate exactly the positions inside the predicted motifs, the independent models no longer score them high.

Concerning perturbations experiments, we used in silico saturation mutagenesis throughout the study:

- Figure 3 before and after generation of repressor motifs in evolved KC enhancer
- Figure 4 and Supplementary Figure 4 for motifs embedded KC enhancers
- Figure 5 for the IRF4 enhancer (confirmed with in vitro saturation mutagenesis)
- Supplementary Figure 2 for in silico evolved PNG enhancers
- Supplementary Figure 5 for GAN KC enhancers
- Supplementary Figure 6 for the IRF4 enhancer (with other DL models), the IRF4 enhancer following generation of repressor sites, and a genomic region containing a large number of repressor sites
- Supplementary Figure 8 before and after the generation of repressor sites in an in silico evolved MEL enhancer
- Supplementary Figure 11 for the rescue of human enhancers
- Supplementary Figure 13 for GAN MEL enhancers

Thus, we agree with this comment that DeepExplainer has to be used carefully and needs to be validated with other approaches.

We provide additional context in the discussion and wrote a supplementary note to give an overview of how the models explainability has been evaluated in our previous studies and to provide recommendations to users regarding the careful validation of DNNs before using them for synthetic design.

Remark 2.6: Nevertheless the experiments do validate that the generated sequences with putative motif patterns lead to functional activity in a cell type. The problem is this corresponds to the sequence and not necessarily the motif "explanations" that is hypothesized. Claims that are made of the motifs should be proven by placing the motif combination into different background sequences and seeing if those are functional. Otherwise, the tone should reflect that it is not a discovery but an observation from attribution analysis.

Answer 2.6: The reviewer proposed above that motif-based functional interpretations can be kept if “the claims are backed by follow up experiments that directly support the claims”, and otherwise should be toned down. We will follow this suggestion.

We provide a table with all the follow up experiments that directly support the claims, and for the other instances we now describe them as putative motifs (observational analysis).

Motif-level analysis	Experiment	Direct evidence or observational
Motifs for Ey and Mef are activator motifs in Kenyon cell enhancers	Kenyon cells are the only cells with high Eyeless & Mef2 expression; ey and Mef2 motifs are enriched in KC enhancers; Eyeless has a phenotype in KCs [Janssens et al. Nature 2022]	Observational
	Eyeless Cut&Tag finds the same motif [Janssens et al. Nature 2022]	Direct
	Mef2 DamID finds the same motif [Janssens et al. Nature 2022]	Direct
	Embedding of Ey+Mef in 6000 random sequences yields significantly high scores, and in vivo validation shows specific activity in Kenyon cells [This study]	Direct
	Embedding of Ey+Mef+Onecut motifs in a minimal 49bp sequence yields a functional enhancer [This study]	Direct (there is no other context besides the motifs)
	Changing the background sequences of the minimal 49bp sequence but keeping the same motifs on their exact location showed low scoring sequences contain repressor motifs [This study]	Observational
	In silico evolution yields functional enhancers with the same motifs for Ey and Mef2 [This study]	Observational (but highly unlikely to be chance)
	GAN-generated sequences with high scores also have Ey and Mef2 motifs [This study]	Observational (GAN work is removed from the main paper)
Repressor motifs decrease or abolish activity in Kenyon cells	Mutation of predicted mammo motif increases in vivo enhancer activity [Janssens et al. Nature 2022]	Direct
	Mutations in functional synthetic enhancers that generate the same repressor motifs are no longer functional in vivo [This study]	Direct
	Random sequences are enriched	Observational (but not only based

	for repressor motifs [This study]	on DeepExplainer, also confirmed with PWM enrichment analysis)
Repo and other TFs are activator motifs for perineurial glia enhancers	In silico evolution yields functional enhancers with motifs for repo and other TFs that were identified in the code table of Janssens et al. Nature 2022 [This study]	Observational
Activator motifs of T-neurons can be combined with KC code	Dual code enhancer was generated by in silico evolution of a KC enhancer [This study]	Observational
SOX10, MITF, and TFAP2A are activator motifs for melanocyte/melanoma (MEL) enhancers	Mutations in SOX10 or MITF motifs from genomic region decrease activity in MEL lines [Mauduit et al., eLife 2021]	Direct
	Embedding of SOX10 and MITF motifs in random sequences generate functional enhancers in MEL lines [Mauduit et al., eLife 2021]	Direct
	Embedding of SOX10, MITF, and TFAP2A motifs in 2000 random sequences yields significantly high scores and in vitro validation shows activity in MEL line [This study]	Direct
	In silico evolution yields functional enhancers with the same motifs for SOX10, MITF, and TFAP2A [This study]	Observational (but highly unlikely to be chance)
	Shortening of an enhancer with embedded SOX10+MITF+TFAP2A motifs into a minimal 51bp sequence remains a functional enhancer [This study]	Direct (there is no other context besides the motifs)
ZEB motifs are repressor motifs in MEL enhancers	ZEB motifs introduced in functional synthetic MEL enhancers decrease/abolish their activity [This study]	Direct
	Tiling experiment highlight the repressor effect of ZEB motif in genomic sequences in melanoma [Mauduit et al., eLife 2021]	Direct
	ZEB2 ChIP-seq performed that ZEB2 is the TF binding to ZEB repressor motifs in MEL melanoma [This study]	Direct

We adapted the text to distinguish motif-level interpretations with direct evidence versus those that are observational.

We already performed the experiments where we keep the same TFBSs, their distance, affinity, and number and changed only the proximal background sequences in between the TFBSs (Fig. R2.1). It showed that the sequences that have lower prediction scores are the ones that the background sequence creates repressor binding sites with clear motifs such as GATC, MAMO, and AAGA. Which is again not about 'contextual' sequence but motif content of an enhancer.

Figure R2.1 (Supplementary data): **a**, Distribution of prediction score for the 49 bp short enhancer with 1,000,000 random background. Dashed lines indicate prediction scores for the 1st (orange), 50,000th (green), 200,000th (red), 500,000th (purple), and 999,900th (brown) worst background. **b**, Nucleotide contribution score (top) and in silico saturation mutagenesis (bottom) of the 1st, 50,000th, 200,000th, and 500,000th worst background.

Remark 2.7: The GAN was largely unable to capture the regulatory grammars that make up strong enhancers, but it was still better than a poorly chosen random baseline (because the model's positional bias was not considered in the null model). However, since generation occasionally leads to high activity by chance, an inefficient, yet effective scheme was to use the DNN scoring function to subselect the rare sequences that yield high functional predictions (i.e. tail behavior of the distribution). This of course can also be done with random sequence generation, at a lower efficiency. This would not be considered a good generator in any machine learning field. Again, it is the oracle scoring function that enables the identification of functional enhancers, i.e. what the authors call enhancer design.

Answer 2.7: We agree that the GAN approach is more efficient than scoring random sequences but not as good as DNN's scoring function. It is also less interpretable compared to the first two techniques, which we used to observe and understand what causes a random sequence to become an enhancer.

In Remark 2.14, the reviewer suggests removing the GAN part from the main text. We concur with this request but would still like to keep the GAN results as a supplementary note if possible.

The GAN results and discussion have been removed from the main text and are now presented as Supplementary Note 2. Data has been removed from the main figures 4 and 6 and have been placed in supplementary figures.

A broader set of critics:

Remark 2.8: Despite making it a point in the first review that code and data was not provided, there is only an illusion that meaningful code is provided in the revision. The provided code is for plotting figures and even then, it still does not contain any of the necessary files to actually run the notebooks. This is unacceptable. In fact, even if the

intermediate files are provided, the code would still be broken as there are missing functions. For instance, the notebook FLY_KC_Repressors.ipynb calls `utils.model_load` but that function doesn't exist in `utils.py`. The zenodo link to code w/ intermediate files is restricted and inaccessible to the reviewer. This is `_unacceptable_!`

Answer 2.8: We kindly disagree with this remark. When one downloads the intermediate files from Zenodo, all the provided notebooks that perform every analysis from scratch and plot every figure in the paper are just in a click-and-run mode. No modifications, no additional codes/functions, no other intermediate files are needed.

We double-checked again and all the functions are there in the `utils` file. Perhaps the reviewer mixed different `utils` files under different folders. Mainly we provide two things; (1) very general notebooks to show how we perform our 3 different enhancer design strategies, (2) notebooks that reproduce every analysis and figure in the final version of the paper. The `utils` files are different for these two different notebook folders.

We apologize that the data files were inaccessible to the reviewer via Zenodo. They were too large (since they contain all the intermediate values as `.pkl` files, 21Gb compressed) to add to the supplement. We now provide the same file as we shared via Zenodo, through a Google Drive, that was prepared by the Editor:

- <https://drive.google.com/drive/folders/1C9wT7PKdQxR58Hrhz1ZVGV-o0wbq3cyo?usp=sharing>

In conclusion, all the code is provided as a supplement for the reviewer, and all the data (and code) can be downloaded via the Google Drive URL. Upon acceptance all data and code will be shared publicly with the community.

Remark 2.9: The main contribution is supposed to be a set of sequence design methods that can be used to design cell type specific enhancers, but I have deep concerns that this is not a general solution. Despite the proof of concept for the designed enhancers validated in vivo, the objective function only considers the cell type of interest and it does not explicitly repress activity in other cell types during the optimization process. This can happen if the cell type chosen has enhancers that use a unique set of motifs. In other scenarios, off-target effects can occur. The benefit here is the experimental validation. The fact that it worked for the chosen cell types might say something about the orthogonality of motifs in enhancers across the cell types explored by DeepFlyBrain. But this requires additional experiments to assess if this is general. I have doubts that this holds across cell types that have overlapping TFs employed in enhancers.

Other approaches that consider cell type specific enhancers directly deal with this effect by including a minimization term for off-target cell types (Gosai et al. *bioRxiv* 2023). In any case, the claims of novelty about cell type specificity should be toned down as cell type specific enhancer design means targeting a cell type of interest and `_not_` other cell types. This basic approach does not solve the general problem of cell type specific enhancer design and does not push the field forward from where it already was.

Answer 2.9: It is correct, when a model is trained on all ATAC peaks or all candidate sequences (MPRA), then a minimization term for off-target cell types would be needed, as is done in Gosai et al. 2023. Note that this preprint was published on 9 August 2023, after we submitted our revised version.

It may indeed be surprising that even though our objective function does not explicitly suppress off-target cell types, the generated enhancers are exquisitely cell type specific. This is a rather technical, but indeed an interesting topic that we can add to the text. We can explain this with three reasonings; (1) because our models are trained on topics and cell type specific regions. In other words, our models are specifically trained to distinguish cell type specific ATAC peaks. This is different from models like ChromBPNet, DeepSEA, Basset, etc., where they are trained to predict every ATAC peak on a given cell type. (2) The model generates a unique combination of motifs for the targeted cell type during the in silico evolution. We also implant the same unique combination during the motif implantation experiments. For example, Mef2 and Onecut are expressed in many cell types in the fly brain, but they are only expressed with Eyeless in Kenyon Cells. Enhancers with [Mef2, Onecut, Ey] motifs are specifically active in Kenyon cells, and are thus not activated by Mef2 alone. Our in silico experiments with motif embedding also confirm this by DNN scores. (3) The model does not remove repressor motifs that are bound by repressor TFs in other cell types just as it does not create activator motifs bound by activator TFs in other cell types.

Also note that for human enhancers we used a simple objective function, and the synthetic enhancers designed were also exquisitely specific across the 674 DNase classes in Enformer.

In agreement with the reviewer's question, we have encountered a situation in the lab where we had to adapt the objective function. Particularly, we trained DNNs on the mouse cortex, and we encountered more noisy topics with mixtures of enhancers and promoters. When we designed enhancers using this model, we used another objective function (z-score) to suppress off-target cell types. In fact, this option was already present in the code we provided (but it was not described in the text). Thus, for the users of our code, they can already choose between a simple objective function (for models that are trained specifically to predict cell type specific accessibility), or the z-score that suppresses off-target cell types. We agree that this is an interesting addition to the text.

The extent to which the motifs between cell types need to be unique is difficult to assess and is outside the scope of our work. We agree that some degree of orthogonality is needed - this is also the focus of our work and relates to cell type specific enhancers. We do not think that the motif set must be 100% unique, because we already observe many shared TFs between cell types (such as Mef2). The encouraging aspect is that many cell types already work (including additional unpublished ones by us and others). The future will probably tell how high we can 'saturate', and which improvements are needed to design enhancers for *all* cell types.

Finally note that when two cell types become so similar that their ATAC-seq profiles cannot be distinguished, then a DNN trained on such data cannot discriminate regulatory regions between the cell types based on the sequence. But that is trivial. Our simple objective function can generate functional enhancers for each cell type that the model can discriminate.

In conclusion, we use (and favor) multi-class (multi cell-type / modality) approaches in which the same model can be used to observe if there are any off-target effects (as compared to single-class models trained on the full ATAC-seq profile). And it is a single command line to update in the code we provide to change the objective function to maximize the z-score of the prediction score compared to other classes, rather than focusing on only one class. Still, our results show that when the models are trained on ATAC-seq topics (or differentially accessible regions), a basic approach can solve this problem through a multi-class model and advances the field forward by showing activator-repressor balance in between different cell types in detail.

We have updated the text to explain the two types of objective function, and that the suppression of off-target cell types is more important for models that are trained on ATAC-seq peaks rather than models trained specifically to predict cell-type specific enhancers.

Remark 2.10: Many general claims of motif grammars are based on observational motif analysis of attribution maps. I would not have had many objections a few years ago. But the standards of model explanations have improved over the past several years and cursory motif analysis solely based on observational analysis should be supplemented with interventional analysis with in silico experiments as done in: Avsec et al. Nat Gen 2021; Koo et al. PLoS Comp Biol 2021; deAlmeida et al. Nat Gen 2021; Toneyan et al. Nat Mach Intell 2022; Karollus et al. Genom Biol 2023; + many more. The over interpretation of attribution maps is a hazard, they come with many insights but also have intrinsic pathologies that are well known in the broader ML community and it is still taking time time to reach the broader genomics community.

Answer 2.10: We respectfully disagree that our motif grammar claims are purely based on DeepExplainer attribution maps (and we agree with the reviewer to be careful with DeepExplainer predictions). Our motif analysis uses PWMs and Hidden Markov Models, and we use many complementary assays and independent models and approaches to validate motif-based predictions.

We have also performed many interventional analyses with in silico experiments as suggested by the reviewer, in this work and in the publications of the DeepMEL and DeepFlyBrain models. These have been summarized in a table under **Answer 2.6**.

Remark 2.11: The validation of the designed sequences in vivo, which contain the putative motifs, demonstrates that the (whole) designed sequence is functional but only indirectly supports motif-level claims.

Answer 2.11: We agree that the in vivo validation of the designed sequences (using in silico evolution) demonstrates that the sequence is functional and cell type specific, but not directly validates motif-level claims.

We improved the text to make this clear.

Some notes on this matter, in support of certain motif-level claims (not from the in vivo validations):

- The validation of designed sequences generated by motif-embedding are already closer to motif-level claims because we explicitly insert specific motifs, we know exactly what and where they are inserted, and they were inserted in many different random backgrounds.
- In addition to the motif implantation (where we directly assess the motif), we found that the mutations that were performed during the in silico evolution approach also generate or destroy motifs, and this observation was not based on DeepExplainer. One can also think about this the other way around: starting from an active synthetic sequence, when mutations are targeting specifically the (predicted) motifs to destroy activators (or create repressors), and every single time such mutations decrease the 'oracle' score of the DNN (and of alternative models like Enformer and ChromBPNet) and eventually destroys the activity completely.
- We did perform several specific motif-level changes that we tested experimentally, which we included in the revision. These included new in vivo, in vitro, and in silico validations of predicted motifs. For example, we added repressor motifs to a designed functional sequence, generated new transgenic fly lines, and demonstrated that the enhancer is no longer functional. We also generated human sequences with different motif compositions and confirmed that activator motifs increase activity. Also mutations that specifically destroy activators have been performed [Janssens et al. Nature, 2022]. See also Answer 2.6.
- Each and every synthetic design from any of thousands of random sequences ends up having the same motifs, which by itself is, we believe, in support of our motif-level claims. Motifs found by DeepExplainer+TFMoDISCO are also found by HMM-scanning and by Homer (for Homer, the results are present in the notebooks we provide, but not included in the manuscript).
- When the synthetic sequences are given to independent models, they corroborate the predicted activity and the predicted motifs, both by DeepExplainer and ISM (Fig. R2.2). If we mutate the motifs on the active synthetic sequences, by following the mutational steps backwards, the other models no longer score them high (Figure 6g-j. Supplementary Figure 9a-c).

Figure R2.2 (Supplementary Note 1 – Figure 1): **Comparison of DeepExplainer and in silico mutagenesis from different models**

Nucleotide contribution scores and in silico saturation mutagenesis assays of DeepMEL2, ChromBPNet, and Enformer for IRF4 (**a**) and EFS-4 (**b**). In vitro saturation mutagenesis from Kircher et al. 2019 is also displayed for IRF4. Each dot on the saturation mutagenesis plots represents a single mutation and its effect on the prediction score (y axis). Motif annotation is indicated with strong (s) or weak (w) motif instances.

Remark 2.12: Validation of motif-level claims could be supported with additional experiments both in silico (and/or in vivo). Ideally, a set of motifs would be implanted in different random sequences (that are sampled from a good null distribution; see other comments below for this). Then the average activity across contexts could inform the motif's effect size. These experiments can be done in silico with model predictions to prove that the model has indeed learned these features or in vivo to provide supporting evidence that this reflects real biology.

We appreciate that the reviewer gives the option to perform these experiments in silico, since we already provided so many in vivo validations in the 1-yr revision, and additional in vivo experiments would take another 6-9 months.

The reviewer writes “Ideally, a set of motifs would be implanted in different random sequences”. This is exactly what we did in our paper: a considerable fraction of the paper is about motifs being implanted in different (thousands) random sequences.

Regarding the null model, we do already use a good null distribution, namely a position-specific GC content matched random sequence (Figure R2.3). Only sequences generated using a Markov model in the GAN part of the study are not GC-adjusted. Note that random sequences, GC-adjusted or not, have prediction scores that are almost zero. Without having motifs, they are just random sequences as in the background sequences of genomic enhancers.

Figure R2.3 (Supplementary Figure 1a): **a**, Distribution of GC-content in GC-adjusted random sequences (green) and genomic regions (red).

We added an additional in silico experiment regarding motif implantations in random locations on random sequences. We implanted motifs at random locations (rather than the best locations as dictated by the model), this is provided in Answer 2.13.

Remark 2.13: For motif embeddings, a good control is to implant numerous activator motifs (from the same set) in random locations. I would suggest implanting more than one of each motif, perhaps up to two or three rounds and measure the design quality based on the oracle predictions. My suspicion is that it would also promote a high cell-type specific function, but perhaps would be less efficient than the proposed greedy search.

We completely agree that implanting numerous activator motifs in random locations will yield a subset of high-scoring sequences. We performed a similar experiment in a previous study (Mauduit et al, eLife 2021). We implanted multiple instances of SOX and MITF motifs at fixed locations and distances in random sequences and observed enhancer activity in MEL lines that was largely dependent on motif number and SOX/MITF cooperation (Figure R2.4).

Figure R2.4 (Mauduit et al. eLife 2021; Figure 5e; Figure 5 - Supplementary Figure 5a,b): **e**, Top panel: cartoon of SOX and MITF motif combinations in a background sequence. Middle and bottom panels: CHEQ-seq activity of synthetic enhancers with background sequence 2 in MM001, MM074, and MM087 sorted by the number of SOX (middle panel) or MITF (bottom panel) motifs present in the sequence. Dashed line indicates the log₂ FC value of the background sequence without any motif. **a**, Synthetic combinations of SOX and MITF motifs DeepMEL2 prediction scores for topic 16 with scores ordered by the number of SOX motifs in the sequence. **b**, Synthetic combinations of SOX and MITF motifs DeepMEL2 prediction scores for topic 17 with scores ordered by the number of MITF motifs in the sequence.

Even pure random sequences without motifs implanted, if tested by the millions or billions, will contain high-scoring sequences that will be functional (and these will have the same motifs, by chance, as shown by the Stark group in the DeepSTARR paper). Such approaches are less efficient than the greedy search that selects the best location per motif, and they provide less insight into local optimum (e.g., we observed that implantation of a motif onto a repressor motif is more efficient).

As suggested, we performed additional implantation experiments using random locations, and found that when the same motif combinations are implanted at random locations, the scores are significantly lower compared to the scores obtained using the DNN to choose the optimal locations (median score of 0.970 vs 0.075) (Figure R2.5). This is most likely due to the random implantation approach not managing to remove the repressor motifs and not obtaining an optimal arrangement of (e.g., distance between) activator motifs.

Figure 4a has been replaced with the panel from Figure R2.5.

Figure R2.5 (Figure 4a): Prediction score distribution of the sequences for the γ -KC class ($n = 2,000$) after each motif implantation at best location (blue) or at random location (orange) and after 15 mutations. Abbreviations are used for Ey (E), Mef2 (M), Onecut (O), and Sr (S). The box plots show the median (center line), interquartile range (box limits), and 5th and 95th percentile range (whiskers).

Remark 2.14: The GAN analysis is a negative result. The predictions of the GAN-generated sequences seem to be substantially lower than the predictions of motif embedding or sequence evolution. However, it remains difficult to draw any strong conclusions from this as the GAN hyperparameters were not thoroughly optimized -- so is the GAN methodology flawed or is it just that it a good setting was not found? The presented comparison with poorly chosen baseline model is not demonstration that the GAN works. Since strong statements cannot be made either way, **it might be better to remove GANs from the story to simplify an increasingly complex story**. The GAN requiring an oracle scoring function means that it is just slightly more efficient at occasionally generating a functional sequence (i.e. from tail behavior of the generated sequence distribution) than a "poorly" chosen null model. Also, here is another example of GAN for DNA regulatory sequences: Zrimec, Jan, et al. "Controlling gene expression with deep generative design of regulatory DNA." Nature communications 13.1 (2022): 5099.

Answer 2.14: We agree that the GAN is the least convincing of the 3 approaches we have explored. This attempt at generating synthetic enhancers via the GAN could still be interesting to the reader who would like to tackle such a challenge. This is why we would like to keep those results as a supplementary note. That way, it will not contribute to the increasing complexity of the story and remain accessible to potential interested readers.

The GAN results and discussion have been removed from the main text and are now presented as Supplementary Note 2. Data has been removed from the main figures 4 and 6 and have been placed in supplementary figures.

Remark 2.15: The lack of consideration for the ML field focused on designing biological sequences makes this paper read like it is less thoughtful about how it fits into the broader field. Moreover, there are serious drawbacks and limitations of using supervised DNNs as oracles, which are well established (eg. this recent manuscript, "Is novelty predictable?" by Fanning and Listgarten, but there are many more over the past several years). This manuscript reads as an over-confidence in the previously established DNNs which are used as oracles for sequence design, in this case DeepFlyBrain and DeepMel. However, it is well known that the reliability of model predictions changes under covariate shifts from the training distribution, that any interpretability is strictly through lens of DNN and may not be a reflection of the full biological process. An argument against this could be that it just works according to some validation. But the cases that the designed sequences failed may reflect pathological regions of the DNN-inferred sequence-function landscape. This was not explored.

Answer 2.15: We agree that we do not explore the relationship between enhancer design and other advances in the ML field regarding design. Given the space constraints and the focus of the paper, we find this topic more relevant for a review article and consider it outside the scope of our work.

Note that we do not argue that the models we use are 100% accurate all the time. Otherwise, we wouldn't perform any in vivo or in vitro experiments. Now we add more detail to the discussion that without rigorous experimental validation, researchers shouldn't blindly trust the predictions and interpretations of the DL models.

Other concerns:

Remark 2.16: The baseline of 5th order Markov is clearly not the right null model given the positional GC bias that DeepFlyBrain learns. A more suitable null should be random sequences that match the GC content per position.

We already partially addressed this remark in Answer 2.12.

We also checked whether random sequences with and without GC bias correction give a different prediction score, but that is not the case, both are always close to zero. The 5th order Markov model we only used to compare with the GAN sequences.

Remark 2.17: "Note that besides the greedy search that selects the best mutation at each iteration, alternative state space searches can be employed that may yield, for the same number of mutations, enhancers with higher scores, albeit with higher computation cost " There is no description of this method and it is written so vaguely that I have no idea what alternative search was used.

Answer 2.17: This new experiment was performed in response to a remark from reviewer 1. We kept it short in the result section to not increase the complexity of the story. However, the method section details the procedure applied to generate the sequences: "To explore the alternative in silico sequence evolution paths besides choosing the best mutation (greedy algorithm), we chose the top 20 mutations on each sequence for every step starting from a random sequence. We followed this procedure for 5 mutations. Starting from 1 random sequence, we obtained 3.2 million paths/sequences at the end."

We now reference the Methods section in the results, expand the captions of the associated figure panels, and rephrased part of the methods that could have been confusing.

Remark 2.18: abstract: "we could accurately define the enhancer code as the optimal strength, combination, and relative distance of TF activator motifs, and the absence of TF repressor motifs"  the validation performed does not prove optimality, just functionality.

Answer 2.18: We agree that this does not prove optimality, which we did not mean to say, but the word "optimal" is misleading. We indeed show that local optima and multiple different arrangements yield a functional enhancer.

We changed this sentence and replaced the word "optimal" to reflect this.

An interesting discussion we had multiple times after presenting this work at conferences, is whether genomic enhancers may actually be sub-optimal. Indeed, for most genomic enhancers, we can identify mutations that increase their strength. Optimality can thus mean different things, such as optimal in regulating the target gene at the correct level to yield a particular phenotype. We include this in the discussion.

Remark 2.19: The work can sell itself without over-the-top language that is currently used throughout. Abstract: "well-trained models are reaching a level of understanding that may be close to complete" This should be removed and the other superlatives throughout should be minimized. (Examples of how sequence-based DNNs are failing and not close to complete: Karollus et al. Genome Biol 2023; Huang et al. bioRxiv 2023; Sasse et al. bioRxiv 2023)

Answer 2.19: We have gone through the text and adjusted over-the-top claims or statements.

This includes:

- Removing "and well-trained models are reaching a level of understanding that may be close to complete"
- Changing "we discovered here that some enhancers have evolved with multiple intertwined codes"
- Minimizing the superlatives throughout the manuscript

Minor comments:

Remark 2.20: For enhancer design towards multiple cell type codes: how were the mutations in the sequence evolution chosen? The description in the main text and the methods are exactly the same and no clear description of the implementation or the objective function is given.

Answer 2.20: Correct, because the experiment was performed on a couple of sequences only, we did not develop an automated pipeline for the dual-code enhancers. The mutations were selected manually and are shown in Suppl. Data Figure 8.

We adjusted the Methods section to clarify that the selection of the mutations was done manually.

Remark 2.21: It would be helpful to see the attribution maps on the same scales, especially when claims are made based on comparing different attribution maps. (eg. Fig 4e among many others).

Answer 2.21: The attribution scores are scaled with the prediction scores and when the prediction scores are lower compared to fully evolved sequences, it is impossible to observe what has been created as a motif. We believe that our scaling is more informative to the reader as they can see the emerging motifs as well as what the scale is.

Remark 2.22: Fig 4e axis labels are not there.

Answer 2.22: Labels are present in all DeepExplainer profiles with the x axis displaying nucleotide position every 10 or 20 nucleotides and the y axis displaying the maximum tick mark, minimum tick mark and 0 tick mark values of the nucleotide contribution score. The reviewer is maybe referring to axis titles instead of labels. Figure captions mention that the values displayed on the plot are nucleotide contribution scores. If required, we can add axis titles to all the DeepExplainer plots.

Referee #3 (Remarks to the Author):

In this revised manuscript, Taskiran et al. have undertaken a number of revisions to address topics that were raised in the first round of review of this manuscript. The manuscript now includes significantly more validated enhancers, including activity of elements that featured intermediate scores during in silico evolution. An expanded discussion of previous related studies better integrates this work into the field. Expanded ATAC-seq analysis demonstrates the concordance between activity and chromatin open-ness in vivo. ZEB2 is identified as the likely repressor involved in IRF4 enhancer regulation in melanocytes. Regarding conclusions for numbers or arrangements of motifs in active elements, the manuscript now provides convenience summaries of numbers and types of motifs, although subtleties of arrangement are not identified.

We thank the reviewer and we are pleased to hear that we addressed the topics that were raised in the first round of review.

With regard to comments from Reviewer 3, the authors addressed the question of whether the cis regulatory elements that were developed indeed act as enhancers by testing activity in silico. These models predict that IRF4 elements should perform similarly in proximal or distal locations. (The authors were unaware of studies that indicate more promiscuous activity for minimal elements at a promoter position; for Dorsal and Twist activators, Szymanski & Levine 1995 describes one example of this frequently-observed phenomenon; the minimal MSU element lacks activity distally, and only synergizes with an existing promoter-proximal MSU when placed in an adjacent, not distal, position).

Indeed, we observed the same phenomenon with our in silico CRISPR experiments using Enformer where we move the enhancer around from its original location. When the enhancer is located next to the promoter of *Irf4* gene, it increases the expression the highest compared to its original location (intergenic) or distally upstream/downstream of the gene (Supplementary Figure 9c).

In the context of whether the computational approaches here will succeed with only “simple” enhancer design, the paper now discusses similarities of the fly and human findings, leading to the conclusion that these functional properties are likely broadly applicable. At the same time, they note that their assays lack the fine spacial, temporal and quantitative resolution of Fuqua et al. for *svb*, thus there may be additional properties still to identify computationally.

We agree with the reviewer that our fly related assays and models do not measure the temporal and quantitative resolution just as our human related assays lack spatial and temporal resolution. Now we extend the discussion mentioning the need of organism-wide deep learning models during development, homeostasis, and disease to fully uncover the additional properties of the enhancer code.

The revised manuscript also better develops insights into the likely balance between repressors and activators at active elements, providing more experimental evidence for assemblies that are dominantly regulated by repressors.

Two reviewers asked about lessons from prediction “failures”, and new data in this paper shows that such inactive elements are just a few mutations away from activity in vivo, indicating the inherent potential of the evolved elements.

Finally, a number of technical points regarding computational approaches, vector design and small writing issues are addressed. Overall, this manuscript describes a significant advance in our understanding of the potential and limitations to identification and ab initio generation of tissue-specific regulatory elements (likely to act as enhancers), and furthermore provides a rich dataset of in silico evolved de novo or genome-derived sequences to undergird studies of enhancer function. By applying these methods to two very different systems, the authors have provided evidence for the applicability of deep learning for cis regulatory elements in metazoans, and their findings will be of high interest to researchers in computational and systems biology, development, and evolution.

We thank the reviewer for their kind words.

Reviewer Reports on the Second Revision:

Referees' comments:

Referee #1 (Remarks to the Author):

I find the additional work performed by the authors in addressing random mutations to be greatly beneficial in supporting the guiding principles of the proposed model. All my concerns have been satisfactorily addressed.

Referee #2 (Remarks to the Author):

The authors have addressed many of my initial concerns. The revised text reads much better now and the code and data is available to the reviewer via the google drive. It is my expectation that the zenodo link in the paper (which is inaccessible for review) should reflect the same files/data provided by the google drive upon publication.

Minor comments:

- Information about ChromBPnet and Enformer should be given in the Methods, including which track they took, which sequences (i.e. chrom coordinates) were chosen for input sequences, how the profile predictions were summarized, etc.
- Some points of confusion that I have in inspecting the code is that the authors point to kipoi.org for the models in the main manuscript under Data availability section. However, there is no import kipoi in the author's code within the supplementary code directory Fly_brain. Moreover, the Kipoi-deposited models are named differently than the models provided in the google drive. Even if they are the same models, the lack of clarity and the misalignment of the text and code make reproducibility unnecessarily challenging.
- The jupyter notebooks and code has minimal (in many cases non-existent) comments for the code. This provides unnecessary burden on users aiming to replicate this work. As this works primary contribution is computational, better code practices should be expected. I suggest the authors consider raising their code standards to align with the Comment "Reproducibility standards for machine learning in the life sciences" by Heil et al in Nature Methods 2021. link: <https://www.nature.com/articles/s41592-021-01256-7><<https://www.nature.com/articles/s41592-021-01256-7>>
- A comprehensive readme file is needed to navigate the code. It should inform the analysis performed in each notebook. It should inform users what order to execute notebooks if there are any dependencies. Importantly, the readme should list of dependencies, versions of python modules used (due to compatibility issues), etc.
- In third sentence in section Enhancer design by in silico evolution, the authors use Fitness scores to

refer to DeepFlyBrain's predictions. For clarity and precision, the authors should use something else like enhancer activity score or differential accessibility score.

- In Human enhancer design section, the authors refer incorrectly to Supp Note 1 in the second to last paragraph; it should be 2.

Author Rebuttals to Second Revision:

Referees' comments:

Referee #1 (Remarks to the Author):

I find the additional work performed by the authors in addressing random mutations to be greatly beneficial in supporting the guiding principles of the proposed model. All my concerns have been satisfactorily addressed.

We are happy that all concerns have been addressed. We thank the reviewer for their time and insightful comments.

Referee #2 (Remarks to the Author):

The authors have addressed many of my initial concerns. The revised text reads much better now and the code and data is available to the reviewer via the google drive. It is my expectation that the zenodo link in the paper (which is inaccessible for review) should reflect the same files/data provided by the google drive upon publication.

We are pleased that the Reviewer's concerns have been addressed and the text reads better now. The zenodo link in the final version of the paper reflects the same file provided but now with comments and readme files. We thank the reviewer for their time and insightful comments.

Minor comments:

- Information about ChromBPnet and Enformer should be given in the Methods, including which track they took, which sequences (i.e. chrom coordinates) were chosen for input sequences, how the profile predictions were summarized, etc.

Done.

- Some points of confusion that I have in inspecting the code is that the authors point to kipoi.org for the models in the main manuscript under Data availability section. However, there is no import kipoi in the author's code within the supplementary code directory Fly_brain. Moreover, the Kipoi-deposited models are named differently than the models provided in the google drive. Even if they are the same models, the lack of clarity and the misalignment of the text and code make reproducibility unnecessarily challenging.

Data availability section is updated. The Zenodo source of the models are mentioned in the readme files now.

- The jupyter notebooks and code has minimal (in many cases non-existent) comments for the code. This provides unnecessary burden on users aiming to replicate this work. As this works primary contribution is computational, better code practices should be expected. I suggest the authors consider raising their code standards to align with the Comment "Reproducibility standards for machine learning in the life sciences" by Heil et al in Nature Methods 2021. link:

<https://www.nature.com/articles/s41592-021-01256-7><<https://www.nature.com/articles/s41592-021-01256-7>>

The jupyter notebooks are updated and now include comments for the code.

- A comprehensive readme file is needed to navigate the code. It should inform the analysis performed in each notebook. It should inform users what order to execute notebooks if there are any dependencies. Importantly, the readme should list of dependencies, versions of python modules used (due to compatibility issues), etc.

Comprehensive Readme files are added. They include the versions of the python modules, descriptions of each notebook, mentioning if there are any dependencies between notebooks, and the location of the input-output files as well as the output figures as pdf.

- In third sentence in section Enhancer design by in silico evolution, the authors use Fitness scores to refer to DeepFlyBrain's predictions. For clarity and precision, the authors should use something else like enhancer activity score or differential accessibility score.

Done.

- In Human enhacer design section, the authors refer incorrectly to Supp Note 1 in the second to last paragraph; it should be 2.

Done.